# Differential response of carbon cycling to long-term nutrient input and altered hydrological conditions in a continental Canadian peatland

Sina Berger[1,2,3], Leandra Praetzel[1,2], Marie Goebel[1,2], Christian Blodau[1,2,†], Klaus-Holger Knorr[1]

[1] University of Muenster, Institute of Landscape Ecology, Ecohydrology and Biogeochemistry Group, Heisenbergstraße 2, 48149 Muenster, Germany

[2] University of Guelph, School of Environmental Sciences, 50 Stone Road East, Guelph, Ontario, N1G 2W1, Canada

[3] Karlsruhe Institute of Technology, Institute of Meteorology and Climate Research (IMK-IFU), Kreuzeckbahnstraße 19, 82467 Garmisch-Partenkirchen, Germany

*Correspondence to*: Sina Berger (gefleckterschierling@gmx.de), Klaus-Holger Knorr (kh.knorr@uni-muenster.de)

**Abstract.** Peatlands play an important role in global carbon cycling, and their responses to long-term anthropogenically changed hydrologic conditions and nutrient infiltration are not well known. While experimental manipulation studies, e.g. fertilization or water table manipulations, exist on the plot scale, only few studies have addressed such factors under in-situ conditions along gradients within larger sites. Therefore, an ecological gradient from center to periphery of a continental Canadian peatland bordering a eutrophic water reservoir, as reflected by increasing nutrient input, enhanced water level fluctuations, and increasing coverage of vascular plants, was used for a case study of carbon cycling along a sequence of four differently altered sites. Here we monitored carbon dioxide ($CO_2$) and methane ($CH_4$) fluxes at the soil/atmosphere interface and dissolved inorganic carbon (DIC) and $CH_4$ concentrations along peat profiles from April 2014 through September 2015. Moreover, we studied bulk-peat and pore-water quality and we applied $\delta^{13}C$-$CH_4$ and $\delta^{13}C$-$CO_2$ stable isotope abundance analyses to examine dominant $CH_4$ production and emission pathways during the growing season of 2015. We observed differential responses of carbon cycling at the four sites, presumably driven by abundances of plant functional types (PFTs) and vicinity to the reservoir. A shrub dominated site in close vicinity to the reservoir, was a comparably weak sink for $CO_2$ (in 1.5 years: -1093 ±794, in 1 year: +135 ±281 g $CO_2$ m$^{-2}$ (=net release)) as compared to two graminoid-moss dominated sites and a moss dominated site (in 1.5 years: -1552 to -2260 g $CO_2$ m$^{-2}$, in 1 year: -896 to -1282 g $CO_2$ m$^{-2}$). Also, the shrub dominated site featured notably low DIC concentrations along peat pore-gas profiles as well as comparably $^{13}C$ enriched $CH_4$ ($\delta^{13}C$-$CH_4$: -57.81 ±7.03 ‰) and depleted $CO_2$ ($\delta^{13}C$-$CO_2$: -15.85 ±3.61 ‰) in a more decomposed peat, suggesting a higher share of $CH_4$ oxidation and differences in predominant methanogenic pathways. The graminoid-moss dominated site in closer vicinity to the reservoir featured a in comparison to all other sites by ~30 % increased $CH_4$ emission (in 1.5 years: +61.4 ±32,

in 1 year: +39.86 ±16.81 g $CH_4$ m$^{-2}$), and low $\delta^{13}C$-$CH_4$ signatures (-62.30 ±5.54 ‰), indicating only low mitigation of $CH_4$ emission by methanotrophic activity here. Methanogenesis and methanotrophy appeared to be related to the vicinity to the water reservoir: the importance of acetoclastic $CH_4$ production apparently increased toward the reservoir, whereas the importance of $CH_4$ oxidation increased toward the peatland center. Plant mediated transport was the prevailing $CH_4$ emission pathway at all sites even where graminoids were rare. Our study thus illustrates an accelerated carbon cycling in a strongly altered peatland with consequences for $CO_2$ and $CH_4$ budgets. However, our results suggest that long-term excess nutrient input does not necessarily lead to a loss of the peatland C-sink function.

## 1 Introduction

Since the end of the last glaciation, northern peatlands have played an important role in global carbon (C) cycling by storing atmospheric carbon dioxide ($CO_2$) as peat, but also emitting significant amounts of C as methane ($CH_4$) (Succow and Joosten, 2012). Carbon sequestration and $CO_2$ and $CH_4$ release are driven by numerous processes and the accumulation of peat results from only a small imbalance of photosynthetic carbon uptake over respiratory losses. $CO_2$ can be released through autotrophic and heterotrophic respiration under aerobic and anaerobic conditions (Limpens et al., 2008). Controls on heterotrophic respiration have been intensively studied and depend e.g. on temperature, substrate quality, energetic constraints and other factors (Blodau, 2002). Methanogenesis is strictly limited to anaerobic conditions (Conrad, 2005) due to thermodynamic controls, $CH_4$ production is only competitive upon depletion of alternative, energetically more favorable electron acceptors for anaerobic respiration, such as nitrate, iron, sulfate or oxidized humics (Blodau, 2002; Klüpfel et al., 2014). $CH_4$ is predominantly produced via two pathways: hydrogenotrophic and acetoclastic methanogenesis. During hydrogenotrophic methanogenesis $CO_2$ is reduced to $CH_4$, while during acetoclastic methanogenesis acetate is split in $CH_4$ and $CO_2$. These pathways differ with respect to their discrimination against the heavier $^{13}C$-isotopes due to the kinetic isotope effect (Hoefs, 1987). Differences in the isotopic composition are thereby commonly presented as $\delta^{13}C$ values, expressed as: $\delta^{13}C = (R_{sample}/R_{standard} - 1) \cdot 1000$ [‰], where R is the ratio of heavy isotope to light isotope of the samples and the respective standard. Acetoclastic methanogenesis results in $\delta^{13}C$-$CH_4$ values of –65 to –50 ‰ while hydrogenotrophic methanogenesis, discriminates stronger against the heavier carbon isotope and results in $\delta^{13}C$–$CH_4$ values of –110 to –60 ‰ and considerably $^{13}C$ enriched $CO_2$ compared to the acetocalstic pathway (Whiticar et al., 1986). Specific patterns have been observed in terms of spatial and temporal occurrence of the major $CH_4$ production pathways, with acetoclastic methanogensis typically increasing in contribution towards the surface or within the rhizosphere (Holmes et al., 2015). On the other hand, an assignment of methanogenic pathways based on $^{13}C$ signatures of $CH_4$ can be biased by microbial oxidation of $CH_4$. This can in particular be the case near and in the unsaturated profile where oxygen can enter by diffusion, or in the rhizosphere where plants deliver oxygen through aerenchyma roots (Chasar et al., 2000). Upon conversion of $CH_4$ into $CO_2$, the residual $CH_4$ gets enriched in $^{13}C$ compared to the source $CH_4$ (Teh et al., 2006), a process which yields similar $\delta^{13}C$-$CH_4$ signatures as observed upon $CH_4$ production by the acetoclastic pathway. $CH_4$ is released to the atmosphere by three different processes: i) through diffusion

through the acrotelm, which is a relatively slow process, ii) through ebullition, i.e. a fast evasion of methane bubbles, and iii) through fast molecular diffusion or pressurized throughflow convection through aerenchymatous tissue of vascular plants (Morris et al., 2011; Schütz et al., 1991; Whiting and Chanton, 1996; van den Berg et al., 2016; Hornibrook et al., 2009). Due to the slow diffusion of methane in peat, up to 100 % of diffusive $CH_4$ is oxidized in the acrotelm before it reaches the atmosphere, while the fast processes effectively bypass oxidation and thus contribute a major fraction to observed fluxes (Whalen et al., 1990; Whalen, 2005). Therefore, a change in vascular plant cover or changes in the peat structure due to altered litter inputs and stronger decomposition can be expected to affect methane emissions.

Carbon cycling and nitrogen (N) cycling in peatlands are coupled and eutrophication of peatlands is one major threat to these normally nutrient-limited ecosystems as demonstrated in several long-term fertilization experiments. For example, a decade of fertilizer applications to bogs in Canada (Mer Bleue), in the UK (Whim Bog), in Sweden (Degerö Stormyr) and seven years of high nitrogen deposition to a bog in the Italian Alps caused a loss of mosses and an increase in vascular plant biomass (Bubier et al., 2007; Wang et al., 2016; Sheppard et al., 2013; Wiedermann et al., 2007; Bragazza et al., 2012). In the Mer Beue bog shrubs benefit most from increased nutrient availability, whereas at Whim bog, Degerö Stormyr and at an Italian mire it remained unclear whether shrubs or graminoids benefit most. Sheppard et al. (2013) further observed differential effects on a peatland plant community when dry deposited ammonia-N and wet deposited reduced N, respectively, were applied. A number of studies supported that an increase of vascular plant cover can reduce the productivity of peat mosses and, in addition, can potentially promote the decomposition of organic matter by affecting the stoichiometry of soil enzymatic activity (Bragazza et al. 2013, Bragazza et al. 2015), ultimately leading to a decreasing ability of peatlands to sequester $CO_2$ from the atmosphere (Bubier et al., 2007), resulting in decomposition of peat (Rydin and Jeglum, 2013). Altered plant communities in peatlands were repeatedly shown to alter $CO_2$ and $CH_4$ fluxes: in fact, maximum net ecosystem exchange (NEE) was found to be reduced after long-term fertilization and a concomitantly promoted vascular plant community in the Mer Bleue bog (Bubier et al., 2007), and increased $CH_4$ emissions were observed at Degerö Stormyr from plots with an increased vascular plant coverage after a decade of excess nutrient supply (Eriksson et al., 2010). Indeed, selective removal of plant functional types (PFTs), although in these particular studies combined with warming and drought experiments, demonstrated a strong impact of vegetation changes on gas exchange (Larmola et al., 2013; Ward et al., 2013; Kuiper et al., 2014; Robroek et al., 2015). While such plot based manipulation experiments as reported above revealed clear patterns, there is still a gap of knowledge in terms of long-term consequences of excess supply of nutrients to a peatland and the resulting interactions and feedbacks between plants and peat, especially under in-situ conditions. There is only a poor understanding of the interplay of plant functional types, substrate quality, and anoxic-oxic conditions, and of how exchange of $CO_2$ and $CH_4$ at the soil/atmosphere-interface would eventually be affected.

To address this research gap, we investigated C cycling of the once oligotrophic Wylde Lake peatland, which since 1954 is exposed to infiltration of nutrients and strongly pronounced water level fluctuations as induced by the nearby water reservoir. The site was in detail described by Berger et al. (2017); a particular finding was that even after decades of excess nutrient supply (currently $5.9 \pm 0.1$ to $4.35 \pm 0.3$ g m$^{-2}$ y$^{-1}$ of N input was determined and several more nutrients were found to infiltrate

the peatland from its periphery), the peatland still featured high peat accumulation rates of ~200 to ~300 g C m$^{-2}$ y$^{-1}$, However, a strong gradient in vascular plant cover was apparent. As pointed out by Berger et al. (2017) lateral nutrient influx through repeated inundation events cannot be easily compared to sites subjected to deposition from the atmosphere; nevertheless, an apparently intact peatland system, i.e. an intact mire, despite such serious anthropogenic impacts is contradictive to findings from above mentioned studies; according to existing studies, already after one decade of N fertilization decomposition and peat degradation would be expected. Moreover, the particular scenario in our study here, the impacts of inundation on nearby ecosystems, is gaining increasing importance as there is a worldwide increase of impoundment area (Tranvik et al., 2009) and serious effects on peatland carbon cycling are likely (Ballantyne et al., 2014; Kim et al., 2015).

The objective of this study was therefore to extend the existing study on nutrient impact, vegetation, and net carbon accumulation, comparing effects on C cycling in more detail. To this end, we assessed current $CO_2$ and $CH_4$ exchange, peat quality and pore water chemistry along a transect, ranging from a shrub dominated site (200 m distance to the reservoir; greatest nutrient input), over graminoid-moss dominated sites (400 and 550 m distance to the reservoir; intermediate nutrient input) to a site composed of equal shares of few graminoids and shrubs above dominant mosses (800 m distance to the reservoir; smallest nutrient input) in the Wylde Lake peatland in Ontario, Canada. Moreover, to address changes in methanogenic pathways and to study predominant pathways of emission, we assessed seasonal variation in $\delta^{13}$C of $CH_4$ and $CO_2$ in peat profiles and in $CH_4$ surface fluxes.

We hypothesized that 1) hydrologically altered and nutrient enriched peripheral sites feature accelerated C cycling, reflected in more decomposed peat, 2) increased abundance of vascular plants can increase $CO_2$ uptake but also change patterns of $CH_4$ production and emission, in particular if graminoids dominate, and 3) long-term nutrient enrichment in combination with hydrologically altered conditions may therefore cause differential responses of carbon cycling and does not necessarily cause a loss of the C-sink function of peatland ecosystems.

## 2 Methods

### 2.1 Description of the study area and study sites

Wylde Lake peatland has been described in detail in Berger et al., (2017). In brief, it is located in southeastern Ontario, 80 km northwest of Toronto (43.920361° N, 80.407167° W) (Fig. 1), and it is part of the Luther Lake Wildlife Management Area. Climate is cool temperate, average July temperature is 19.1 °C, average January temperature is −8.0 °C and the mean annual temperature is about 6.7 °C. Annual precipitation amounts to 946 mm, with the major portion falling in summer (1981 to 2010, Fergus Shand Dam, National Climate Data and Information Archive, 2014). Peat formation started about 9000 years before present on calcareous limnic sediments and total peat depth today is about 5 m.

For flood control and water management, the "Luther Lake" reservoir, neighboring Wylde Lake peatland, had been created in 1954. Through flooding of the reservoir, Wylde Lake peatland has been exposed to altered hydrological conditions in a way that the water reservoir enhanced water level fluctuations in a large part of the site: in summer or under dry conditions, water

is released from the reservoir, thereby draining water out of the peatland; under wet conditions, water table levels of the reservoir increase and water is pushed into the peatland. Those sites in closer vicinity to the reservoir are presumably more affected than sites further away from the reservoir (Berger et al., 2017).

Four intensively investigated measurement sites (Fig. 1) were arranged along a transect stretching from nearby the shoreline of the Luther Lake reservoir about ~1 km south into the central treed bog area. Each site featured an individual mosaic of hummocks, hollows, and lawns, however, all measurements as considered in this study were taken in and all samples were collected from hollows.

Site 4 was located about 200 m away from the reservoir in an area overgrown by *Myrica gale* and where *Sphagnum* mosses were in retreat. Site 4 will hereinafter be referred to as "*shrub dominated site 4*". Site 3 and site 2 were in the open poor fen - bog transition area with site 2 being further away from "Luther Lake" reservoir (550 m) than site 3 (400 m). Site 2 and site 3 were dominated by *Sphagnum* mosses and graminoids with only few shrubs and will hereinafter be referred to as "*graminoid-moss dominated sites*". These sites featured a variety of arenchymatous graminoid species, such as *Eriophorum* spp. at the sites 3 and 2, and *Dulichium arundinaceum* at site 3. Site 1 (~800 m away from the reservoir) accommodated equal shares of few graminoids and shrubs above dominant *Sphagnum* mosses, and will be referred to as "*moss dominated site 1*". The four sites also differed with respect to their most abundant *Sphagnum* species, reflecting increasingly minerotrophic conditions towards the lake. While *S. capillifolium*, an ombrotrophic to slightly minerotrophic hummock species (Laine et al., 2011), was abundant at the sites 1, 2 and 3, its abundance was decreased at site 4. Moreover, site 1 featured the abundant *S. magellanicum* (another ombrotrophic to weakly minerotrophic hummock species (Laine et al., 2011)), site 2 featured the abundant *S. angustifolium* (tolerating ombrotrophic to minerotrophic conditions (Laine et al., 2011)) and site 3 featured the abundant *S. girgensohnii*, a minerotrophic hollow species (Laine et al., 2011). The two most abundant *Sphagnum* species at site 4 were *S. fuscum* (mostly on hummocks but also in hollows, an ombrotrophic species (Laine et al., 2011), with a great ability to recover from desiccation (Nijp et al., 2014)) and again the minerotrophic hollow species *S. girgensohhnii*. See Table 1 for a detailed overview of the vegetation at the sites and see Fig. S1 for photographs of the sites.

As presented in Berger et al. (2017) the study area is subject to nutrient infiltration most likely from the "Luther Lake" water reservoir as indicated by increasing concentrations of nitrogen, phosphorus, sulfur, potassium, calcium, magnesium, iron, copper, and zinc as well as other metals in peat mostly toward the peatland periphery. N input rates of $5.9 \pm 0.1$ g N m$^{-2}$ y$^{-1}$ were reported for site 4 and $4.35 \pm 0.3$ g N m$^{-2}$ y$^{-1}$ for site 1; moreover, C/P and N/P ratios of surface peat suggested fen typical P limitation, C/Ca and C/Mg ratios indicated Ca and Mg limitation, while C/K ratios indicated higher K availability as compared to bog typical values presented in Wang et al. (2015). The peatland periphery appeared to act as a buffer for nutrients in a way that site 4 received the highest loads of nutrients but also areas further away were to some extent affected. Nevertheless, surface peat accumulation rates of ~200 to ~300 g C m$^{-2}$ y$^{-1}$ at the four sites revealed great recent carbon sequestration.

The impact of anthropogenic activities, in particular the formation of the reservoir, is evident from peat quality found at the sites: a quite similar peat quality at depths accumulated before dam construction at Wylde Lake peatland was obvious in peats

of all sites. A clear difference before and after dam construction was the enrichment in nutrients in the upper depths and the concomitantly altered vegetation.

## 2.2 Determination of organic matter quality of peat and pore-water

Peat samples were taken in July 2014, in depths of 5, 10 and 20 cm below the living *Sphagnum* layer by manual cutting. Peat from 75 cm depth was taken with a Russian peat corer. Peat was filled in jars avoiding any headspace and closed air-tight to maintain anoxic conditions as far as possible during transport to the laboratory.

To collect in-situ pore-water, suction samplers (Macro Rhizons, Eijkelkamp, Giesbeck, The Netherlands; pore size ~ 0.2 μm) were inserted into the peat at 5, 10 and 20 cm depth. Sampling was done by applying vacuum and collecting water with syringes, covering syringes with aluminum foil and peat to avoid exposure to light. Pore-water from 75 cm depth was pumped from 75 cm deep piezometers that were emptied one day prior to sampling to ensure sampling of fresh pore-water. Samples from piezometers were filtered using Macro Rhizons in the laboratory to ensure similar treatment of pore-water of all depths. All samples were taken and analyzed as three replicates.

Prior to Fourier-transform infrared spectroscopic (FTIR) analysis, oven-dried (70 °C) bulk peat samples were ground with a ball mill. Pore-water samples were oven-dried (70 °C) until all water was evaporated; afterwards 2 mg of the remaining organic matter were scraped off the sample bottles and ground in a mortar with 200 mg of potassium bromide (KBr) and pressed to pellets for analysis. We recorded spectra on a FT-IR Spectrometer (Varian 660, Palo Alto, USA) over a scan range of 4000 to 650 cm$^{-1}$ with a resolution of 2 cm$^{-1}$ and 32 scans per sample. A KBr background was subtracted from the spectra and spectra were baseline corrected. We identified spectral peaks (average location +/- 30 cm$^{-1}$) and related them to functional moieties as described in Niemeyer et al. (1992). As absorbance values do not give quantitative information on absolute values of functional groups, we related peaks of around 1620 cm$^{-1}$ to 1610 cm$^{-1}$ (aromatic C=C compounds/aromatic moieties) to polysaccharide peaks at 1170 cm$^{-1}$ to 950 cm$^{-1}$ wavenumbers (Niemeyer et al., 1992). A relative increase in ratios thus indicates a relative decrease in the labile polysaccharide moieties and thus an increase in the degree of decomposition in regard of a residual enrichment of refractory aromatics (Broder et al., 2012).

Pore-water samples were analyzed by absorption spectroscopy in the ultra violet and visible range (UV-VIS-spectroscopy; Varian UV 1006 M005 spectrometer, Palo Alto, USA). We recorded UV-VIS spectra over a range of 200 to 800 nm with a resolution of 0.5 nm using a 1 cm quartz cuvette. Prior to measurement, a blank spectrum of ultrapure water was recorded and subtracted from each sample. We additionally recorded fluorescence properties of dissolved organic matter (DOM) on a fluorescence spectrometer (Varian Cary Eclipse, Palo Alto, USA) at a scan rate of 600 nm/min. Excitation wavelengths (ex) were 240 to 450 nm in 5 nm steps, emission wavelengths (em) 300 to 600 nm in 2 nm steps to obtain excitation-emission-matrices (EEMs). If necessary, UV-VIS absorption at 254 nm was adjusted to a range of 0.1 to 0.3 by dilution with ultrapure water in order to be able to correct for inner filter effects in fluorescence spectroscopy. Repeated blanks were run to ensure cleanliness of the cuvette. Raman spectra of a blank were recorded each day to check analytical drift and to normalize fluorescence to Raman units (Murphy et al., 2010).

To evaluate DOM quality, we calculated commonly used indices, such as specific ultraviolet absorbance $SUVA_{254}$ (as a proxy for aromaticity, Weishaar et al., 2003) and the E2:E3 ratio (the ratio of UV absorbance at 250 nm divided by absorbance at 365 nm) providing information about molecular weight of organic matter (Peuravouri and Pihlaja, 1997) from UV‑VIS data. From fluorescence data, we calculated a humification index HIX (Ohno, 2002). (see Table S3 for equations used).)

## 2.3 Measurements of environmental variables

Air temperature and photosynthetically active radiation (PAR) were recorded about 1 km south of site 1 in an open area by a HOBO U30 weather station (U30-NRC-SYS-B, Onset, Bourne, MA, USA) at a temporal resolution of 5 min. Water table depth below surface (wtd), water temperature ($T_{water}$) and air pressure were measured in 30-min intervals using one pressure transducer (Solinst Levelogger Edge) in a monitoring well at each site, corrected for barometric pressure (Barologger Gold at site 2; Solinst Ltd., Georgetown, Canada). On each day of closed chamber measurements, an extra PAR sensor (Smart Sensor, Onset; Part # S-LIA-M003) and an extra temperature sensor (Temperature Smart Sensor, Onset; Part # S-TMB-M0XX) recorded PAR and air temperature at a temporal resolution of 10 secs at the site were chamber measurements were being taken.

## 2.4 Determination of $CO_2$ and $CH_4$ fluxes

In the hollows of sites 1-4 a set of six collars for chamber measurements were established in April 2012. The collars - installed 0.1-0.15 m into the soil- were cylindrical, had a diameter of 0.4 m and a total height of 0.2 m. Through object based image analysis (OBIA) based on aerial imagery obtained from UAV flights, the spatial coverages of PFTs at each site were obtained (data summarized in Table 1). Accordingly, the locations for chamber measurement collars of our study sites were defined to proportionally reflect the distribution of PFTs.

Closed chamber measurements were performed following Burger et al., (2016). Measurements were taken every 10 to 30 days at each site from April 20th, 2014 through September 22nd, 2015. In total 19 to 23 daily courses per site could be accomplished. Cylindrical plexiglas chambers were used for the flux measurements: a transparent chamber to measure net ecosystem exchange (NEE), a chamber covered with reflective insolation foil for ecosystem respiration ($R_{eco}$). Chamber closure time was 180 secs.

Air was circulated between the chamber and a trace gas analyzer (Ultraportable Greenhouse Gas Analyzer 915-001, Los Gatos Research Inc., Mountain View, USA) through 2 mm inner diameter polyethylene tubing, recording trace gas concentrations of $CO_2$ and $CH_4$ at a temporal resolution of 1 sec. According to the manufacturer, the reproducibility of $CH_4$ and $CO_2$ is < 2 ppb and < 300 ppb, respectively. The analyzer was factory-calibrated immediately before the campaign. Stability of the calibration was checked repeatedly during summer of 2014. In January and July 2015, the analyzer was again re‑calibrated. If $CH_4$ concentrations increased sharply within the first 60 secs of the measurement due to $CH_4$ bubble release caused by the positioning of the chamber, the measurement was discarded and repeated.

During each measurement day, each collar was monitored several times with the transparent and dark chamber at different times (typically between 5 am and 8 pm) and different PAR levels (typically 5 to 2000 µmol m$^{-2}$ s$^{-1}$) throughout the day. Unfortunately, due to the remoteness of our study site, measurements at night were not possible.

Gas fluxes were determined by Eq. 1:

$$F_{chamber} = \frac{\Delta c}{\Delta t\, A} \cdot \frac{P\, V}{R\, T},$$

based on the changes of concentration over time inside the chamber, applying the ideal gas law with the ideal gas constant $R$,
and correcting for atmospheric pressure $P$ and temperature inside the chamber $T$. The chamber volume $V$ and basal area $A$ were calculated from the chamber's physical dimensions, taking into account each collar´s vegetation volume, determined in May, July and October 2014 as well as in April and August 2015 and extrapolated for the other campaigns. The concentration change over time was derived from the slope of a linear regression of concentration vs. time. The first 40 secs after chamber deployment were discarded to account for the analyzer's response time. If the slope was not significantly different from 0
(tested with an F-test, $\alpha = 0.05$), the flux was set to zero.

An empirical description of the measured NEE fluxes of each site was accomplished with the help of a hyperbolic light response model (Owen et al., 2007). The non-linear least squares fit of the data to the model was done according to Eq. 2:

$$NEE = \frac{\alpha \beta Q}{\alpha \beta} + y$$

where $NEE$ is in mol m$^{-2}$ s$^{-1}$, $\alpha$ is the initial slope of the light response curve (in mol $CO_2$ m$^{-2}$ s$^{-1}$ per mol photon m$^{-2}$ s$^{-1}$), $\beta$ is the maximum NEE in mol $CO_2$ m$^{-2}$ s$^{-1}$, $Q$ is the photosynthetic active radiation in mol photon m$^{-2}$ s$^{-1}$, and $\gamma$ is an estimate of the average $R_{eco}$. Integration of NEE over the course of one day gave net daily ecosystem production (NEP). Gross primary productivity (GPP) was retrieved by subtracting $R_{eco}$ from NEP. Average $CH_4$ fluxes of measurement days of each site were
obtained and lastly, cumulative emissions of $CO_2$ and $CH_4$ were calculated likewise according to Tilsner et al. (2003).

To determine isotopic signatures of $CH_4$ fluxes, we carried out additional chamber flux measurements once a month from May to September 2015 using a shrouded chamber and the Los Gatos UGGA. The chamber was closed until $CH_4$ concentrations reached > 10 ppm for analysis of isotopic composition, but not more than 30 min. Samples for isotopic analysis were extracted from the chamber with 60 ml syringes through a polyethylene tube with a three-way stopcock on one end and filled into 40 ml
crimp vials that had before been flushed with nitrogen ($N_2$) and sealed with rubber stoppers. To correct isotopic values of $CH_4$ for background isotopic signature in the chamber, we collected six air samples at each site on every sampling day. Analysis was carried out as outlined for dissolved gases (see below).

## 2.5 Sampling of gases and dissolved gases in the peat

Concentrations of $CH_4$ and dissolved inorganic carbon (DIC/$\Sigma CO_2$) along peat profiles were analyzed in 5, 15, 25, 35, 45 and 55 cm depth with three replicates at each site using diffusive equilibration samplers made of permeable silicone tubes (Kammann et al. 2001). Using three-way valves, samples were taken with 10 ml syringes every two to three weeks from June 2014, to September 2015. Samples were stored overnight at 5 °C and analyzed the next day.

To determine temporal dynamics of isotopic signatures of $CH_4$ and $CO_2$ in the peat, we installed a separate set of silicone tubes in 5, 15, 25 and 35 cm depth with three replicates each per site. Silicone tubes for isotope sampling had an inner diameter of 1 or 0.5 cm, corresponding to a volume of 20 or 5 ml. The samplers with a volume of 20 ml were installed in 5 cm depth and the smaller samplers below, as close to surface larger volumes of samples were necessary in order to obtain sufficiently high concentrations (2.5 < x < 2000 ppm) for isotope analysis.

Tightness of all samplers was confirmed prior to installation. Equilibrium of gases such as $N_2O$, $CH_4$, and $CO_2$ at the silicone membrane has been shown to adjust with hours to days and isotopic fractionation can be expected to be negligible (Nielsen et al., 1997; Panikov et al., 2007; Pack et al., 2015). All samplers were installed one month prior to the first sampling and samples were taken once a month from May 2015 to September 2015 with 10 and 60 ml syringes and filled in 10 respectively 40 ml crimp vials that were before flushed with $N_2$ and sealed with rubber stoppers. Silicone samplers were refilled with $N_2$ to avoid oxygen entering the system.

To obtain high resolution depth profiles of concentration and isotopic signatures of $CH_4$ and DIC, pore-water peepers of 60 cm length and a 1 cm resolution (Hesslein 1976) were inserted on three occasions in June, July and September 2015 and allowed to equilibrate for four weeks prior to sampling. As results of pore-water peepers generally confirmed the data of the silicone samplers, results are not presented here but described in the supporting information (see Fig. S6).

## 2.6 Analyses of $CO_2$ and $CH_4$ concentrations and $\delta^{13}C$-$CO_2$ and $\delta^{13}C$-$CH_4$-values

Gaseous $CO_2$ and $CH_4$ concentrations were analyzed with a gas chromatograph (SRI 8610 C, SRI Instruments, Torrance, US) equipped with a Flame Ionization Detector (FID) and a Methanizer. Samples from pore-water peepers were analyzed by measuring the headspace concentration in the vials.

Ratios of $\delta^{13}C$ of $CO_2$ and $CH_4$ were determined by Cavity Ringdown Spectroscopy (CRDS; Picarro G2201-*i*, Picarro Inc., Santa Clara, US), simultaneously determining $^{13}C$ isotopic composition of $CO_2$ and $CH_4$ with a precision of <0.16 ‰ for $\delta^{13}C$-$CO_2$ and <1.15 ‰ for $\delta^{13}C$-$CH_4$. The analyzer was calibrated before each measurement with two working standards of $CO_2$ (1000 ppm, -31.07 ‰) and $CH_4$ (1000 ppm, -42.48 ‰). Standard deviation for $\delta^{13}C$-$CO_2$ was below 2 ‰ and below 4 ‰ for $\delta^{13}C$-$CH_4$. Isotopic signatures were expressed in the $\delta$-notation in ‰ versus VPDB-Standard according to Eq. 3:

$$\delta^{13}C = (R_{sample}/R_{standard} - 1) \cdot 1000 \ [\text{‰}]$$

where $R_{Sample}$ is the $^{13}C/^{12}C$ ratio of the sample and $R_{Standard}$ is the $^{13}C/^{12}C$ ratio of the standard.

As the accuracy of $\delta^{13}C\text{-}CO_2$ values was affected by high $CH_4$ concentrations present in the samples, we established a correction formula to revise $\delta^{13}C\text{-}CO_2$ values. This formula was applied for molar concentration ratios of $CO_2:CH_4$ between 0.3 and 1.5. Samples with $CO_2:CH_4$ ratios < 0.3 could not be corrected and were discarded; samples with higher ratios did not

need correction. To cross-check values of $\delta^{13}C\text{-}CO_2$, two additional standards, carbonic acid from fermentation (-26.61 ‰), natural carbonic acid (-0.19 ‰) and a mixture of both (-15.16 ‰) were measured both with CRDS and an isotope-ratio mass spectrometer (EA/TC-IRMS Nu Horizon, Hekatech/Nu Instruments, Wrexham, UK). Additionally, $\delta^{13}C\text{-}CO_2$ values had to be corrected for a storage effect. As samples were stored for several weeks, $CO_2$ was lost from the vials and isotopic signatures increased by 0.056 ‰ per day. There was no such effect detectable for $CH_4$.

Dissolved concentration of $CO_2$ and $CH_4$ were recalculated from partial pressures inside the silicone samplers applying Henry's Law according to Eq. 4:

$$c = K_H * p$$

where $c$ is the concentration in µmol/L, $p$ is the pressure in atm and $K_H$ is the in-situ temperature corrected Henry-constant in mol L$^{-1}$ atm$^{-1}$ (Sander, 1999). $CO_2$ dissociation was considered using equilibrium constants from Stumm and Morgan (1996) to calculate the total amount of DIC.

DIC and $CH_4$ concentrations in samples from pore-water peepers were recalculated from gas concentrations in the headspace, applying the ideal gas law and temperature corrected Henry-constants for laboratory conditions.

To gain information about the dominant $CH_4$ production pathway, the isotope fractionation factor $\alpha_C$ (for 35 cm depth) was calculated according Eq. 5 after Whiticar et al. (1986):

$$\alpha_C = (\delta^{13}C\text{-}CO_2 + 1000) / (\delta^{13}C\text{-}CH_4 + 1000).$$

**2.7 Statistical analysis**

Statistics software R i386 version 3.1.0 was used to verify if observed differences in organic matter quality varied between depths and sites were statistically relevant for each indicator separately. Data was tested for normal distribution (Shapiro-Wilk-Test, $\alpha = 0.05$) and homogeneity of variance (Levene-Test, $\alpha = 0.05$). In case both requirements were met, we carried out a one-way ANOVA (Analysis of Variance) ($\alpha = 0.05$) with a post-hoc Tukey's Honest Significant Difference (HSD) test ($\alpha = 0.05$) to identify which depths or which sites differed significantly. If either normal distribution or homogeneity of variance

were not given, a Kruskal-Wallis test ($\alpha = 0.05$) with a multiple comparison test after Kruskal-Wallis ($\alpha = 0.05$) as post-hoc test was executed.

Using *RStudio* Version 0.99.902 as well as R i386 3.2.3 we examined whether there were significant differences in $\delta^{13}C$ values of $CO_2$ and $CH_4$, $CO_2$ and $CH_4$ concentrations and cumulative emissions between the sites. Means were compared with t-Tests

(if data was normally distributed) respectively Kruskal-Wallis and post hoc Wilcoxon-Mann-Whitney-Test (if data was not normally distributed). The confidence level for the statistical tests was α = 0.05. Normality was tested with Shapiro-Wilk-Test (α = 0.05) and homogeneity of variance was confirmed with Levene-Test (α = 0.05). Correlations between environmental variables and fluxes, concentrations and isotopic signatures were determined with Pearson's product-moment correlation for
normally distributed data or with Spearman's rank correlation if data was not normally distributed. With ANOVA (α = 0.05), the effect of categorical variables on $CH_4$ fluxes and $\delta^{13}C$ values was computed.

## 3 Results

### 3.1 Organic matter quality of peat and pore-water

The highest degree of bulk peat decomposition, as indicated by the highest 1618.5/1033.5 absorption ratios, was found at site 4 between 5 and 20 cm depth (p < 0.05 in 10 and 20 cm depth), indicating the highest degree of bulk peat decomposition at this site and these depths (Fig 2 (a)), whereas the 1618.5/1033.5 ratios of the sites 1-3 were not significantly different. Pore-water samples´ 1618.5/1033.5 ratios of site 3 were smallest between 5 and 20 cm depth as compared to all other sites (p < 0.05), indicating the lowest degree of decomposition of DOM here (Fig 2 (b)). Aromaticity as determined with *SUVA$_{254}$* (Fig
2 (c)) did not show significant differences between sites in pore-water samples (exception: site 1 and site 3 in 20 cm depth (p = 0.033), where site 1 SUVA$_{254}$ was significantly higher than site 3 SUVA$_{254}$). The degree of humification, as depicted by *HIX* (Fig 2 (d)), was significantly lowest in site 3 pore-water (5 cm site 3 and 4: p = 0.026; 10 cm site 1 and 3: p = 0.014; 20 cm site 3 and 4: p = 0.020). The slope ratio *E2:E3* (Fig 2 (e)), indicative of molecular size and aromaticity, was not significantly different at the sites 1-4.

**3.2 Development of wtd and T$_{water}$ during the study period**

During our study period, hollow wtd showed strong seasonal fluctuations; maximum wtd (i.e. highest water table levels) throughout the study period were reached during snowmelt in spring 2014 (site 1: 6.94 cm, site 2: 4.99 cm, site 3: 16.26 cm, site 4: 23.18 cm above hollow surface), minimum wtd (i.e. lowest water table levels) were reached during the summer of 2015 (site 1: 32.5 cm, site 2: 31.75 cm, site 3: 13.34, site 4: 19.11 cm below hollow surface). The sites 1 to 4 all showed similar
courses of wtd, however, at site 3 and site 4 water levels were generally higher as compared to site 1 and 2 (p < 0.05). The range between maximum and minimum wtd at all sites was overall similar (site 1: ~39.5 cm, site 2: ~36.7 cm, site 3: ~30 cm (logger failure when water levels were lowest), site 4: ~42.3 cm). T$_{water}$ varied between ~2 °C in winter and ~16 °C in summer. Detailed courses of wtd and T$_{water}$ are presented in the panels (a) and (b) of Fig. 3.

## 3.3 Fluxes of $CH_4$ and $CO_2$ at the soil/atmosphere interface, concentrations of $CH_4$ and DIC along soil profiles during the study period

Fluxes of $CH_4$ and $CO_2$ (Fig. 3, panels (c) - (f)) showed strong annual variability. $CH_4$ fluxes (Fig. 3 (c)) were positive (fluxes from soil to atmosphere) throughout the entire study period; greatest fluxes occurred during the growing season, minute fluxes
were detected during the dormant season. In general, site 3 $CH_4$ fluxes plotted above the other sites' fluxes, whereas site 4 fluxes plotted below the other sites' fluxes (except for August 16th, 2015 when a mean flux of $0.76 \pm 0.58$ g $CH_4$ $m^{-2}$ $d^{-1}$ was detected, exceeding the fluxes measured at all other sites. During the entire study period (April 2014 through September 2015), hollows of site 3 released significantly ($p < 0.001$) more methane ($61.4 \pm 32$ g $CH_4$ $m^{-2}$) than the sites 1 ($41.8 \pm 25.4$ g $CH_4$ $m^{-2}$), 2 ($44.6 \pm 13.7$ g $CH_4$ $m^{-2}$), and 4 ($46.1 \pm 35.2$ g $CH_4$ $m^{-2}$); see also Fig. S5. Annual $CH_4$ emissions from May 2014 to May
2015 were $22.18 \pm 8.96$ at site 1, $30.66 \pm 7.63$ at site 2, $39.86 \pm 16.81$ at site 3, and $12.53 \pm 11.38$ g $CH_4$ $m^{-2}$ at site 4; thus emissions at site 3 were significantly ($p<0.05$) higher than at site 4, but $CH_4$ emission at sites 3 and 4 did not differ significantly from sites 1 and 2 emissions.

Fluxes of $CO_2$ (Fig. 3 (d), (e), (f)) showed a strong seasonal variability, with highest photosynthetic uptake and highest ecosystem respiration in summer and lowest fluxes in the dormant season. Site 3 NEP plotted below all other sites' fluxes,
indicating most $CO_2$ net uptake, whereas site 4 NEP apparently plotted above all other sites' fluxes, indicating less net uptake of $CO_2$ if not a net emission of $CO_2$ (Fig. 3 (d)). Regarding $R_{eco}$ (Fig. 3 (e)), patterns were similar at all sites. Regarding GPP (Fig. 3 (f)), site 3 plotted below all other sites, indicating highest photosynthetic uptake here, whereas site 4 mostly plotted above the other sites, indicating smallest productivity. During the study period, the cumulative NEP of hollows of site 1 was $1552 \pm 652$ g $CO_2$ $m^{-2}$, site 2 accumulated $1637 \pm 184$ g $CO_2$ $m^{-2}$, site 3 accumulated $2260 \pm 480$ g $CO_2$ $m^{-2}$ and site 4
accumulated $1093 \pm 794$ g $CO_2$ $m^{-2}$ (see Fig. S4). Thus, between May 19th, 2014 and September 23rd, 2015 site 4 accumulated significantly less $CO_2$ as compared with the other three sites ($p < 0.001$), while there were no statistically significant differences in terms of $CO_2$ uptake for the sites 1, 2 and 3. Annual cumulative NEP from May 2014 to May 2015 was $-896 \pm 151$ g $CO_2$ $m^{-2}$ at site 1, $-1023 \pm 615$ g $CO_2$ $m^{-2}$ at site 2, $-1282 \pm 361$ g $CO_2$ $m^{-2}$ at site 3, while site 4 released $135 \pm 281$ g $CO_2$ $m^{-2}$. Annual cumulative NEP of the sites 1, 2 and 3 was significantly lower than the site 4 NEP ($p < 0.05$).
Interestingly, site 4 $CH_4$, NEP and GPP fluxes differed notably between the growing seasons of 2014 and 2015. This was particularly caused by two plots, which in 2015 dramatically increased productivity and $CH_4$ emissions as compared to the previous year (data not shown). Concentration of $CH_4$ along depth profiles (Fig. 4, top panels) of all sites varied strongly throughout the year: they generally increased during the growing season, reached maximum values in the winter season 2014/2015 and comparably decreased during snowmelt in spring. A similar pattern was observed for DIC concentrations along
depth profiles (Fig. 4, lower panels). Maximum DIC concentrations were observed below 20 cm depth in autumn 2014 and winter 2014/2015 and minimum concentrations were observed during snowmelt in March and April 2015. DIC concentrations at site 4 at all depths were overall lower and significantly decreased ($p < 0.05$) in comparison to all other sites from February 23rd through April 4th, 2015; moreover, site 4 DIC concentrations were significantly ($p < 0.05$) lower than site 3 DIC

concentrations on August 6th, 2014 and between April 19th through July 18th, 2015. Concentrations in the uppermost depths of both $CO_2$ and $CH_4$ were strongly affected by fluctuations of wtd, with strong decreases upon water table decline and vice versa (see table S4 for statistical results).

### 3.4 Temporal and spatial variability of $\delta^{13}C$-$CO_2$ and $\delta^{13}C$-$CH_4$ -values in peat pore-gas profiles during the growing season in 2015

Values of $\delta^{13}C$ of the sampled $CH_4$ in the peat ranged from -78.74 to -26.77 ‰, $\delta^{13}C$ signatures of $CO_2$ ranged from -25.81 to +4.03 ‰ (see Fig. 5). Highest $\delta^{13}C$-$CH_4$ and $CO_2$ values were measured at site 1 in 5 respectively 35 cm depth in September. Lowest $\delta^{13}C$-$CH_4$ and $CO_2$ values were detected at site 1 in 15 cm depth in June and at site 2 in 15 cm depth in August respectively.

Overall, $\delta^{13}C$-$CH_4$ values showed an increasing trend with time from June to August in all depths. Average signatures in 5 to 35 cm depth differed significantly between sampling dates at all sites except between August and September ($p < 0.05$). Concomitant to a decline in water table levels in August and September, $\delta^{13}C$-$CH_4$ signatures shifted to less negative values in the upper 5 cm at sites 1 to 3; this shift was most distinctive at site 1 and least distinctive at site 3. At site 4, such shift occurred at 15 cm depth.

For $\delta^{13}C$-$CO_2$ signatures, significant differences between some sampling dates were found at sites 1, 2 and 4 for average values in 5 to 35 cm depth. At sites 1 and 2, signatures in August and September were higher than in June and July, paralleling the trend in $\delta^{13}C$-$CH_4$. At site 3 and 4, such significant shifts could not be observed.

No significant differences between depths could be found for either $\delta^{13}C$-$CH_4$ or $\delta^{13}C$-$CO_2$ signatures at any site. Anyway, at sites 1 and 2, $\delta^{13}C$-$CH_4$ signatures apparently increased with depth in June and July, no trend was observable at sites 3 and 4. In August and September, $\delta^{13}C$-$CH_4$ signatures seemed to decrease with depth except for site 4. Values of $\delta^{13}C$ of $CO_2$ increased with depth except at site 1 in July and at site 2 in July and August.

Mean signatures of $\delta^{13}C$-$CH_4$ at site 4 (-57.81 ±7.03 ‰) differed significantly from those at the other sites (site 1: -61.48 ±10.71 ‰, site 2: -60.28 ±5.57 ‰, site 3: -62.30 ±5.54 ‰) for the whole sampling period ($p < 0.01$, $p < 0.05$, $p < 0.001$).

Signatures of $\delta^{13}C$-$CO_2$ at site 3 were significantly higher than at the other sites in July ($p < 0.05$, $p < 0.01$, $p < 0.01$). Overall, highest mean values were found at site 1 (-12.05 ± 8.23 ‰) whereas site 4 revealed lowest $\delta^{13}C$-$CO_2$ signatures (-15.85 ± 3.61 ‰).

Isotopic composition of $CH_4$ and $CO_2$ as determined from pore-water peepers confirmed results obtained from the silicone gas samplers. Data is presented in the Fig. S5 in the supplemental information.

Fractionation factors $\alpha_C$ to characterize methanogenic pathways (according to Whiticar et al. (1986)) were calculated for water saturated, presumably anoxic conditions at -35 cm depth only (Table 2), as frequent or prevailing unsaturated conditions above this depth would rather favor methanotrophy. Given that $\alpha_C$ values between 1.04 and 1.055 indicate the prevalence of the acetoclastic methane production pathway, whereas $\alpha_C$ values higher than 1.065 support a shift towards the hydrogenotrophic

pathway, the acetoclastic pathway was apparently favored in July and August at the sites 1 and 2, in August at site 3 and in July, August and September at site 4; indicationg that a shift towards a higher contribution of the hydrogenotrophic pathway was only observed in June and September at site 1, and in June at site 2.

### 3.5 $\delta^{13}C$ signatures of emitted $CH_4$ during summer 2015

Values of $\delta^{13}C$ of emitted $CH_4$ ranged from $-81.87 \pm 3.81$ and $-55.61 \pm 1.20$ ‰ (see Fig. 6, panel (a) to (d)). Thereby, $\delta^{13}C$-$CH_4$ signatures increased from July to August and slightly decreased again in September. This pattern was thus related to the course of the wtd. Significant differences were only found at sites 3 and 4 between July and August ($p < 0.01$, $p < 0.05$), however, from visual inspection of the panels (a) – (d) of Fig. 6 $\delta^{13}C$- $CH_4$ values seemed to increase between May and September at site 3, while they appeared to decrease at site 4, with very distinct values in August. There was no such pattern

observable at sites 1 and 2. Summing up $\delta^{13}C$- $CH_4$ signatures from all sites, isotopic signatures in July differed significantly from those in August and September ($p < 0.05$). In September, isotopic signatures of the $CH_4$ flux at site 2 differed significantly from those at the other sites ($p < 0.05$).

Comparing isotopic signatures of dissolved $CH_4$ in the peat and emitted $CH_4$, plant mediated transport was the dominant $CH_4$ emission pathway during the summer 2015 at all sites and all sampling dates according to Hornibrook (2009) (Fig. 6 (e)).

**4 Discussion**

As expected from our studied transect ranging from a strongly altered (site 4) to an only slighty altered site (site 1) in terms of nutrient supply, hydrological conditions, and coverage of PFTs, we observed pronounced differences in gas fluxes and peat quality. On the other hand, the dominant $CH_4$ emission pathway was plant-mediated transport at any site. We are aware that observed effects of anthropogenic impact are much more difficult to constrain in an in-situ study as ours, compared to well

defined ecosystem manipulation studies. Nevertheless, our results support a clear and an obvious interplay of processes, fluxes, and vegetation that can be related to the observed impacts of nutrient enrichment and altered hydrology, as discussed in the following paragraphs.

### 4.1 Long-term insights into carbon cycling at the sites

Long-term plant community changes were recently shown to affect peatland organic matter composition (Hodgkins et al.,

2014), while such an effect was not identified in a short-term study (Robroek et al., 2015). Along our transect of study sites, affected by consequences of a construction of a dam in 1954, we observed the highest degree of bulk peat decomposition in the upper peat layers of shrub dominated site 4, which was located in closest vicinity to the water reservoir and which was the most altered one among our four sites (Fig 2 (a)). Our initial hypothesis 1 that peripheral sites feature accelerated C cycling, reflected in more decomposed peat, could thus only partly be verified: the fact that we did not find a gradual decrease in terms

of degree of bulk peat decomposition with increasing distance from the reservoir, but observed significant differences only for site 4, suggests that the observed differences could also be primarily induced by the shift to a predominance of shrubs. Shrubs

contain more woody parts and thus have higher lignin contents and more phenolic groups than graminoids or mosses and they are also more productive than mosses and graminoids (Bragazza et al., 2007). In recent studies, an increasing ericaceous shrub cover was associated with increasing polyphenol content in plant litter and pore-water, as well as increasing phenol oxidase in litter of ericaceous shrubs. Also, a higher release of labile C from vascular plant roots was observed. These changes, however

along an altitudinal gradient, were accompanied by a decreasing *Sphagnum* productivity (Bragazza et al., 2013; Bragazza et al., 2015). Even though at site 4 we primarily dealt with eutrophication, rather than warming, there might be similar processes explaining our observations: Shrubs outcompete *Sphagnum* mosses after long term-nutrient infiltration and a reduced recalcitrance of the peat arising from shrub litter may result in a reduced C storage, i.e. peat accumulation (Turetsky et al., 2012; Larmola et al., 2013; Ward et al., 2013). This was further suggested by the lowest $CO_2$ uptake and lowest observed DIC

concentrations along peat pore-gas profiles throughout the study period at that particular site.

At the graminoid-moss dominated site 3, pore-water DOM quality indices revealed a significantly lower share of aromatic compounds, and thus suggested a lower degree of humification and comparably increased molecular weight at that site (Fig 2 (c)-(e)). This more labile nature of dissolved organic matter compared to otherwise similar bulk peat quality suggested either an input from the vegetation (Robroek et al., 2015) or some inflow of water and solutes from the nearby reservoir. Given that

the site 4 pore-water DOM characteristics differed strongly from those at site 3, as did predominant PFTs, the distinctive features of the site 3 pore-water were probably also induced by the vegetation. However, the fact that the vegetational composition of site 3 and site 2 were rather similar, whereas the pore-water DOM quality was again significantly different, suggested that DOM properties were likely affect by vegetation, i.e. photosynthetic productivity and concomitantly higher input of labile compounds, as well as inflow of DOM from the reservoir.

The nature of our results does of course not allow for an unambiguous conclusion in terms of whether it is the vicinity to the reservoir or the plant community composition, which drives carbon cycling and peat accumulation at the sites. However, since peatland plant community compositions are known to be remarkably stable over time but experience changes in relative abundances (Rydin and Barber, 2001, Bragazza et al., 2006), we suggest that it was probably the vicinity to the reservoir that shaped the plant community composition at the sites over time, whereas the plant community actually drives carbon cycling.

**4.2 Seasonal development of carbon fluxes**

Different PFTs were recently shown to have a strong impact on peatland ecosystem $CO_2$ fluxes (Ward et al., 2013; Kuiper et al., 2014). This could be confirmed by our results: Our shrub dominated, strongly altered site 4 showed the lowest cumulative $CO_2$ uptake, whereas, at our graminoid-moss dominated sites 3 and 2, and at the least altered moss dominated site 1, very high $CO_2$ uptake rates were observed. The $CO_2$ uptake rates of our sites 1, 2 and 3 exceeded reported $CO_2$ uptake rates of bogs by

far: for instance, Teklemariam et al. (2010) reported that net ecosystem exchange of the ombrotrophic, continental Mer Bleue bog ranged from -140 to -20 g C m$^{-2}$while hollow $CO_2$ uptake rates of our study were notably higher, and were rather comparable to uptake rates reported for fens (Lund et al., 2010). These latter values also compare well with the surface peat accumulation rates of ~200 to ~300 g C m$^{-2}$ observed at our site (Berger et al. (2017)). Net $CO_2$ exchange of shrub dominated

site 4 was significantly lower compared to the other sites' fluxes, however, a strong inter-annual variability was apparent. In the light of the strong alterations in terms of vegetation cover and the most decomposed surface peat at shrub dominated site 4, our findings from an in-situ transect support earlier findings of a reduced net $CO_2$ exchange and a concomitantly promoted vascular plant community in a controlled long-term fertilization experiment at the Mer Bleue bog (Bubier et al., 2007).

Partitioning of NEP into $R_{eco}$ and GPP further illustrated the observed differences in $CO_2$ fluxes between sites. While $R_{eco}$ of all sites was in a comparable range, differences were predominantly driven by GPP. At shrub dominated site 4 during the growing season of 2014 photosynthetic uptake was clearly decreased (maximum GPP of only $-18.41 \pm 1.54$ g $CO_2$ m$^{-2}$ d$^{-1}$) as compared to the other sites but during the growing season of 2015, maximum GPP increased up to $-26.34 \pm 5.1$ g $CO_2$ m$^{-2}$ d$^{-1}$, which was well comparable to other sites' GPP, indicating a strong inter-annual variability in terms of photosynthetic activity
at our shrub dominated site 4.

    With regard to $CH_4$ emissions, the graminoid-moss dominated site 3 exceeded the other three sites by on average 30 %. In existing studies, greatest emissions were similarly found in wetter habitats dominated by graminoids (Levy et al., 2012, Gray et al. 2013). Given that $CH_4$ emissions of site 2 were significantly smaller than those from site 3, even though the two sites feature a very similar graminoid-moss dominated vegetation cover, the differences in $CH_4$ fluxes could either be attributed to
i) the wetter conditions at site 3 or ii) a greater nutrient supply to site 3, stimulating greater $CH_4$ production and emissions as observed earlier (Eriksson et al., 2010) or iii) a mixture of both effects. Interestingly, shrub dominated site 4, which experienced similar water table fluctuations like site 3, but featured a notably different vegetation cover, emitted $CH_4$ in a similar range like the graminoid-moss dominated site 2 and moss dominated site 1. High methane production due to input of labile organic matter nearby the reservoir was probably outweighed by lower methane transport and therefore emission due to a lower
graminoid cover.

    In our study NEP, GPP and $CH_4$ emissions were negatively correlated with $CH_4$ and DIC concentrations in the uppermost 50 cm of the profiles at the moss-dominated site 1 and at the graminoid-moss dominated sites 2 and 3. Such a decoupling of $CO_2$ and $CH_4$ fluxes from pools in the peat was already observed in previous studies: Graminoids are known to be important facilitators of $CH_4$ emissions because they can transport $CH_4$ from deeper, water-saturated layers of the peat into the
atmosphere via aerenchymatous tissue and bypass the zone of $CH_4$ oxidation (Shannon and White, 1994; Marushchak et al., 2016). Moreover, they supply exudates via their roots, stimulating microbial activity and accordingly methanogenesis (Bubier et al., 1995). Through their deeper rooting system, graminoids may thus have connected the $CH_4$ pools of deeper layers below our studied profile, i.e. below 50 cm depth, to fuel the observed surface fluxes. The decreasing concentrations of $CH_4$ in near surface layers due to a decrease in water table levels and partial aeration did thus not translate in lower fluxes, a similar effect
as suggested by Strack et al. (2006). Also, at low water tables and unsaturated conditions higher diffusivity for $CO_2$ can occur, leading to notably higher diffusive fluxes despite low concentrations (Knorr et al., 2008). It is striking that DIC concentrations at the shrub dominated site 4 were notably lower as compared with other sites. A reasonable explanation is a lower peat quality resulting from repeated peat oxygenation upon water table fluctuations of the reservoir, stimulating microbial decomposition in the presence of deciduous shrubs (Bragazza et al., 2016), which are apparently promoted in closer vicinity to the eutrophic

water reservoir. Such effect of aeration might appear contradictory, as wetter conditions would be expected near the water reservoir. However, repeated water table fluctuations driven by management of the reservoir could effectively recharge electron acceptor pools to support ongoing decomposition, as e.g. shown in water table manipulation experiments (Blodau et al., 2004; Knorr et al., 2009). Moreover, near the reservoir, also an advective redistribution and removal of $CO_2$ and $CH_4$

through advective flow cannot be excluded.

## 4.3 Methane production, methanotrophy and pathways of $CH_4$ emissions as inferred from stable isotopes

Distinguishing $CH_4$ production pathways in peatlands using $\delta^{13}C$-signatures along depth profiles is a common approach (e.g. Holmes et al., 2015; McCalley et al., 2014; Hodgkins et al., 2014; Kotsyurbenko et al., 2004; Chasar et al., 2000). However, methanogenesis is a strictly anaerobic process and thus saturated, anoxic conditions are a prerequisite for an unbiased

differentiation of pathways using $^{13}C$ only (Conrad, 1996). Methanotrophy would otherwise bias the interpretation of $^{13}C$ isotopic signatures of methane, as residual methane gets enriched in $^{13}C$, mimicking values as observed under methanogenic conditions predominated by the acetoclastic pathway (Whiticar, 1999). Indeed, summer water table levels at all sites at Wylde Lake peatland dropped down to 32.5 cm (site 1), 31.75 cm (site 2), 13.34 (site 3) and 19.11 cm (site 4) below surface and we could thus only assume saturated, anoxic conditions below that depth. Therefore, we will limit the discussion of $CH_4$ production

pathways to depths below -35 cm. For shallower depths (-5 to -25 cm), effects of under such conditions much more favorable methanotrophic activity can be expected to predominate: If the proportion of methanogenesis vs. methanotrophy is comparatively shifted toward methanogenesis, a relative $^{13}C$-$CH_4$ depletion would be detected, and if the proportion of methanogenesis vs. methanotrophy is comparatively shifted toward methanotrophy, a relatively $^{13}C$ enrichment in $CH_4$ would be detected). Methane oxidation is known to cause $^{13}C$-enriched $CH_4$ that is even more $^{13}C$ enriched compared to the ambient

$CH_4$, as was observed in the top -5 to -15 cm along our study transect during the summer months and $\alpha_C$ values typically observed for the acetoclastic pathway would in this case clearly arise from methanotrophic activity (Alstad and Whiticar, 2011). Our least negative values (-26.77 ‰), observed at 5 cm depth of moss-dominated site 1, markedly exceeded those found in other studies (e.g. Bellisario et al., 1999; Corbett et al., 2013). This is not surprising, though, as the latter studies had been conducted in peats under anoxic conditions, in which oxidation of $CH_4$ did not play an important role. Moreover, $\delta^{13}C$-$CH_4$

signatures at 5 cm depth of different sampling dates appeared to be most variable at the sites 1 and 2, which were also found to be drier than the sites 3 and 4, where less pronounced shifts of $\delta^{13}C$-$CH_4$ signatures occurred throughout the sampling period. However, also at the latter sites, variations in $\delta^{13}C$-$CH_4$ were apparently driven by fluctuations of the water table levels. Isotopic composition of $CO_2$ and $CH_4$ thus indicated clear differences between sites and depths (see Fig. 5), corresponding to observed patterns in concentrations and inferred production, consumption, or emissions. In general, $\delta^{13}C$-$CH_4$ signatures along soil

profiles of all sites increased between June and August, i.e. $CH_4$ got more enriched in $^{13}C$, suggesting that $CH_4$ oxidation must have been an important factor throughout the dry season in summer when the water table notably dropped. Another interesting finding was the strong $\delta^{13}C$-$CH_4$ signal pointing to notable $CH_4$ oxidation at 15 cm depth of shrub dominated site 4 in August 2015 (-39.10 ‰) as compared to more $^{13}C$ depleted $CH_4$ (-57.73 ‰) in 5 cm depth. This was probably due to the particularly

low $CH_4$ concentrations and more negative $\delta^{13}C$-$CH_4$, suggesting an input of atmospheric methane into the surface peat. The shrub dominated site 4 also featured the most enriched $\delta^{13}C$-$CH_4$ signatures in general, suggesting either least $CH_4$ production or most $CH_4$ oxidation here. The graminoid-moss dominated site 3, showed the smallest variations in $\delta^{13}C$-$CH_4$ signatures throughout the sampling period, suggesting least modification of $\delta^{13}C$-$CH_4$ from oxidation here, which corresponds well with

greatest $CH_4$ emissions measured at that site.

Our $\delta^{13}C$-$CO_2$ values ranging from -25.81 to + 4.03 ‰ also agreed well with other studies (e.g. Landsdown et al., 1992; Hornibrook et al., 2000), although comparably less data on $^{13}C$ in $CO_2$ from other studies is available. Overall lowest values were found at shrub dominated site 4, where least negative values of $\delta^{13}C$-$CH_4$ were observed, suggesting a higher share of $CO_2$ from increased $CH_4$ oxidation. Our $\delta^{13}C$-$CO_2$ values generally got more enriched in $^{13}C$ with depth at all sites and sampling

dates, as expected from ongoing fractionation by methanogenesis. Great shifts in $\delta^{13}C$-$CO_2$ values of the drier sites 1 and 2 during the entire sampling period could again be explained by increased exchange of peat derived $CO_2$ with atmospheric $CO_2$ under unsaturated conditions with dropping water tables in August.

Regarding observed ranges of $\alpha_C$ values at -35 cm depth at the sites, also a gradient in terms of methane production along the transect of study sites became apparent: moss dominated site 1 and graminoid-moss dominated site 2, which experienced the

lowest water tables during the summer, and which were located in farthest distance from the water reservoir, feature a distinct shift from mostly hydrogenotrophic $CH_4$ production in June to acetoclastic $CH_4$ production in July and August and another shift back to hydrogenotrophic $CH_4$ production in September, with these shifts being more pronounced at site 1. This could be related to increased vascular plant activity in the growing season and concomitant substrate supply to methanogens, e.g. though exudation; an increased share of acetoclastic methanogenesis within the rhizosphere has previously been reported (Chasar et

al., 2000; Hornibrooket al., 1997). At graminoid-moss dominated site 3 and at shrub dominated site 4, such obvious shifts of methane production pathways could not be observed, though; $\alpha_C$ values indicated either acetoclastic $CH_4$ production or a co-occurrence of acetoclastic and hydrogenotrophic $CH_4$ production. As acetoclastic methanogenesis is in particular supported in minerotrophic peatlands in presence of vascular plants (Alstad and Whiticar, 2011; Chasar et al., 2000), predominance of that pathway -in particular in closer vicinity to the reservoir- is not a surprising finding for Wylde Lake peatland. Indeed, under

predominance of sedges, which supply labile organic matter through roots, aceotclastic $CH_4$ production prevailed (Bellisario et al., 1999; Popp et al., 1999; Strom et al., 2003). However, the fact that $CH_4$ production pathways at site 3 and 4, featuring a very different vegetation, were similar, whereas $CH_4$ production pathways at the site 2 and 3 were different, even though the sites featured a similar vegetation, suggested that variation of $\alpha_C$ would rather reflect the impact of the reservoir, either a) by sustaining higher water tables, or b) by increased nutrient input, than the presence of sedges at the sites.

Regarding isotopic signatures of the emitted $CH_4$, in general, the emitted $CH_4$ (see Fig. 6 (a) – (d)) was lighter than the $CH_4$ in the pore-gas (see Fig. 5) of all sampled peat layers. This suggests that the emitted $CH_4$ must have been produced in the deeper peat layers (Marushchak et al., 2016), where $\delta^{13}C$-$CH_4$ signatures were probably more depleted and during transport through plant aerenchyma, the lighter $CH_4$ could bypass oxidation. Moreover, plant-mediated transport also discriminates against the $^{13}C$-$CH_4$ (Chanton, 2005). Interestingly, plant mediated transport was the dominant $CH_4$ emission pathway even at the shrub

dominated site 4 and moss dominated site 1, where graminoid cover accounted only for about 10 %. We suggest that this is due to the great $CH_4$ oxidation in the upper peat layers and rather high concentrations at greater depth, facilitating plant mediated transport and ebullition. From visual inspection of the panels (a) – (d) of Fig. 6 and Fig. 5 we suggest that the emitted $CH_4$ originated from at least -35 cm depth or below.

Regarding hypothesis 2 stating that increased abundance of vascular plants can increase $CO_2$ uptake but also change patterns of $CH_4$ production and emission, in particular if graminoids dominate, this hypothesis can only partly be accepted. If increased vascular plant cover translated into increased $CO_2$ uptake, we should have observed increasing uptake in the order of site 1 < 2 = 3 < 4, but in fact we observed only significantly decreased uptake at site 4. The $CO_2$ uptake at moss dominated site 1 (the site with the least coverage of vascular plants) was not statistically different from the cumulative NEP observed at the
graminoid-moss dominated sites 2 and 3. Moreover, we cannot directly state that $CH_4$ production and emission was increased where graminoids dominated. Although greatest $CH_4$ emission was observed at graminoid-moss dominated site 3, cumulative $CH_4$ emission at site 2 was significantly lower, despite relatively similar vegetation. Moreover, besides by PFTs, $CH_4$ production appeared to be affected by the vicinity of the water reservoir, whereas plant-dominated $CH_4$ emission was the dominant $CH_4$ emission pathway at all study sites, even where graminoid coverage accounted for only 10 %. So, we conclude
that an interplay of nutrient input, water table depth and composition of vegetation shaped $CO_2$ uptake, $CH_4$ production and emissions and there is likely no unique driver in our in-situ study, compared to well defined manipulation experiments.

## 5 Concluding remarks

Our study and earlier work at this particular site confirm that despite long-term increased nutrient supply, peatland ecosystem functioning in terms of C sequestration was largely maintained. However, along a sequence of study sites it became apparent
that the affected sites responded differently to the altered conditions after dam construction in 1954. A shrub dominated site, which was in closest vicinity to the reservoir and accordingly faced greatest nutrient input and most pronounced water level fluctuations, indeed showed indications of degradation, such as most decomposed bulk peat, least atmospheric $CO_2$ uptake, and reduced coverage of *Sphagnum* mosses. However, even here, overall net $CO_2$ uptake still exceeded net $CO_2$ release. Two graminoid-moss dominated sites and a moss dominated site featured very high $CO_2$ uptake rates despite apparent impact of
nutrients and altered hydrology. Therefore, as hypothesized, our case study supports that long-term nutrient enrichment in combination with hydrologically altered conditions may cause differential responses of C cycling and does not necessarily cause a loss of the C-sink function of peatland ecosystems.

Moreover, methanogenesis and methanotrophy featured a pattern which appeared to be related not predominantly to vegetation, but primarily to the vicinity to the reservoir and thus nutritional status and hydrologic regime. On the other hand,
plant-mediated transport was determined to be the dominant methane emission pathway along our transect at all sites, even if graminoid cover was only 10 %. All surface peat layers indicated high methanotrophic activity, mitigating $CH_4$ emission through diffusion.

Lastly, our results suggest that a graminoid-moss dominated peatland site can obviously withstand eutrophication in combination with frequent inundation better than a shrub dominated peatland site, when regarding the overall carbon budget. Straightforward results from manipulation experiments of individual factors (e.g. fertilization or water table changes) may therefore not be easily transferred to complex in-situ conditions. We suggest that there could be a tipping point, when a peatland

system shifts from still being a net C sink -even though experiencing eutrophic conditions- to decreasing productivity, which might be related to an expansion of shrub dominated vegetation, decreasing overall carbon uptake.

## 6 Data availability

The data can be accessed by email request to the corresponding authors.

*Author contributions*. Christian Blodau, Klaus-Holger Knorr and Sina Berger designed the experiments, Sina Berger, Leandra Praetzel and Marie Goebel conducted field work and analyses with help of Klaus-Holger Knorr. Sina Berger prepared the manuscript with contributions from Klaus-Holger Knorr, Leandra Praetzel and Marie Goebel.

The authors declare that they have no conflict of interest.

*Acknowledgements*. This study was funded by the Deutsche Forschungsgemeinschaft (German Research Foundation - DFG) (BL563/21-1). We thank Martin Neumann from the Grand River Conservation Authority for the permission to carry out research in the Luther Marsh Wildlife Management Area and we thank Claudia Wagner-Riddle, Peter Smith and Linda Wing

for their help on organizational issues. We also thank Inge-Beatrice Biro, Magdalena Burger, Ines Spangenberg, Niclas Kolbe, Eike Esders, Michael Rammo, Nils Vickus, Fabian Benninghoff, Leonie Fröhlich, Jörg Rostek and Cornelia Mesmer for their support in the field. Analyses of $CO_2$ and $CH_4$ concentrations, $\delta^{13}C$ abundance and spectral analyses of various samples were carried out at the institutional lab or the Institute of Landscape Ecology, University of Münster. We thank Stefanie Holm, Ronya Wallis and Madelaine Supper for assistance during analysis of numerous samples in the laboratory.

This paper is dedicated to the memory of Christian Blodau, who led the Wylde Lake peatland project until he tragically passed away in July 2016.
We thank two anonymous reviewers for their thoughtful comments that helped to improve this manuscript.

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

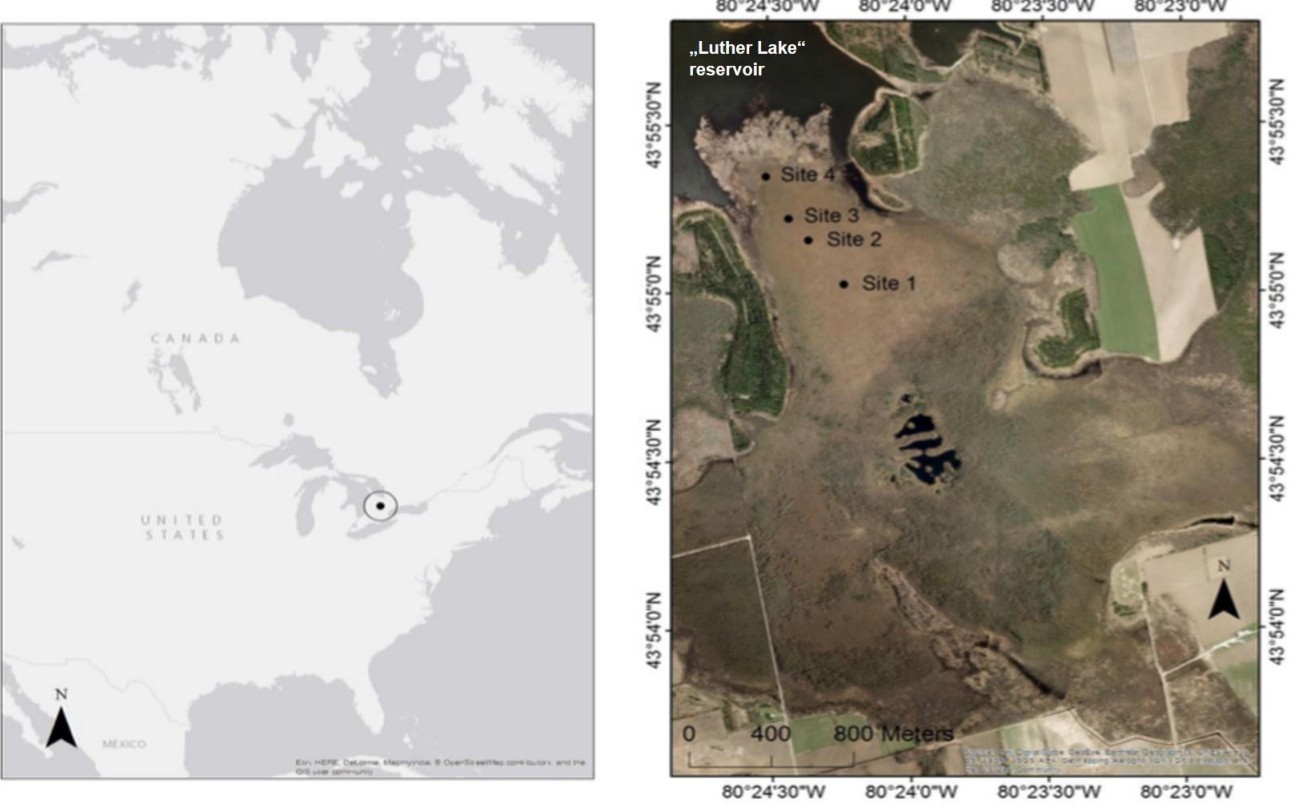

**Figure 1: Location of Wylde Lake peatland complex in North America (left), and sampling sites (black dots) within Wylde Lake peatland complex (right). Source: ArcGIS.**

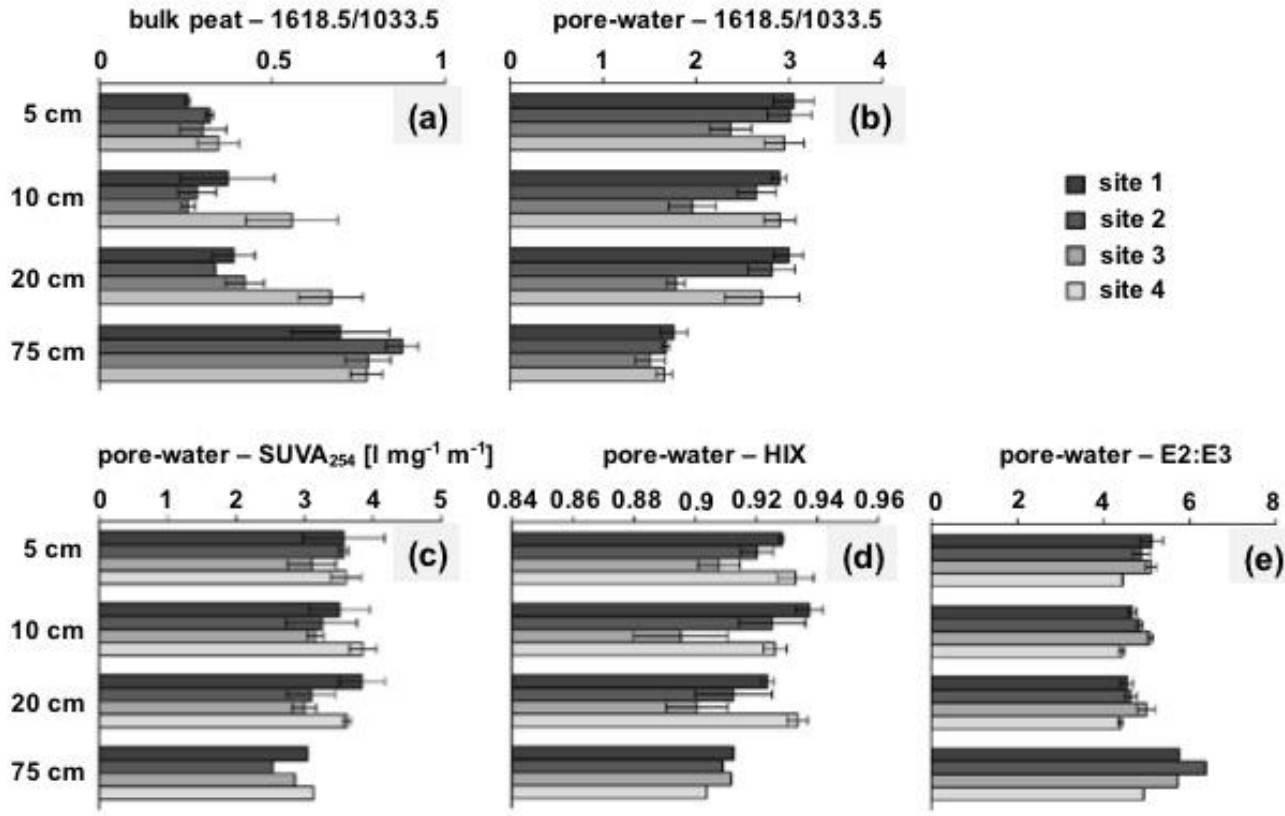

**Figure 2: FTIR ratios 1618.5/1033.5 in bulk peat (a) and pore-water (b) as well as SUVA$_{254}$, indicating aromaticity, (c), HIX, humification index, (d) and E2:E3, indicative of molecular size and aromaticity, (e) for pore-water samples of the sites 1 to 4. n = 3. Error bars indicate +/- standard deviation.**

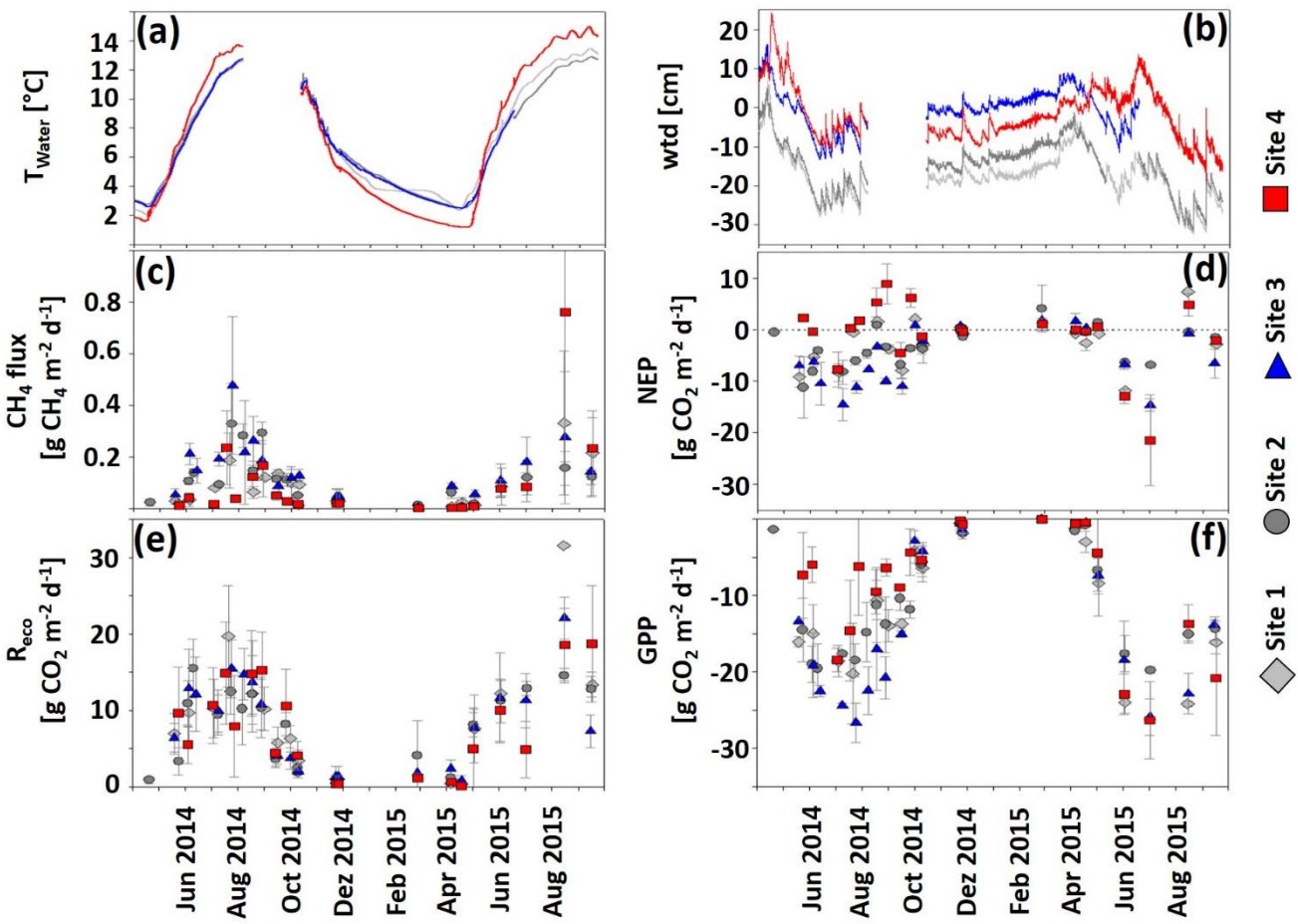

**Figure 3:** Development of (a) $T_{water}$ [°C], (b) wtd [cm], (c) $CH_4$ fluxes [g $CH_4$ m$^{-2}$ d$^{-1}$] and (d) – (f) $CO_2$ fluxes (NEP partitioned into $R_{eco}$ and GPP) [g $CO_2$ m$^{-2}$ d$^{-1}$], ± 1 SD (n=6) in hollows of the sites 1–4 from April 1st, 2014 through September 22nd, 2015. Negative $CO_2$ and $CH_4$ fluxes indicate uptake, positive fluxes indicate a release to the atmosphere. The dashed grey line in the NEP graph indicates a 0-flux.

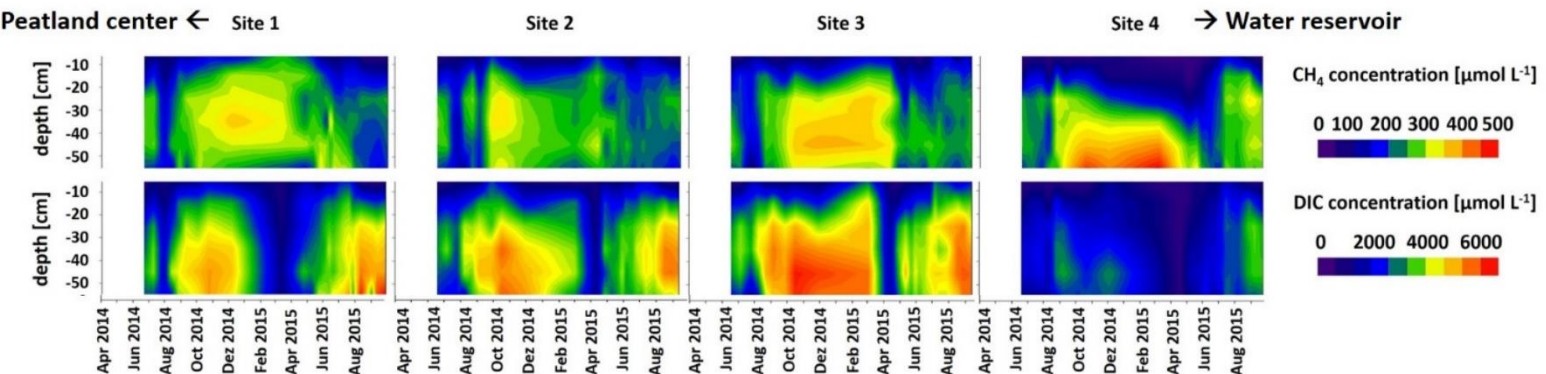

**Figure 4: Development of mean CH₄ and mean DIC concentrations [μmol L⁻¹], in hollows of the sites 1–4 from April 1st, 2014 through September 22nd, 2015. Concentrations were interpolated based on biweekly sampling at depths of 5, 15, 25, 35 45, and 55 cm.**

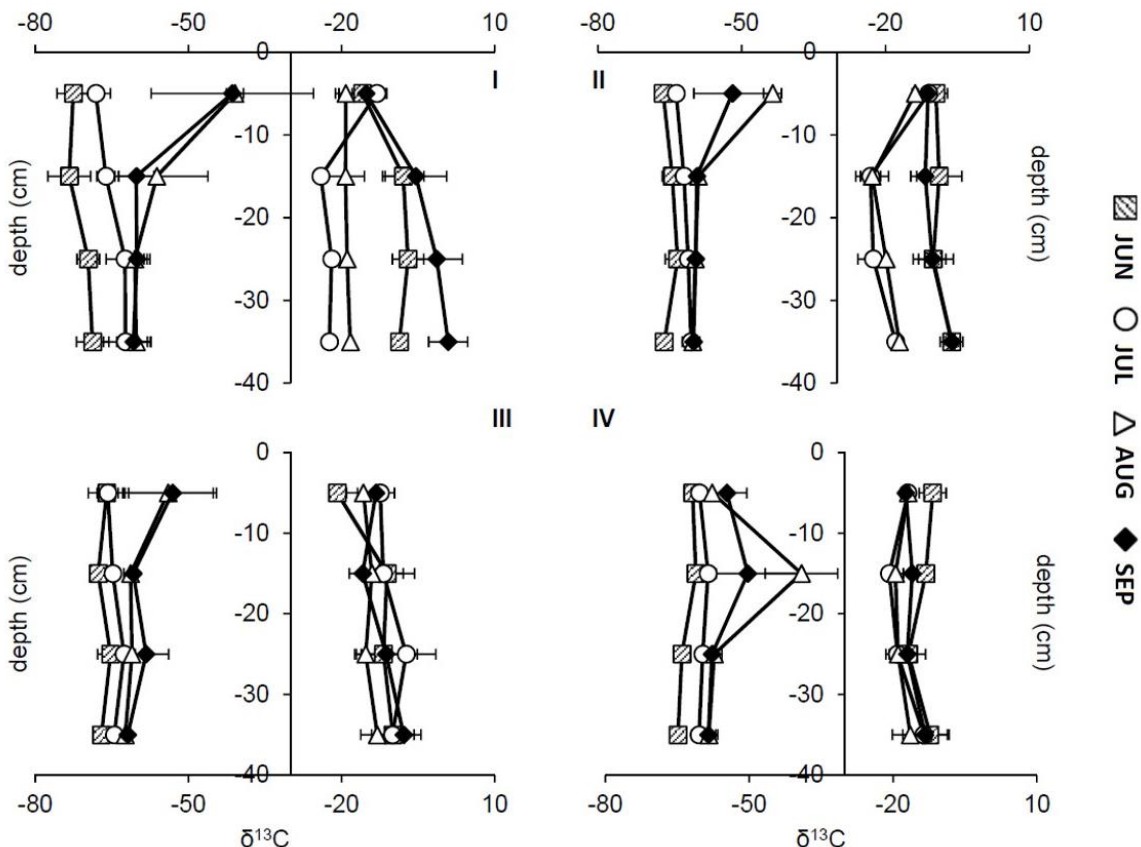

**Figure 5: Profiles of δ¹³C-CH₄ (left) and δ¹³C-CO₂ (right) signatures at sites 1-4 in the peat in 5-35 cm depth at different points in time. Squares = June (06/11), circles = July (07/08), triangles = August (08/27), diamonds = September (09/17). Graphs show mean values and standard deviations from three replications at each site. n = 1-3.**

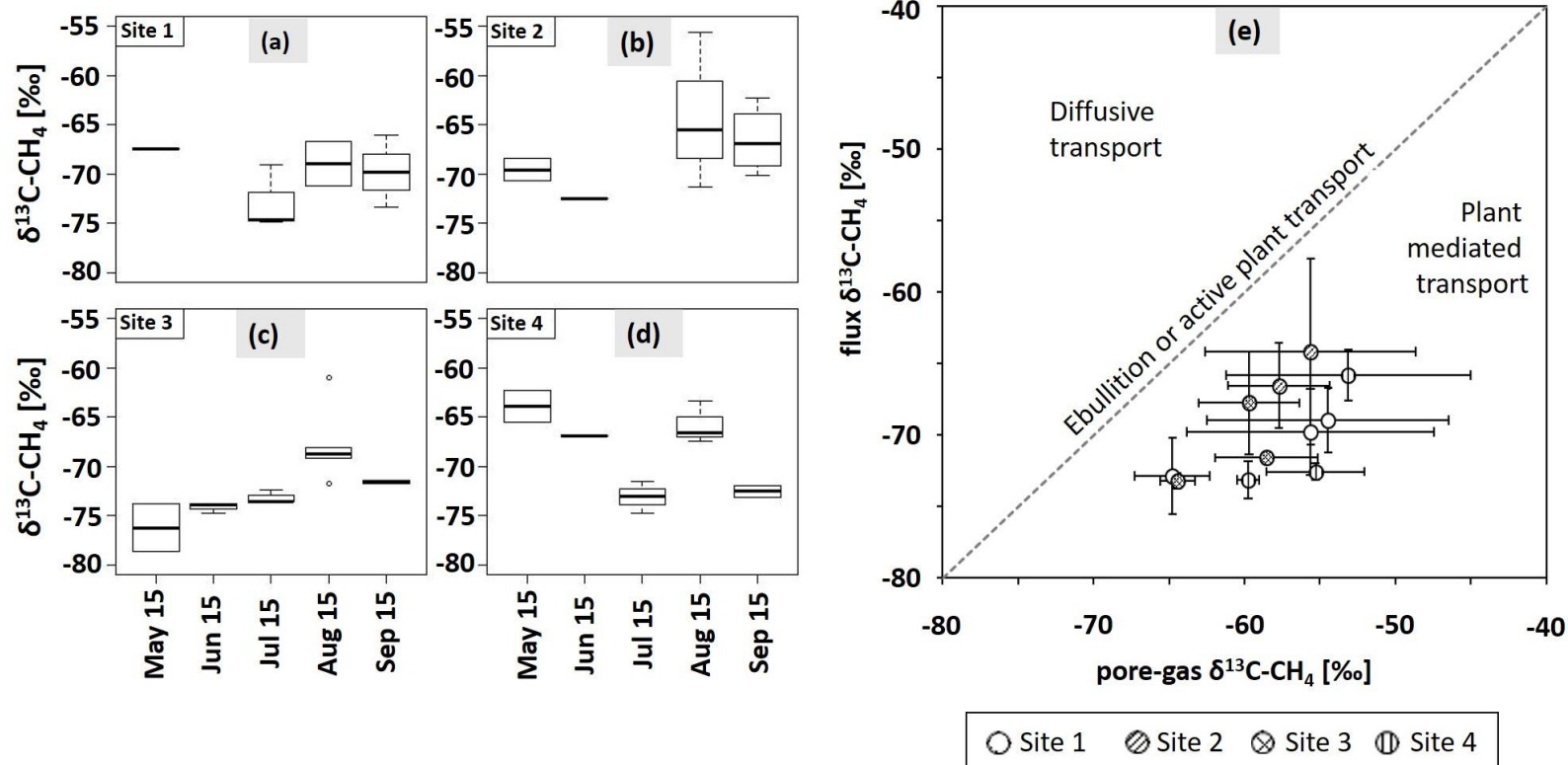

**Figure 6: δ¹³C-CH₄ signatures (‰) of CH₄ fluxes from May to September 2015 for the sites 1 (a), 2 (b), 3 (c) and 4 (d). n = 1-4. In July 2015, sampling at site 2 was not possible. Bold lines are the median, boxes show the 25 and 75 percentile, whiskers indicate minima and maxima within 1.5 times the interquartile range. Single points show outlier. (e): dominant flux pathway of CH₄ according to (Hornibrook, 2009). Empty circles = site 1, circles with diagonal lines = site 2, circles with crosses = site 3, circles with vertical lines = site 4. Dashed line represents transport via ebullition or active plant transport without any isotopic fractionation. Values are means of pore-gas samples from 5-35 cm depth and chamber flux measurements. Graphs show mean values and standard deviations from three replications at each site. n = 1-3.**

Table 1: Overview of the four sites' distances to the water reservoir and of the composition of the vegetation in terms of coverage [%] of plant functional types (PFT) in hollows and abundances of plant species (vascular plants and mosses, excluding liverworts and hornworts). Abundances are abbreviated as follows: "*d*" means "dominant" (> 75 %), "*a*" means "abundant" (51-75 %), "*f*" means "frequent" (26-50 %), "*o*" means "occasional" (11-25 %), and "*r*" means "rare" (1-10 %). Because *Sphagnum* mosses were very hard to distinguish in the field, we only determined the abundance of the most abundant *Sphagnum* species of each site.

| Site 1 | | Site 2 | | Site 3 | | Site 4 | |
|---|---|---|---|---|---|---|---|
| **Distance to reservoir [m]** | | | | | | | |
| 800 | | 550 | | 400 | | 200 | |
| | | | | | | | |
| **Coverage of PFT [%]** | | | | | | | |
| ***Sphagnum* spp.:** 100 | | 100 | | 100 | | 60 | |
| **Graminoids:** 10 | | 30 | | 30 | | 10 | |
| **Shrubs:** 8 | | 5 | | 4 | | 30 | |
| | | | | | | | |
| **Plant species** | | | | | | | |
| *Sphagna* | | | | | | | |
| *S. magellanicum* | a | *S. magellanicum* | | *S. magellanicum* | | *S. magellanicum* | |
| *S. capillifolium* | a | *S. capillifolium* | a | *S. capillifolium* | a | *S. capillifolium* | |
| *S. fuscum* | | *S. fuscum* | | *S. fuscum* | | *S. fuscum* | f |
| *S. squarrosum* | | *S. wulfianum* | | *S. girgensohnii* | a | *S. girgensohnii* | f |
| *S. angustifolium* | | *S. angustifolium* | a | *S. squarrosum* | | *S. wulfianum* | |
| | | | | *S. wulfianum* | | | |
| | | | | *S. angustifolium* | | | |
| | | | | *S. cuspidatum* | | | |
| | | | | | | | |
| **Other mosses** | | | | | | | |
| *Polytrichum* spp. | o | *Polytrichum* spp. | o | *Polytrichum* spp. | o | *Polytrichum* spp. | r |
| *Rhytidiadelphus triquestrus* | r | *Polytrichum* spp. | o | *Polytrichum* spp. | o | | |
| | | | | | | | |
| **Graminoids** | | | | | | | |
| *Carex disperma* | f | *Scheuchzeria palustris* | o | *C. disperma* | o | *C. disperma* | o |
| *Carex oligosperma* | r | *C. disperma* | o | *C. magellanica* | o | *E. angustifolium* | r |
| *Eleocharis palustris* | r | *C. oligosperma* | r | *Dulichium arundinaceum* | f | *E. vaginatum* | o |
| *Eriophorum angustifolium* | r | *Carex limosa* | r | *E. palustris* | o | *E. virginicum* | o |
| *Eriophorum vaginatum* | o | *Carex magellanica* | r | *E. angustifolium* | o | *J. effusus* | r |
| *Eriophorum virginicum* | o | *E. palustris* | o | *E. vaginatum* | f | | |
| *Juncus effusus* | r | *E. angustifolium* | o | *E. virginicum* | f | | |
| | | *E. vaginatum* | f | *J. effusus* | o | | |
| | | *E. virginicum* | f | | | | |
| | | *J. effusus* | r | | | | |

**Shrubs**

| | | | | | | | |
|---|---|---|---|---|---|---|---|
| *Aronia melanocarpa* | r | *Myrica gale* | r | *M. gale L.* | o | *M. gale* | a |
| *Andromeda glaucophylla* | r | *A. glaucophylla* | r | *A. glaucophylla* | r | *A. glaucophylla* | r |
| *Chamaedaphne calyculata* | o | *C. calyculata* | o | *C. calyculata* | o | *C. calyculata* | o |
| *Kalmia polifolia* | o | *K. polifolia* | r | *K. polifolia* | r | *K. polifolia* | r |
| *Rhododendron groenlandicum* | o | *R. groenlandicum* | r | *R. groenlandicum* | r | *R. groenlandicum* | o |
| *Vaccinium myrtilloides* | r | | | | | *V. oxycoccos* | o |
| *Vaccinium oxycoccos* | r | | | | | | |

**Trees**

| | | | | | | | |
|---|---|---|---|---|---|---|---|
| *Larix laricina* | r | *L. laricina* | r | *P. strobus* | r | *B. pumila* | r |
| *Picea mariana* | r | *P. strobus* | r | | | | |
| *Pinus strobus* | r | *B. pumila* | r | | | | |
| *Betula pumila* | r | | | | | | |

**Herbs**

| | | | | | | | |
|---|---|---|---|---|---|---|---|
| *Sarracenia purpurea* | r | *S. purpurea* | r | *Maianthemum trifolium* | r | *S. purpurea* | r |
| *Drosera rotundifolia* | r | *D. rotundifolia* | r | *S. purpurea* | r | *D. rotundifolia* | r |
| | | | | *D. rotundifolia* | r | | |

Table 2: Values of $\alpha_C$ ±1 SD obtained from silicone samplers in 35 cm depth at sites 1 to 4 from June to September 2015. $\alpha_C$ values between 1.04 and 1.055 indicate the prevalence of the acetoclastic methane production pathway, $\alpha_C$ values higher than 1.065 indicate the hydrogenotrophic pathway.

| Site | 1 | | 2 | | 3 | | 4 | |
|---|---|---|---|---|---|---|---|---|
| | $\alpha_C$ | *std.dev* | $\alpha_C$ | *std.dev* | $\alpha_C$ | *std.dev* | $\alpha_C$ | *std.dev* |
| **June** | 1.068 | | 1.064 | *0.004* | 1.061 | *0.004* | 1.056 | *0.004* |
| **July** | 1.042 | | 1.044 | | 1.058 | *0.001* | 1.048 | |
| **August** | 1.043 | | 1.046 | *0.001* | 1.052 | *0.004* | 1.045 | *0.002* |
| **September** | 1.066 | *0.007* | 1.057 | *0.002* | 1.058 | *0.003* | 1.051 | *0.002* |