# Peer review of "Differential response of carbon cycling to long-term nutrient input and altered hydrological conditions in a continental Canadian peatland"

_Biogeosciences, 2017_

## Referee Comment (RC1) · Anonymous Referee #1 · 3 Jul 2017

This manuscripts reports an interesting dataset on C cycling at a temperate peatland, affected by increased nutrient input from a nearby reservoir. Carbon dioxide and methane fluxes were measured over a period of 1.5 years from four sites representing variable wetness, vegetation type and distance from the reservoir, and the flux measurements were accompanied by detailed soil profile measurements of $CH_4$ and DIC concentration. Carbon stable isotopes were used in order to gain more information about $CH_4$ production, oxidation and transport.

The paper is well written and logically structured, and the appearance of the figures

is very good. The methods are described in great detail, and the authors are clearly experts in selection and implementation of their field and analytical methods. The value of this work is in the high quality and completeness of the data set. I still belief that these data could be used more effectively, and the overall relevance of the paper could be greatly improved, by taking the following comments into consideration.

***Major comments

1. The match between the content of the manuscript and the title is not ideal at the moment. The title and especially the starting sentence of the abstract make one expect a comparison of carbon cycling between anthropogenically altered vs. natural sites. If this is the focus, it would be important to describe the transect better in the abstract and also in the methods section (page 4, lines 2-3 & from lines 16 onwards): How much did the hydrological condition change along the transect, and was the human impact related to drying or wetting or to fluctuating water table? And, in the abstract, how much did the nutrient infiltration change along the transect (data in Table 1)?

Further, instead of reporting just the results from the two highly affected sites 3&4 in the abstract, you should compare the results between anthropogenically altered vs. natural sites. This would justify the last sentence which claims clear anthropogenic effects on C cycling.

This comparison between affected and unaffected sites should be the view-point throughout the MS. For example: • By rearranging Fig. 3 & 4 so that instead of showing various parameters from the same site in the same figure you would show a single parameter from all of the sites in the same figure. • By adding here some indication of the reservoir effect: Distance of the reservoir in the table itself, or descriptive sentence in the table caption. • By focus the introduction better from general description of factors affecting in peatland C cycling towards a description of the effects of anthropogenic activities on it. It should be stated clearly, citing the relevant literature, why is it important to understand effects of increased nutrient inputs and changed

water level on carbon cycling. This topic it touched on the paragraph starting on page 2, line 24, but also there anthropogenic effects are not sufficiently discussed to match with the title of the paper. Also, the motivation statement on page 3, lines 4-8 is very general. Could you develop a more specific research question that suits for this particular study? • By rewriting the Concluding remarks section to answer the questions posed by the title and the introduction section.

As you state in the discussion section (page 14, lines 13-15), it is hard if not impossible to separate the wetness effects from the nutrient infiltration effects. Thus, to draw any conclusions about anthropogenic effects on C cycling, it should be carefully considered how the data presentation is organized to serve that purpose.

2. The MS includes interesting isotopic data of the CH4 emission and porewater DIC and CH4. A better explanation of how the stable isotopic data can be interpreted would be very much needed already in the introduction section. In the discussion section (page 15, line 24) you mention that the isotopic signature in methane is affected by methane production, oxidation and transport but you do not explain anywhere why and how the isotopic composition is affected by these processes. Furhter, the discussion of isotopic data is related to methane oxidation. Could the dominant methane production pathway (acetoclastic, hydrogenotrophic) or transport pathway have caused differences in isotopic sigantures and how? At the moment, the discussion on isotopic signatures is related to methane oxidation only.

3. In many occasions, you refer often to your own, yet unpublished work (Berger et al., submitted). Since that work seems to contain information quite crucial for the present paper, it is somewhat problematic that the paper is not available for the reader. If the submitted paper has not been published meanwhile, you should consider elaborate those results in more detail when necessary, e.g., in the methods section, page 4, line 2-3 about the hydrological changes caused by the reservoirs and page 4, line 23.

***Minor comments

page 1, lines 18-19 & page 11, lines 19-21: The study period includes a full year of measurements. It would be good to give values also on cumulative annual fluxes. This was enable using this carefully collected data in flux syntheses, and facilitate comparison with literature values.

page 2, lines 3-5: One-sentence paragraphs should be avoided. I suggest combining this sentence with the next paragraph. See also page 4, line 29; and page 17, lines 16-19.

page 2, lines 26-28: Reference missing.

page 3, lines 12-15 & page 4, line 2: Also here, I would like to see a mentioning about how the hydrology is altered – drying or wetting, or more variable in the course of the year?

page 4, line 13: microtopography is a single word

page 4, lines 18-19: Listing the sites starting from number 4 is counterintuitive. Would it be possible to change the order in which you mention the sites, or simply change the numbering? You indicate that site 2 was further away from the reservoir than site 3, but it would be better to describe the whole transect, e.g., that the distance from reservoir decreases with growing number.

page 4, lines 25-28: It is not clear if and how this is related to the vicinity of reservoir?

page 5, line 8: At the first appearance, write the complete instrument type instead of the abbreviation FTIR.

page 5, line 18: Regarding UV-VIS, see the previous comment.

page 6, line 12: Was the image analysis based on satellite/aerial or other imagery? Please specify!

page 8, lines 5-6: Have you tested if there were any discrimination against the lighter isotope during diffusion into the silicon collectors?

page 8, lines 13: Please check the sentence (some words missing/in a wrong order).

page 9, lines 29-31: Please check the sentence (some words missing/in a wrong order).

page 11, lines 22-24: These are important results for this paper. But, can you really say that it is anthropogenic effect, or just a consequence of different location (edge effect, more mineral site?). It's interesting that the site receiving more nutrients is showing lower CO2 uptake.

page 13, lines 19-21: Or, is the higher lability of organic matter caused by higher productivity and high input of labile compounds from vegetation? This site showed the highest C accumulation (page 11, lines 19-21). If it is a reservoir effect, should not the organic matter at site 4 be even more labile? Now, the site 4 was showing the highest proportion of aromatic compounds.

page 13, lines 19-21: Please add references: In recent studies by Bragazza et al. . ...

page 14, line 21: Decrease in the CO2-sink strength in response to what?

page 16, lines 3-4: You write about "deepening of soil oxygenation probably promoting a highly active methanotrophic bacteria community, which drew CH4 from the atmosphere down to that depth". Why do you think it was atmospheric and not peat-derived CH4 that was oxidized at 15 cm? Atmospheric methane cannot diffuse to the soil against the concentration gradient (when the pore-water concentrations are above ambient).

page 16, line 5: Why enriched signals would mean low CH4 production? Do you mean more CH4 production via the acetoclastic pathway that results in heavier methane than the hydrogenotrophic pathway?

page 16, lines 20-21: Besides transporting CH4 through the aerobic peat layers without exposing it to oxygen plant-mediated transport also strongly discriminates against the heavier methane (Chanton, 2005). Because of this, plant transport can create even

lighter methane that is present anywhere in the peatland. It would be important to mention this in the discussion about isotopic signatures.

page 16, lines 23-28: Yes, probably most of the methane is oxidized during the diffusion, and thus, the amount of methane reaching the atmosphere by diffusion is low. So even with low coverage of aerenchymous plants, most of the methane that is actually entering the atmosphere is emitted through them.

page 16, lines 28-30: It seems to me that you have done all the necessary pre-cautions to avoid methodological biases in the data. Please specify, what actually makes you suspect some methodological problems particularly in low water table conditions.

page 17, line 9: Instead of just saying results, it would be better to specify which particular parameter you mean here.

page 17, lines 16-19: Long and complicated sentence, please consider splitting it into two sentences.

Fig. 1: For better clarity, please mark the Luther lake reservoir in the figure.

Fig 6 (and page 12, lines 28-29). In this figure, you have decided to use the porewater d13C-CH4 at 5 cm. However, the methane pool at this depth does not necessarily represent very well the origin of the methane emissions, since the ebullitive and plant-mediated fluxes are originating from deeper layers. Hornibrook (2009) was using the average from 0 to 50 cm. It would be interesting to see how the figure looks if the porewater methane at depth is not excluded. Although the differences between different depths were not significant, Fig. 5 shows that especially at sites 1 and 2, the porewater CH4 has different isotopic composition at 5 cm than at deeper depths.

Fig 6. caption, line 5: A typo in the word 'circles'. Table 1: The differences in stoichiometry are not evident, and I do not see a clear transect here. Could you test the differences statistically and mark it in the table? Or, would the amounts instead of ratios reveal the pattern more clearly?

[Figure]

---

## Referee Comment (RC2) · Anonymous Referee #2 · 4 Aug 2017

The manuscript by Berger and co-authors is very interesting because the authors provide a comprehensive dataset on how soil carbon cycling changes along a transect of four study sites (from undisturbed to disturbed conditions) in a peatland complex in Ontario from April 2014 until September 2015. They used a variety of methods that complement each other in space and time (e.g. chamber flux measurements of $CO_2$ and $CH_4$, DIC and $CH_4$ concentration measurements at different soil depths, stable isotope measurements of $CO_2$ and $CH_4$, FTIR analysis of organic matter and pore-water and measurements of ancillary variables such as air and water temperature, photosynthetically active radiation and water table depth below surface).

[Figure]

The authors raise the major question, how peatland carbon fluxes respond to anthropogenically changed hydrological conditions and long-term nutrient-infiltration effects. Their major answer is that plant functional type may be a key variable to predict how soil carbon cycling in peatlands will respond to future nutrient inputs and changes in hydrology. Shrub dominated disturbed peatlands may turn into carbon sources, while graminoid-moss dominated peatlands "may maintain the peatland's carbon storage function".

However, I have few major concerns but after a thorough revision and/or modification of the manuscript it would be great to see this manuscript published in the Biogeoscience journal.

Major comments:

The authors point out that it is not new that plant functional types may have a strong influence on ecosystem soil carbon dynamics but I completely agree with the authors that "there is a gap of knowledge in terms of interactions between peat and plants under IN-SITU CONDITIONS". This makes this manuscript very valuable. However, I am not an author of the paper "Peatland vascular plant functional types affect methane dynamics by altering microbial community structure. (Robroek et al. 2015, doi: 10.1111/1365-2745.12413)" but the authors of this manuscript should cite that paper and compare both results. Robroek et al. (2015) nicely demonstrate that resilience of peatland CH4 dynamics, and therefore also CO2 dynamics, to climate change may depend on interaction between microbes and plant functional types.

I think the manuscript would greatly benefit from a more thorough discussion about the potential role of methanogens driving soil methane dynamics at the four different sites. In the current study, the authors measured stable carbon isotope ratios of CH4 and CO2 comprehensively. Hence, apparent fractionation factors could be easily measured (Angel et al. 2011; doi:10.1371/journal.pone.0020453 or McCalley et al. 2014; doi:10.1038/nature13798), the different pathways of methanogenesis identified and discussed. Now, the authors attribute the change of isotopic signals to changes in methane oxidation. This is very speculative and not sufficient. The change in 13CH4 may result from a shift from hydrogenotrophic to aceitclastic methanogenesis, especially during drier months (see Hodgkins et al. 2014; www.pnas.org/cgi/doi/10.1073/pnas.1314641111 2014 or McCalley et al. 2014; ; doi:10.1038/nature13798). However, this should be discussed in the manuscript.

The authors state that it is clearly visible that ratios of C/N, C/Mg and C/K in peat soil are decreasing from site 1 to site 4. I do not see this pattern when I look at Table 1. C/K is higher at site 2 and 3 than at site 1. C/Mg is lowest at site 1. I guess C/N ratios do not differ between site 1, 2 and 3. Furthermore, I guess there are no significant differences in C/P ratios between the different sites. N/P ratios are higher at site 2 than site 3 and C/Ca ratios are lowest at site 2. Please, check your data.

However, it would be great to have a look at the submitted publication or if the authors would incorporate more convincing information. Otherwise the authors cannot state that "it becomes evident that the peatland was exposed to nutrient infiltration form the water reservoir and thus elevated nutrient concentrations occurred in vicinity to the water reservoir." (P13, L2-L6) and should reformulate the whole discussion section!

Specific comments:

Titel: Currently, I do not see that nutrients drive carbon dynamics at your sites.

Abstract: If you mention the other methods in the abstract, you should mention FTIR analysis as well.

P1, L17-L19: All the sites are characterized by wet conditions. These are peat soils.

P1, L19-L20: Low 13CH4 may be caused by more hydrogenotrophically produced CH4.

P1, L24: or more aceticlastically produced CH4. More labile organic matter may favor aceticlastic methanogenesis.

P3, L8: I do not see a gradient in nutrient availability.

P3, L10: Please, calculate apparent fractionation factors for methanogenic pathways.

P3, L16: I do not see that nutrient inputs are greatest in peatland periphery (see Table 1).

P3, L 18: Why should CH4 emissions be greatest at the graminoid dominated sites? There is no link to this hypothesis in the introduction.

P3, L19-21: You should also discuss CH4 production pathways.

P4, L22-28: This paragraph is very essential for the main message of the manuscript. Unfortunately, the data do not support these statements. It would be great to see more data that support these ideas.

P5, L8: Please, write out FTIR analysis.

P5, L8-L10: I am not familiar with FTIR analysis. "For pore-water samples 2 mg of oven-dried organic matter..." is that correct?

RESULTS section: The presentation of the results are too cluttered. Results that are not significant are described quite often (see P10, L7, L10) and sometimes it is not clear if results are significant or not (see P10, L16-17, L19-26). It would be better to mark significant differences in the figures and to highlight significant results or only few non-significant results in text, if they really enrich and/or support the guiding questions in the manuscript.

P13, L15-L16: This is repetition of results.

P13, L19-L21: This is very speculative. Did you check FTIR ratios of inflowing water? May be you can provide some references.

P13, L22-L23: I am not convinced by this statement. The difference between site 4 and the three other sites may be simply by chance.

P13, L24-L27: So, it is not the vicinity to the reservoir but the vegetation that drives carbon cycling processes?

P14, L11-L19: This is repetition of results.

P14, L25: It would be great to see the data.

P15, L1-L2: Site 4 shows second highest CH4 release. Then you cannot state that graminoid sites show highest CH4 emissions. I would emphasize to reformulate the introduction and the hypothesis in such a manner that it becomes clearer to the reader why you have stated your hypothesis. Now, the discussions seems to be much too blurred.

P15, L6: What means "healthy" Sphagnum moss community?

P15, L22 – P16, L33: see major comments.

P17, L2 – L19: see major comments.

Figures: It would be great to have figures with higher resolution. In Figure 6, I can hardly identify the difference between the circles.

Table1: Please, mark significant differences.

---

## Author Comment (AC1) · 4 Sep 2017

Dear Reviewer,

we very much appreciate your thoughtful review of our manuscript. Thank you for your time and your valuable ideas. We absolutely agree that taking up your suggestions greatly improves our manuscript and we will do so as outlined. Please find our answers to your comments below.

* This manuscript reports an interesting dataset on C cycling at a temperate peatland, affected by increased nutrient input from a nearby reservoir. Carbon dioxide and

[Figure]

methane fluxes were measured over a period of 1.5 years from four sites representing variable wetness, vegetation type and distance from the reservoir, and the flux measurements were accompanied by detailed soil profile measurements of CH4 and DIC concentration. Carbon stable isotopes were used in order to gain more information about CH4 production, oxidation and transport. The paper is well written and logically structured, and the appearance of the figures C1 BGD Interactive comment Printer-friendly version Discussion paper is very good. The methods are described in great detail, and the authors are clearly experts in selection and implementation of their field and analytical methods. The value of this work is in the high quality and completeness of the data set. I still belief that these data could be used more effectively, and the overall relevance of the paper could be greatly improved, by taking the following comments into consideration.

***Major comments

1. The match between the content of the manuscript and the title is not ideal at the moment. The title and especially the starting sentence of the abstract make one expect a comparison of carbon cycling between anthropogenically altered vs. natural sites. If this is the focus, it would be important to describe the transect better in the abstract and also in the methods section (page 4, lines 2-3 & from lines 16 onwards): How much did the hydrological condition change along the transect, and was the human impact related to drying or wetting or to fluctuating water table? And, in the abstract, how much did the nutrient infiltration change along the transect (data in Table 1)?

– We understand your concern. This present study is following up on a study, which was very recently published in SBB (114 (2017) 131-144; http://dx.doi.org/10.1016/j.soilbio.2017.07.011). While writing the current paper, this other study had not been published so we could not well refer to it. The study site was described in detail in this SBB-paper. As elaborated there, the water reservoir affects the entire northern tip of our peatland site. We find in our data that areas further away from the reservoir are likely less affected, but it is not entirely possible to distinguish

'natural' from 'anthropogenically altered' sites among the sites 1-4 (800-200 meters distance from the reservoir). Only areas further away from the reservoir could likely be regarded as pristine. So, our transect of study sites as presented in this present study rather assesses strongly altered sites vs. less altered sites with respect to the investigated features. Unfortunately, this entire gradient was not as obvious in the beginning of the study, so this part is indeed missing. So, our objective for this paper was to investigate carbon cycling along this transect and to identify effects of altered conditions along the transect on carbon cycling and fluxes. We agree: the transect should be described better. In that sense, the abstract and the method section will be extended, providing the information you requested.

As far as the title is concerned, another option would be: "Differential response of carbon cycling in a continental Canadian peatland to long-term nutrient input and altered hydrological conditions"

\* Further, instead of reporting just the results from the two highly affected sites 3&4 in the abstract, you should compare the results between anthropogenically altered vs. natural sites. This would justify the last sentence which claims clear anthropogenic effects on C cycling.

– We agree. So far, we were only pointing out the results from the sites 3 and 4. The reason is that here we found most statistically significant differences. At the same time, the sites 3 and 4 were the most altered ones. So, we assumed that these significantly different results for distinctly altered sites would be most convincing, and focused our discussion on the sites 3 and 4. The last sentence of the abstract is indeed somehow misleading, we agree. We will consider including results from site 1 and 2. As reviewer 2 mentioned that we should reduce the presentation of non-significant differences, we will try to balance these two issues.

\* This comparison between affected and unaffected sites should be the view-point throughout the MS. For example: âAËŸ c By rearranging Fig. 3 & 4 so that instead of

′ showing various parameters from the same site in the same figure you would show a single parameter from all of the sites in the same figure. âAËŸ c By adding here some ′ indication of the reservoir effect: Distance of the reservoir in the table itself, or descriptive sentence in the table caption. âAËŸ c By focus the introduction better from general ′ description of factors affecting in peatland C cycling towards a description of the effects of anthropogenic activities on it. It should be stated clearly, citing the relevant literature, why is it important to understand effects of increased nutrient inputs and changed water level on carbon cycling. This topic it touched on the paragraph starting on page 2, line 24, but also there anthropogenic effects are not sufficiently discussed to match with the title of the paper. Also, the motivation statement on page 3, lines 4-8 is very general. Could you develop a more specific research question that suits for this particular study? âAËŸ c By rewriting the Concluding remarks section to answer the questions ′ posed by the title and the introduction section.

– Unfortunately, the manuscript cannot provide a comparison between "affected" and "unaffected" sites (see also above). However, we can provide a comparison of carbon cycling of strongly altered and less altered sites. We will clarify this point to avoid misunderstandings here.

We re-arranged figures 3 & 4 as suggested (please see the following two figures below). We agree that this very much improves clarity. The development of pore gas DIC and CH4 concentrations over time could, however, not be re-arranged as the 3D-plots would get too complicated.

Having re-arranged the figures as you suggested, differences between the sites in terms of CO2 and CH4 fluxes become indeed much more obvious.

The stochiometric ratios from table 1 will be removed. Table 1 in its previous form was thought to summarize some results from the SBB-paper, but it turns out that presenting only that little information about the sites is quite misleading. Instead a better description of the study sites will be added to the methods section and the reader will be

referred to the SBB-paper. However, more information on plant species abundances will be added to table 1 in a way that the vegetation gradient becomes more visible.

The introduction will be revised according to your suggestion. The paragraph (starting on page 2, line 24) dealing with factors affecting peatland C cycling will be extended towards a description of the effects of anthropogenic activities. In this regard, we will better elaborate the results from long-term peatland fertilization experiments (Mer Bleue, Whim Bog, Degerö Stormyr) as well as impacts of inundation on neighboring peatland ecosystems (e.g. Kim et al., 2015; Ballantyne et al., 2014) and we will also provide a more adequate summary of what is known about gaseous carbon fluxes in relation to an altered plant community (e.g. Robroek et al., 2015). Taking also into account our own recently published paper (Berger et al., 2017) it should then be much more obvious why there still is a need to study changes in peatland C-cycling after increased nutrient inputs and changed water levels.

As far as a more specific research question is concerned, we developed new hypotheses which better meet the focus of our study:

1) peripheral nutrient input accelerates C cycling at the affected sites, reflected in more decomposed peat, 2) increased abundance of vascular plants can increase $CO_2$ uptake but also change patterns of $CH_4$ production and emission, in particular if graminoids dominate, 3) long-term nutrient enrichment in combination with hydrologically altered conditions may cause differential responses of carbon cycling and does not necessarily cause a loss of the C-sink function of peatland ecosystems.

Including these changes into the abstract, introduction, site description, presentation of results, and discussion, of course, the concluding remarks section will be adjusted accordingly.

* As you state in the discussion section (page 14, lines 13-15), it is hard if not impossible to separate the wetness effects from the nutrient infiltration effects. Thus, to draw any conclusions about anthropogenic effects on C cycling, it should be carefully considered

how the data presentation is organized to serve that purpose.

– We agree. We believe that re-arranging the figures 3 and 4 really helped in this regard.

* 2. The MS includes interesting isotopic data of the CH4 emission and porewater DIC and CH4. A better explanation of how the stable isotopic data can be interpreted would be very much needed already in the introduction section.

– A better explanation will be added. This will also cover the interpretation regarding underlying pathways and methanotrophic activity.

* In the discussion section (page 15, line 24) you mention that the isotopic signature in methane is affected by methane production, oxidation and transport but you do not explain anywhere why and how the isotopic composition is affected by these processes. Furhter, the discussion of isotopic data is related to methane oxidation. Could the dominant methane production pathway (acetoclastic, hydrogenotrophic) or transport pathway have caused differences in isotopic sigantures and how? At the moment, the discussion on isotopic signatures is related to methane oxidation only.

– We understand your concern. Of course, the dominant CH4 production and transport pathway cause differences in isotopic signatures. This will be better explained in the revised version in the introduction. You might be surprised by reading that a previous version of our manuscript contained also an interpretation to differentiate dominant methane production pathways. According to Whiticar et al. (1986) the isotope fractionation factor $\alpha C$ was calculated:

$\alpha C = (\delta 13C\text{-}CO2 + 1000) / (\delta 13C\text{-}CH4 + 1000)$.

Accordingly, Fig. 3 had been part of our manuscript, but was removed later.

We removed the figure and everything related to distinguishing methane production pathways for the following reasons:

1) Whiticar et al. (1986) stated that $\alpha C$ values between 1.04 and 1.055 indicate the prevalence of the acetate fermentation pathway, whereas $\alpha C$ values higher than 1.065 support a shift towards the $CO_2$ reduction pathway. On the other hand, values of $\alpha C$ typically observed for the acetoclastic pathway could also arise from methanotrophic activity, yielding an enrichment in 13C of the residual $CH_4$. Values of $\alpha C$ measured in our study site covered a broad range from 1.013 to 1.082. This is a wide range compared to other studies which found values between 1.028 and 1.061 (Hornibrook et al., 2000), 1.03 and 1.07 (Kotsyurbenko et al., 2004), 1.046 and 1.075 (Steinmann et al., 2008) or 1.022 and 1.053 (Hornibrook et al., 1997). As Fig. 3 shows, the average water table in many cases dropped below our second sampling depth (-15 cm) and sometimes even approached to -25 cm. The fact that $\alpha C$ values were lowest when water levels were lowest during the summer 2015, i.e. the clear relation of $\alpha C$ to water table levels, suggests that not only a change in methane production pathway but even more so methane oxidation affected our $\alpha C$ values. We think that existing studies often did not cover this range of partly unsaturated conditions at sampling depths and thus less data is so far available from samples clearly affected by methanotrophic activity. Moreover, a discussion of values of $\alpha C$ is from our point of view too often limited to a discussion of pathways although at the water table under partly oxic conditions methanotrophy may be much more likely to affect the isotopic signature of methane.

2) We think that distinguishing methane production pathways would thus only be possible when considering only $\alpha C$ values from below the water table levels, i.e. from the saturated zone. However, as water table depth may even be different within replicates of gas sampling spots within one site and the water table does not necessarily coincide with the transition from oxic to anoxic conditions (roots!) obtaining $\alpha C$ values from saturated, anoxic layers only was difficult. So, we decided not to present the fractionation factors in such detail and therewith we excluded the story on different $CH_4$ production pathways.

To summarize: we think distinguishing methane production pathways with our data
would be critical. That is why we would prefer to refrain from it. However, since reviewer 2 also raised that point, we will add a paragraph, which discusses methane production pathways and how isotopic composition is affected by these processes and which also explains, why we think that our data is not suitable for distinguishing methane production pathways. We hope that the reviewer could agree with our position here following the given explanation above; in case not, we would consider adding more discussion of pathways, but we would first like to clarify and support why we prefer to primarily discuss the impact of methanotrophy.

* 3. In many occasions, you refer often to your own, yet unpublished work (Berger et al., submitted). Since that work seems to contain information quite crucial for the present paper, it is somewhat problematic that the paper is not available for the reader. If the submitted paper has not been published meanwhile, you should consider elaborate those results in more detail when necessary, e.g., in the methods section, page 4, line 2-3 about the hydrological changes caused by the reservoirs and page 4, line 23.

– The paper is published now in Soil Biology & Biochemistry 114 (2017) 131-144 http://dx.doi.org/10.1016/j.soilbio.2017.07.011 We apologize that this information could not be presented earlier.

***Minor comments

* page 1, lines 18-19 & page 11, lines 19-21: The study period includes a full year of measurements. It would be good to give values also on cumulative annual fluxes. This was enable using this carefully collected data in flux syntheses, and facilitate comparison with literature values.

– Will be done.

* page 2, lines 3-5: One-sentence paragraphs should be avoided. I suggest combining this sentence with the next paragraph. See also page 4, line 29; and page 17, lines 16-19.

[Figure]

– Will be done.

\* page 2, lines 26-28: Reference missing.

– Will be added.

\* page 3, lines 12-15 & page 4, line 2: Also here, I would like to see a mentioning about how the hydrology is altered – drying or wetting, or more variable in the course of the year?

– It is a bit tricky to do so in a generalized way. As explained in the SBB-paper, we lack particular historic data for a detailed overview of the impact of the reservoir flooding on the adjacent peatland. We only know for sure that flooding of the reservoir in 1954 rewetted large areas of the peatland, which had been subject to drainage during previous decades. So, the reservoir had a clear wetting impact on the peatland. Through measurements with an eddy covariance station (results will be part of another, future manuscript) we observed that during dry and hot days the water loss to the footprint area through evapotranspiration was much smaller than the soil water level dropdown measured with our water level sensors (even more so when reservoir water was released through the dam). On the other hand, when there was heavy rainfall, soil water levels rose more than the amount of precipitation measured in our peatland would have suggested. That clearly indicated that peatland and reservoir water levels were strongly connected in a way that on hot, dry days, the reservoir drew water out of the peatland, while on rainy days the reservoir pushed water into the peatland and the amount of water moved seemed to be related to the amount of rainfall and to the severeness of summer heat. However, such fluxes of water cannot be adequately analyzed without a sound modelling approach. Nevertheless, it became obvious that the reservoir seems to control to a large extent the observed peatland water level fluctuation. Probably, the water levels of the entire northern tip of the peatland are more variable than it would be expected under natural conditions. Unfortunately, from our own water level data (May to September 2012 and November 2013 to September 2015) it appeared difficult

evaluating how the reservoir water level is affecting the peatland water levels in more detail. Based on our available data and the work published in the earlier manuscript about the site, we have strong support that those sites in closer vicinity to the reservoir are more affected. The focus of this present manuscript are the peatland hollows. When only looking at the hollows, site 4 and site 3 were indeed wetter than the sites 2 and 1, but when also considering hummocks (as done in the SBB manuscript), site 4 and site 1 were drier than the sites 2 and 3. We will add more information to the introduction and to the description of the study site in terms of how the peatland water levels are affected by the reservoir.

* page 4, line 13: microtopography is a single word

– Will be corrected.

* page 4, lines 18-19: Listing the sites starting from number 4 is counterintuitive. Would it be possible to change the order in which you mention the sites, or simply change the numbering? You indicate that site 2 was further away from the reservoir than site 3, but it would be better to describe the whole transect, e.g., that the distance from reservoir decreases with growing number.

– We are sorry, but we don't think that would be possible. As the site names were established in the SBB paper, changing the names now would be a bit confusing. . .

* page 4, lines 25-28: It is not clear if and how this is related to the vicinity of reservoir?

– You are right. Sorry for the confusion. We were thinking it could be helpful to provide that additional information as through FTIR and UV-vis the manuscript includes data on quality of peat and pore water. However, presenting that information in that context might be misleading. We will probably remove the two sentences here, and add them to the more general description of the site a bit further above.

* page 5, line 8: At the first appearance, write the complete instrument type instead of the abbreviation FTIR.

– Will be done.

* page 5, line 18: Regarding UV-VIS, see the previous comment.

– Will be done.

* page 6, line 12: Was the image analysis based on satellite/aerial or other imagery? Please specify!

– It was done via aerial imagery with images obtained from UAV flights. This information will be added.

* page 8, lines 5-6: Have you tested if there were any discrimination against the lighter isotope during diffusion into the silicon collectors?

– This is indeed a good point. We tested this some years ago and found no differences in the isotopic composition for CH4 and CO2 (at low pH and predominance of H2CO3/CO2). This is however, not published as meanwhile also other studies have used this technique and we thus assumed that it is accepted. The concentrations in silicon tubes adjust at time scales of hours (tested by us for CO2, CH4, H2, and N2O) as described in Kamman et al. (2001). So isotopic discrimination when sampling monthly should be smaller than the analytical error of our measurements. The samplers were for example previously used in Goldberg et al. (2008, 2010), Knorr et al. (2008), Zou et al. (2011), Berger et al. (2013), Novak et al. (2015).

* page 8, lines 13: Please check the sentence (some words missing/in a wrong order).

– Will be done. "Silicon tubes for isotope sampling had a volume of 20 ml in 5 cm depth and 5 ml in the other depths. Bigger sampler sizes in the close to surface peat layer were necessary as a sample volume of at least 20 ml at sufficiently high concentration (2.5 < x < 2000 ppm) was needed for the isotope analysis."

* page 9, lines 29-31: Please check the sentence (some words missing/in a wrong order).

– Will be done.

* page 11, lines 22-24: These are important results for this paper. But, can you really say that it is anthropogenic effect, or just a consequence of different location (edge effect, more mineral site?). It's interesting that the site receiving more nutrients is showing lower CO2 uptake.

– We agree that the observed effects can have different causes. However, looking at the entire transect the peat quality found at the sites suggest a quite similar history before dam construction at the site. So, based on our analysis the factor we can identify is the enrichment in nutrients and the concomitantly altered vegetation. Of course, site 4 thus had longest exposure to more minerotrophic conditions from intrusion of lake water. However, we would also consider this effect as anthropogenic then. We will provide a statement in this regard in our discussion.

* page 13, lines 19-21: Or, is the higher lability of organic matter caused by higher productivity and high input of labile compounds from vegetation? This site showed the highest C accumulation (page 11, lines 19-21). If it is a reservoir effect, should not the organic matter at site 4 be even more labile? Now, the site 4 was showing the highest proportion of aromatic compounds.

– That is a very interesting question! You are right, it could very well be the case that the higher lability of organic matter is caused by the vegetation (higher productivity → higher input of labile compounds). That idea will be included into the discussion. We first came to our conclusions as DOM is usually a small but easily accessible pool and should thus reflect a residual enrichment of refractory compounds. Presence of labile DOM we thus attributed to external input. Indeed, this would then also apply to site 4. This discussion was not straightforward and will be modified considering your suggestion.

* page 13, lines 19-21: Please add references: In recent studies by Bragazza et al. . ..

– Will be added.

* page 14, line 21: Decrease in the CO2-sink strength in response to what?

– In response to drought. That information will be added.

* page 16, lines 3-4: You write about "deepening of soil oxygenation probably promoting a highly active methanotrophic bacteria community, which drew CH4 from the atmosphere down to that depth". Why do you think it was atmospheric and not peat-derived CH4 that was oxidized at 15 cm? Atmospheric methane cannot diffuse to the soil against the concentration gradient (when the pore-water concentrations are above ambient).

– You are right. That is speculative. We thought it probably was atmospheric CH4, which entered the soil because the CO2 and CH4 concentrations in the upper peat layers were particularly low on that day (as indicated in Fig. 3). The water table even was below -15 cm depth (please see Fig. 3). Also, the surface peat looked quite dry during that summer period and the Sphagnum mosses were white and inactive (later in the year they recovered). So, our idea was that the CH4 in 5 cm depth was of atmospheric origin. But we have indeed no proof for this interpretation – we are aware of that. Therefore, we will adjust our discussion here and will be more cautious. Another observation which however supported our idea was that we could sometimes observe a CH4 flux from the atmosphere into the soil for certain plots of our study area. Such phenomena could be observed from dawn to sunset and in some occasions, this was observed for several weeks, unfortunately, we could not yet figure out an explanation. Of course, we know that peatlands can consume CH4 from the atmosphere (there are several studies), but we were surprised to observe this for several weeks in a row for certain plots. This data is not part of this manuscript but there is going to be another manuscript dealing with CH4 flux dynamics in which this issue will probably be addressed. Thus, we will downgrade our interpretation here as suggested.

* page 16, line 5: Why enriched signals would mean low CH4 production? Do you mean

more CH4 production via the acetoclastic pathway that results in heavier methane than the hydrogenotrophic pathway?

– Actually, we were trying to avoid speculations about acetoclastic and hydrogenotrophic pathways, because values of $\alpha C$ typically observed for the acetoclastic pathway could also arise from methanotrophic activity. Both processes would yield an enrichment in 13C of the residual CH4. We believe that it is rather CH4 oxidation, which took place because of the low water tables and the unsaturated conditions during the summer. So, we tried to boil it down to: methane production would yield less enriched 13C - CH4, while CH4 oxidation would leave behind comparatively more enriched CH4. We then interpreted more oxidation as less net production. We will clarify this point.

* page 16, lines 20-21: Besides transporting CH4 through the aerobic peat layers without exposing it to oxygen plant-mediated transport also strongly discriminates against the heavier methane (Chanton, 2005). Because of this, plant transport can create even lighter methane that is present anywhere in the peatland. It would be important to mention this in the discussion about isotopic signatures.

– Okay! Will be mentioned.

* page 16, lines 23-28: Yes, probably most of the methane is oxidized during the diffusion, and thus, the amount of methane reaching the atmosphere by diffusion is low. So even with low coverage of aerenchymous plants, most of the methane that is actually entering the atmosphere is emitted through them.

– That was our idea, too.

* page 16, lines 28-30: It seems to me that you have done all the necessary precautions to avoid methodological biases in the data. Please specify, what actually makes you suspect some methodological problems particularly in low water table conditions.

– For gas sampling for later isotope abundance analyses we used the same chambers

as used for the flux measurements. It is known that chamber measurements tend to overestimate CH4 fluxes a bit for several reasons (spatial heterogeneity, artificial pressure fluctuations induced by the chambers. . .). So, artefacts cannot be fully excluded. To counteract possible concerns in terms of data quality we verified our chamber flux data with eddy covariance flux data (Fig. 4). The sets of data are nicely comparable, however, CH4 fluxes measured with chambers were slightly increased in July and August when the water tables were lowest, which we think could have something to do with deeper CH4 pools becoming connected to the atmosphere under unsaturated conditions with dropping water tables as explained in Estop-Aragones et al. (2016). With chamber induced pressure fluctuations such CH4 pools might have been forced out of the peat. We would like to clarify that in the course of quality assurance/ quality checks while processing data, most likely all low-quality data was eliminated. Thus, we are sure that our data provided (fluxes and isotope data) is of very high quality. By mentioning about the issue in the first place we intended to point out the common shortcomings of the method; instead it probably downgraded our results, so we might remove the related sentences from the manuscript to avoid any misunderstandings.

* page 17, line 9: Instead of just saying results, it would be better to specify which particular parameter you mean here.

– Will be done.

* page 17, lines 16-19: Long and complicated sentence, please consider splitting it into two sentences.

– Will be done.

* Fig. 1: For better clarity, please mark the Luther lake reservoir in the figure.

– Will be done.

* Fig 6 (and page 12, lines 28-29). In this figure, you have decided to use the porewater d13C-CH4 at 5 cm. However, the methane pool at this depth does not necessarily

represent very well the origin of the methane emissions, since the ebullitive and plant-mediated fluxes are originating from deeper layers. Hornibrook (2009) was using the average from 0 to 50 cm. It would be interesting to see how the figure looks if the pore-water methane at depth is not excluded. Although the differences between different depths were not significant, Fig. 5 shows that especially at sites 1 and 2, the porewater CH4 has different isotopic composition at 5 cm than at deeper depths.

– We redrew the panel (e) of figure 6 using also $\delta$13C-CH4 -values from the deeper CH4. (please see Fig. 5)

* Fig 6. caption, line 5: A typo in the word 'circles'.

– Will be corrected.

* Table 1: The differences in stoichiometry are not evident, and I do not see a clear transect here. Could you test the differences statistically and mark it in the table? Or, would the amounts instead of ratios reveal the pattern more clearly?

– Thank you for pointing it out. We decided to remove the stochiometric ratios from the table and to describe the transect in the text while referring the reader to the SBB paper. Since the results are very complex, only presenting this little information turned out not to be helpful at all.

Figure captions:

Figure 1 (new Figure 3): Development of (a) Twater [° C], (b) wtd [cm], (c) CH4 fluxes [g CH4 m-2 d-1] and (d) – (f) CO2 fluxes (NEP partitioned into Reco and GPP) [g CO2 m-2 d-1], $\pm$ 1 SD (n=6) in hollows of the sites 1–4 from April 1st, 2014 through September 22nd, 2015. Negative CO2 and CH4 fluxes indicate uptake, positive fluxes indicate a release to the atmosphere. The dashed grey line in the NEP graph indicates a 0-flux.

Figure 2 (new Figure 4): Development of mean CH4 and mean DIC concentrations [$\mu$mol L-1], in hollows of the sites 1–4 from April 1st, 2014 through September 22nd,

2015.

Figure 3 (not to be included into the manuscript): $\alpha C$ values obtained from silicon samplers in 5, 15, 25 and 35 cm depth at sites 1 to 4 from June to September. squares = June (06/11), circles = July (07/08), triangles = August (08/27), diamonds = September (09/17). n = 1-3. Blue bars indicate the water table level at each site (averaged over one week before sampling). Vertical dashed lines indicate the thresholds for acetoclastic and hydrogenotrophic methanogenesis according to Whiticar et al., 1986. Values marked by the red circle probably reflect atmospheric signatures of CH4 and CO2 as the water table as well as the concentrations determined in the samplers were very low on these dates.

Figure 4 (not to be included into the manuscript): A comparison of our eddy covariance vs. chamber measurements for the NEE (left) and CH4 flux (right). The gray lines indicate the 1:1 lines. A good agreement is apparent; however, the chamber measurements detected higher CH4 emissions in July and August (circled in gray).

Figure 5 (manuscript Figure 6): dominant flux pathway of CH4 according to (Hornibrook, 2009). Empty circles = site 1, circles with diagonal lines = site 2, circles with crosses = site 3, circles with vertical lines = site 4. Dashed line represents transport via ebullition or active plant transport without any isotopic fractionation. Values are means of pore-gas samples from 5-35 cm depth and chamber flux measurements. Graphs show mean values and standard deviations from three replications at each site. n = 1-3.

References:

Ballantyne, D. M., Hribljan, J. A., Pypker, T. G., Chimmer, R. A.: Long-term water table manipulations alter peatland gaseous carbon fluxes in Northern Michigan, Wetlands Ecology and Management, 22, 35–47, 2014.

Berger, S., Gebauer, G., Blodau, C., Knorr, K.-H.: Peatlands in a eutrophic world –

Assessing the state of a poor fen-bog transition in southern Ontario, Canada, after long term nutrient input and altered hydrological conditions, Soil Biology and Biochemistry, 114, 131–144, 2017.

Berger, S., Jang, I., Seo, J., Kang, H., Gebauer, G.: A record of N2O and CH4 emissions and underlying soil processes of Korean rice paddies as affected by different water management practices, Biogeochemistry, 115, 317–332, 2013.

Estop-Aragones, C., Zajac, K., Blodau, C.: Effects of extreme experimental drought and rewetting on CO2 and CH4 exchange in mesocosms of 14 European peatlands with different nitrogen and sulfur deposition, Global Change Biology, 22 (6), 2285–2300, 2016.

Goldberg, S. D., Knorr, K.-H., Gebauer, G.: N2O concentration and isotope signature along profiles provide deeper insight into the fate of N2O in soils, Isotopes in Environmental and Health Studies, 44 (4), 377–391, 2008.

Goldberg, S. D., Knorr, K.-H., Blodau, C., Lischeid, G., Gebauer, G.: Impact of altering water table height of an acidic fen on N2O and NO fluxes and soil concentrations, Global Change Biology, 16 (1), 220–233, 2010.

Hornibrook, E. R. C.: The stable carbon isotope composition of methane produced and emitted from northern peatlands. Andrew J. Baird, Lisa R. Belyea, Xavier Comas, A. S. Reeve und Lee D. Slater (Ed.): Carbon Cycling in Northern Peatlands, Bd.184. Washington, D. C.: American Geophysical Union (Geophysical Monograph Series), pp. 187–203, 2009.

Hornibrook, E. R. C., Longstaffe, F. J., William, F. S.: Evolution of stable carbon isotope compositions for methane and carbon dioxide in freshwater wetlands and other anaerobic environments, Geochimica et Cosmochimica Acta, 64 (6), 1013–1027, 2000.

Hornibrook, E. R. C., Longstaffe, F. J., William, F. S.: Spatial distribution of microbial methane production pathways in temperate zone wetland soils: Stable carbon and hydrogen isotope evidence, Geochimica et Cosmochimica Acta, 61 (4), 745–753, 1997.

Kamman, C., Grünhage, L., Jäger, H.-J.: A new sampling technique to monitor concentrations of CH4, N2O and CO2 in air at well-defined depths in soils with varied water potential, European Journal of Soil Science, 52, 297–303, 2001.

Kim, Y., Ullah, S., Roulet, N. T., Moore, T. R.: Effect of inundation, oxygen and temperature on carbon mineralization in boreal ecosystems, Science of the Total Environment, 511, 381–392, 2015.

Knorr, K.-H., Glaser, B., Blodau, C.: Fluxes and 13C isotopic composition of dissolved carbon and pathways of methanogenesis in a fen soil exposed to experimental drought, Biogeosciences Discussions, European Geosciences Union, 5 (2), 1319–1360, 2008.

Kotsyurbenko, O. R., Chin, K.-J., Glagolev, M. V., Stubner, S., Simankova, M. V., Nozhevnikova, A. N., Conrad, R.: Acetoclastic and hydrogenotrophic methane production and methanogenic populations in an acidic West-Siberian peat bog, Environmental Microbiology, 6 (11), 1159–1173, doi:10.1111/j.1462-2920.2004.00634.x, 2004.

Novak, M., Gebauer, G., Thoma, M., Curik, J., Stepanova, M., Jackova, I., Buzek, F., Barta, J., Santruckova, H., Fottova, D., Kubena, A. A.: Denitrification at two nitrogen polluted, ombrotrophic Sphagnum bogs in Central Europe: Insights from porewater N2O-isotope profiles, Soil Biology and Biochemistry, 81, 48–57, 2015.

Robroek, B. J. M., Jassey, V. E. J., Kox, M. A. R., Berendsen, R. L., Mills, R. T.E., Cecillon, L., Puissant, J., Meima-Franke, M., Bakker, P. A. H. M., Bodelier, P. L. E.: Peatland vascular plant functional types affect methane dynamics by altering microbial community structure, Journal of Ecology, 103, 925–934, 2015.

Steinmann, P., Eilrich, B., Leuenberger, M., Burns, S. J.: Stable carbon isotope composition and concentrations of CO2 and CH4 in the deep catotelm of a peat bog, Geochimica et Cosmochimica Acta, 72 (24), 6015–6026, doi:10.1016/j.gca.2008.09.024, 2008.

Whiticar, M. J, Faber, E., Schoell, M.: Biogenic methane formation in marine and freshwater environments. CO2 reduction vs. acetate fermentation—Isotope evidence, Geochimica et Cosmochimica Acta, 50 (5), 693–709, doi:10.1016/0016-7037(86)90346-7, 1986.

Zou, Y., Hirono, Y., Yanai, Y., Hattori, S., Toyoda, S., Yoshida, N.: Isotopomer analysis of nitrous oxide accumulated in soil cultivated with tea (Camellia sinensis) in Shizuoka, central Japan, Sol Biology and Biochemistry, 77, 276–291, 2014.

[Figure]

[Figure]

Fig. 1. (new Figure 3): Development of (a) Twater [° C], (b) wtd [cm], (c) CH4 fluxes [g CH4 m-2 d-1] and (d) – (f) CO2 fluxes (NEP partitioned into Reco and GPP) [g CO2 m-2 d-1], ± 1 SD (n=6) ...

[Figure]

**Fig. 2.** (new Figure 4): Development of mean CH4 and mean DIC concentrations [$\mu$mol L-1], in hollows of the sites 1–4 from April 1st, 2014 through September 22nd, 2015.

[Figure]

**Fig. 3.** (not to be included into the manuscript): $\alpha C$ values obtained from silicon samplers in 5, 15, 25 and 35 cm depth at sites 1 to 4 from June to September. squares = June (06/11), ...

[Figure]

**Fig. 4.** (not to be included into the manuscript): A comparison of our eddy covariance vs chamber measurements for the NEE (left) and CH4 flux (right). The gray lines indicate the 1:1 lines. A good agreement..

[Figure]

**Fig. 5.** (manuscript Figure 6, panel e): dominant flux pathway of CH4 according to (Hornibrook, 2009). Empty circles = site 1, circles with diagonal lines = site 2, circles with crosses = site 3, ...

---

## Author Comment (AC2) · 4 Sep 2017

Dear Reviewer,

Thank you very much for your thoughtful review of our manuscript. We very much appreciate your time your time and your valuable ideas shared to improve our manuscript. Please find our answers to your comments below. As some of your concerns were also raised by reviewer 1, we kindly ask you to read our response to reviewer 1 too.

* The manuscript by Berger and co-authors is very interesting because the authors provide a comprehensive dataset on how soil carbon cycling changes along a transect

of four study sites (from undisturbed to disturbed conditions) in a peatland complex in Ontario from April 2014 until September 2015. They used a variety of methods that complement each other in space and time (e.g. chamber flux measurements of CO2 and CH4, DIC and CH4 concentration measurements at different soil depths, stable isotope measurements of CO2 and CH4, FTIR analysis of organic matter and porewater and measurements of ancillary variables such as air and water temperature, photosynthetically active radiation and water table depth below surface). The authors raise the major question, how peatland carbon fluxes respond to anthropogenically changed hydrological conditions and long-term nutrient-infiltration effects. Their major answer is that plant functional type may be a key variable to predict how soil carbon cycling in peatlands will respond to future nutrient inputs and changes in hydrology. Shrub dominated disturbed peatlands may turn into carbon sources, while graminoid-moss dominated peatlands "may maintain the peatland's carbon storage function". However, I have few major concerns but after a thorough revision and/or modification of the manuscript it would be great to see this manuscript published in the Biogeoscience journal.

* Major comments:

The authors point out that it is not new that plant functional types may have a strong influence on ecosystem soil carbon dynamics but I completely agree with the authors that "there is a gap of knowledge in terms of interactions between peat and plants under IN-SITU CONDITIONS". This makes this manuscript very valuable. However, I am not an author of the paper "Peatland vascular plant functional types affect methane dynamics by altering microbial community structure. (Robroek et al. 2015, doi: 10.1111/1365-2745.12413)" but the authors of this manuscript should cite that paper and compare both results. Robroek et al. (2015) nicely demonstrate that resilience of peatland CH4 dynamics, and therefore also CO2 dynamics, to climate change may depend on interaction between microbes and plant functional types.

– Okay. Will be done.

* I think the manuscript would greatly benefit from a more thorough discussion about the potential role of methanogens driving soil methane dynamics at the four different sites. In the current study, the authors measured stable carbon isotope ratios of $CH_4$ and $CO_2$ comprehensively. Hence, apparent fractionation factors could be easily measured (Angel et al. 2011; doi:10.1371/journal.pone.0020453 or McCalley et al. 2014; doi:10.1038/nature13798), the different pathways of methanogenesis identified and discussed. Now, the authors attribute the change of isotopic signals to changes in methane oxidation. This is very speculative and not sufficient. The change in $13CH_4$ may result from a shift from hydrogenotrophic to aceitclastic methanogenesis, especially during drier months (see Hodgkins et al. 2014; www.pnas.org/cgi/doi/10.1073/pnas.1314641111 2014 or McCalley et al. 2014; ; doi:10.1038/nature13798). However, this should be discussed in the manuscript.

– We understand yours and reviewer 1's concern. Of course, the dominant $CH_4$ production and transport pathway cause differences in isotopic signatures. It is not that we missed those pathways and obtaining fractionation factors but we decided to exclude it from our manuscript because we think distinguishing methane production pathways with our data would be critical. Please see our response to Reviewer 1 for a thorough explanation of this point.

* The authors state that it is clearly visible that ratios of C/N, C/Mg and C/K in peat soil are decreasing from site 1 to site 4. I do not see this pattern when I look at Table 1. C/K is higher at site 2 and 3 than at site 1. C/Mg is lowest at site 1. I guess C/N ratios do not differ between site 1, 2 and 3. Furthermore, I guess there are no significant differences in C/P ratios between the different sites. N/P ratios are higher at site 2 than site 3 and C/Ca ratios are lowest at site 2. Please, check your data.

– You are right. This table was supposed to be a short summary of the previous study, which is now published in Soil Biology & Biochemistry (114 (2017) 131-144 http://dx.doi.org/10.1016/j.soilbio.2017.07.011). We agree that this table, providing only this little information is not convincing, as the results from the SBB paper are

quite complex. We are going to remove the stochiometric ratios from table 1 and to provide a better description of the most important results from the SBB paper, providing more convincing evidence that those sites in closer vicinity to the reservoir are more affected.

* However, it would be great to have a look at the submitted publication or if the authors would incorporate more convincing information. Otherwise the authors cannot state that "it becomes evident that the peatland was exposed to nutrient infiltration form the water reservoir and thus elevated nutrient concentrations occurred in vicinity to the water reservoir." (P13, L2-L6) and should reformulate the whole discussion section!

– We would be pleased if you were to have a look at the paper (SBB 114 (2017) 131-144). Here, the observed differences in the transect are elaborated. Due to the complexity of the dataset only a short summary can be provided in the present manuscript. We will revise our text to improve clarity and refer to the now published study. We apologize that the manuscript had not been available at the time of submitting this paper.

* Specific comments:

Titel: Currently, I do not see that nutrients drive carbon dynamics at your sites.

– We came up with a new title: "Differential response of carbon cycling in a continental Canadian peatland to long-term nutrient input and altered hydrological conditions"

* Abstract: If you mention the other methods in the abstract, you should mention FTIR analysis as well.

– Will be done.

* P1, L17-L19: All the sites are characterized by wet conditions. These are peat soils.

– What we were trying to say was that the site 3 hollows experienced higher water levels. We will make our point clearer.

* P1, L19-L20: Low 13CH4 may be caused by more hydrogenotrophically produced

CH4.

* P1, L24: or more aceticlastically produced CH4. More labile organic matter may favor aceticlastic methanogenesis.

– Yes, we agree, but given the unsaturated conditions and strong water table drop downs during the summer and as explained further above, we think that it would be critical to distinguish methane production pathways; an influence of methanotrophic conditions is much more likely. Indeed, we think that currently not many studies have presented results from around the unsaturated zone and thus most existing studies focused on discussion pathways at greater, saturated depths.

* P3, L8: I do not see a gradient in nutrient availability.

– That is probably because the provided data here was in its current presentation not convincing. The table presents only a small fraction of the data provided in the SBB paper, however, the line of argument of the SBB paper is based on a greater data set (peat ages and accumulation rates, depth profiles of element concentrations, stochiometric ratios of surface peat, $\delta$15N-values and C/N ratios of the vegetation as well as composition of the vegetation) in order to properly describe the impact of the water reservoir on the study area and its sites. When considering only single factors (e.g. only stochiometric ratios of surface peat), their explanatory power decreases; it is the entirety of factors which shapes our knowledge of the study area. We are sorry for taking the stochiometric ratios out of context. We are going to remove the stochiometric ratios from the table and we will provide a more convincing description of the study sites in terms of nutrient availability in the methods section while referring the reader to the SBB manuscript. After removing the stochiometric ratios from table 1 we will add more information on the vegetational gradient so that differences between sites become more obvious.

* P3, L10: Please, calculate apparent fractionation factors for methanogenic pathways.

– Please see our answer above. We have calculated such fractionation factors for a previous version of the manuscript, but eventually removed the figure and the related explanations and story because we were very skeptical of the results. Please let us know if you agree with our decision. We will seriously consider yours and reviewer 1's suggestions.

* P3, L16: I do not see that nutrient inputs are greatest in peatland periphery (see Table 1).

– The nutrient data from table 1 is going to be removed. Instead the sites will be described better and the reader is going to be referred to the SBB paper which has more convincing evidence of greatest nutrient inputs to the peatland periphery.

* P3, L 18: Why should CH4 emissions be greatest at the graminoid dominated sites? There is no link to this hypothesis in the introduction.

– A better explanation will be provided and this point will be introduced in the new hypothesis 2.

* P3, L19-21: You should also discuss CH4 production pathways.

– As we removed the fractionation factors and everything related to the different CH4 production pathways from the previous manuscript version, we also removed the related explanations from the introduction. But since you and the other reviewer came up with concerns, we agree that we should add a paragraph to our discussion which explains about the different pathways, which also explains why we are not distinguishing CH4 production pathways with our data.

* P4, L22-28: This paragraph is very essential for the main message of the manuscript. Unfortunately, the data do not support these statements. It would be great to see more data that support these ideas.

– We agree, the table and the statements are not convincing here. The reader will be referred to the SBB paper and a more convincing description of the sites will be added.

\* P5, L8: Please, write out FTIR analysis.

– Will be done.

\* P5, L8-L10: I am not familiar with FTIR analysis. "For pore-water samples 2 mg of oven-dried organic matter. . ." is that correct?

– Yes, it is correct, but maybe the sentence is a bit misleading. The pore water samples were dried in an oven until all the water was gone. Then the remaining solid material was scratched off the bottom of the sample containers and underwent FTIR analysis. We will provide a better description.

\* RESULTS section: The presentation of the results are too cluttered. Results that are not significant are described quite often (see P10, L7, L10) and sometimes it is not clear if results are significant or not (see P10, L16-17, L19-26). It would be better to mark significant differences in the figures and to highlight significant results or only few non-significant results in text, if they really enrich and/or support the guiding questions in the manuscript.

– We will try to compromise between you and reviewer 1 in this respect. We will improve the presentation of the results, focus on significant results and provide less space for results that were not significant, but we will also try to include results from the sites 1 and 2 as reference sites into our discussion.

\* P13, L15-L16: This is repetition of results.

– The sentence will be rewritten.

\* P13, L19-L21: This is very speculative. Did you check FTIR ratios of inflowing water? Maybe you can provide some references.

– We agree. Unfortunately, we did not sample the inflowing water. Reviewer 1 was also concerned about this statement and provided an alternative explanation (higher productivity of the graminoid vegetation → higher input of labile compounds), which we
will include into the discussion. By doing so we can remove the present statement.

* P13, L22-L23: I am not convinced by this statement. The difference between site 4 and the three other sites may be simply by chance.

– We understand your concern. You are right, the observed effects can have different causes. However, looking at the entire transect the peat quality found at the sites suggests a quite similar history before dam construction at the site. So, based on our analysis the factor we can identify is the enrichment in nutrients and the concomitantly altered vegetation. Of course, site 4 thus had longest exposure to more minerotrophic conditions from intrusion of lake water. However, we would also consider this effect as anthropogenic then. Our results are derived from an in-situ study; of course, experimental set-ups under controlled (laboratory) conditions provide more explicit results and such results can be more reliably related to certain factors. As compared to such studies, in-situ studies have short-comings indeed; however, in-situ studies are needed to verify concepts based on such controlled conditions and we think that we have taken all necessary pre-cautions to avoid misinterpretations and to not over-interpret our data. As mentioned above, when revising our manuscript, we will provide more convincing information in terms of differences between our sites, so that you will hopefully agree with us in terms of the significance of results derived from those sites.

* P13, L24-L27: So, it is not the vicinity to the reservoir but the vegetation that drives carbon cycling processes?

– We are not sure if we can provide a final answer to that question with our data. We think we are dealing with a complex interplay between vegetation, microorganisms and location factors. Site 4 and site 3 appeared to have received a similarly high amount of nutrients (well, site 4 probably received a bit more); around site 4 a dense Myrica belt established while at site 3 graminoids established. (The SBB paper provides information on vegetation etc.) With Myrica being present at the site, site 4 developed in a different way than site 3, where graminoids are established. So, we think it is both, the

vicinity to the reservoir and the vegetation community that drives carbon cycling. By our study we thus also want to support that the response of a peatland to nutrient input and altered hydrological conditions may not be as simple as identified in studies with controlled variation of individual factors.

* P14, L11-L19: This is repetition of results.

– The paragraph will be rewritten.

* P14, L25: It would be great to see the data.

– Please see the SBB paper.

* P15, L1-L2: Site 4 shows second highest CH4 release. Then you cannot state that graminoid sites show highest CH4 emissions. I would emphasize to reformulate the introduction and the hypothesis in such a manner that it becomes clearer to the reader why you have stated your hypothesis. Now, the discussions seems to be much too blurred.

– Only the CH4 emission from site 3 is significantly increased; the CH4 emissions from the sites 1, 2 and 4 are not significantly different. But we agree, the sentence can be misleading and will be rewritten. Anyway, hypotheses 2 from the introduction will be rewritten as well as other parts of the introduction, taking into account your suggestions.

* P15, L6: What means "healthy" Sphagnum moss community?

– The mosses at site 4 showed severe signs of desiccation and thus inactivity in 2014 and 2015 during the summer months (June $\sim$ September) and recovered afterwards. (Reduced photosynthetic activity of Sphagnum mosses while facing severe drought was previously observed in several studies (e.g. Alm et al., 1999, Aurela et al., 2007)). Given the pitiful appearance of Sphagnum mosses during the summers at site 4, and given also that Sphagnum covers only 60 % of the site 4 area, we concluded that Sphagnum was in retreat at site 4. In contrast, at the sites 1, 2 and 3 the Sphagnum

mosses looked green or red (whatever their natural color was) and always moist, which we then termed a "healthy" appearance of Sphagnum mosses. We apologize for utilizing the colloquial expression "healthy". We will work in a more appropriate description of the vegetation of our study sites into our manuscript while revising it, to avoid any misunderstandings.

\* P15, L22 – P16, L33: see major comments.

\* P17, L2 – L19: see major comments.

– Please see our answers above.

\* Figures: It would be great to have figures with higher resolution. In Figure 6, I can hardly identify the difference between the circles.

– Figure 6 will be provided in a higher resolution.

\* Table1: Please, mark significant differences.

– Stochiometric ratios from table 1 will be removed as explained above.

References:

Alm, J., Schulman, L., Walden, J., Nyknen, H., Martikainen, P. J., Silvola, J.: Carbon balance of a boreal bog during a year with an exceptionally dry summer; Ecohydrology, 4, 733–743, 1999.

Aurela, M., Riutta, T., Laurila, T., Tuovinen, J.-P., Vesala, T., Tuittila, E.-S., Rinne, J., Haapanala, S., Laine, J.: CO2 exchange of a sedge fen in southern Finland – the impact of a drought period, Tellus B, 59, 826–837, 2007.

Berger, S., Gebauer, G., Blodau, C., Knorr, K.-H.: Peatlands in a eutrophic world – Assessing the state of a poor fen-bog transition in southern Ontario, Canada, after long term nutrient input and altered hydrological conditions, Soil Biology and Biochemistry, 114, 131–144, 2017.

---

## Author Response (AR1)

**Dear Reviewer 1,**

we very much appreciate your thoughtful review of our manuscript. Thank you for your time and your valuable ideas. We absolutely agree that taking up your suggestions greatly improved our manuscript and we did so as described below.

\* This manuscript reports an interesting dataset on C cycling at a temperate peatland, affected by increased nutrient input from a nearby reservoir. Carbon dioxide and methane fluxes were measured over a period of 1.5 years from four sites representing variable wetness, vegetation type and distance from the reservoir, and the flux measurements were accompanied by detailed soil profile measurements of CH4 and DIC concentration. Carbon stable isotopes were used in order to gain more information about CH4 production, oxidation and transport. The paper is well written and logically structured, and the appearance of the figures C1 BGD Interactive comment Printer-friendly version Discussion paper is very good. The methods are described in great detail, and the authors are clearly experts in selection and implementation of their field and analytical methods. The value of this work is in the high quality and completeness of the paper could be greatly improved, by taking the following comments into consideration.

**\*\*\*Major comments**

1. The match between the content of the manuscript and the title is not ideal at the moment. The title and especially the starting sentence of the abstract make one expect a comparison of carbon cycling between anthropogenically altered vs. natural sites. If this is the focus, it would be important to describe the transect better in the abstract and also in the methods section (page 4, lines 2-3 & from lines 16 onwards): How much did the hydrological condition change along the transect, and was the human impact related to drying or wetting or to fluctuating water table? And, in the abstract, how much did the nutrient infiltration change along the transect (data in Table 1)?

**– We understand your concern.**

This present study is following up on a study, which was very recently published in SBB (114 (2017) 131-144; http://dx.doi.org/10.1016/j.soilbio.2017.07.011). While writing the current paper, this other study had not been published so we could not well refer to it. The study site was described in detail in this SBB-paper. As elaborated there, the water reservoir affects the entire northern tip of our peatland site. We find in our data that areas further away from the reservoir are likely less affected, but it is not entirely possible to distinguish 'natural' from 'anthropogenically altered' sites among the sites 1-4 (800-200 meters distance from the reservoir). Only areas further away from the reservoir could likely be regarded as pristine. So, our transect of study sites as presented in this present study rather assesses 'strongly altered' sites vs. 'less altered' sites with respect to the investigated features. Unfortunately, this entire gradient was not as obvious in the beginning of the study, so this part is indeed missing. So, our objective for this paper was to investigate carbon cycling along this transect and to identify effects of altered conditions along the transect on carbon cycling and fluxes.

In the revised version of our manuscript we described the transect better in the abstract (page 1, lines 14-17) and in the method section (page 4, line 31 to page 6, line 2), providing the information you requested.

With respect to the hydrological impact, particular information was added (page 4, line 31 to page 5, line 3): "Through flooding of the reservoir, Wylde Lake peatland has been exposed to altered hydrological conditions in a way that the water reservoir enhanced water level fluctuations in a large part of the site: in summer or under dry conditions, water is released from the reservoir, thereby draining water out of the peatland; under wet conditions, water table levels of the reservoir increase and water is pushed into the peatland. Those sites in closer vicinity to the reservoir are presumably more affected than sites further away from the reservoir (Berger et al., 2017)".

The title was changed to: "Differential response of carbon cycling to long-term nutrient input and altered hydrological conditions in a continental Canadian peatland". We believe the new title better reflects the contents of our manuscript.

\* Further, instead of reporting just the results from the two highly affected sites 3&4 in the abstract, you should compare the results between anthropogenically altered vs. natural sites. This would justify the last sentence which claims clear anthropogenic effects on C cycling.

We agree. The abstract was rewritten, however, still the focus is on the results from the sites 3 and
 This is because the two sites were mostly altered and here we also found most significant differences. The last sentence of the previous abstract was deleted.

\* This comparison between affected and unaffected sites should be the view-point throughout the MS. For example:

By rearranging Fig. 3 & 4 so that instead of ' showing various parameters from the same site in the same figure you would show a single parameter from all of the sites in the same figure.
By adding here some ' indication of the reservoir effect: Distance of the reservoir in the table itself, or descriptive sentence in the table caption.

- By focus the introduction better from general ' description of factors affecting in peatland C cycling towards a description of the effects of anthropogenic activities on it. It should be stated clearly, citing the relevant literature, why is it important to understand effects of increased nutrient inputs and changed water level on carbon cycling. This topic it touched on the paragraph starting on page 2, line 24, but also there anthropogenic effects are not sufficiently discussed to match with the title of the paper. Also, the motivation statement on page 3, lines 4-8 is very general. Could you develop a more specific research question that suits for this particular study?

- By rewriting the Concluding remarks section to answer the questions 'posed by the title and the introduction section.

Unfortunately, the manuscript cannot provide a comparison between "affected" and "unaffected" sites (see also above). However, it can provide a comparison of carbon cycling of *strongly altered* and *less altered* sites. We have clarified this point to avoid misunderstandings here.
 Please see:

-page 1, lines 16-17 -page 5, line 8 to page 6 line 2 -page 14, lines 16-17

We re-arranged figures 3 & 4 as suggested. We agree that this very much improves clarity. The development of pore gas DIC and  $CH_4$  concentrations over time could, however, not be re-arranged as the 3D-plots would get too complicated.

Having re-arranged the figures as you suggested, differences between the sites in terms of  $CO_2$  and  $CH_4$  fluxes become indeed much more obvious.

Table 1 in its previous form was thought to summarize some results from the SBB-paper, but it turned out that presenting only that little information about the sites was quite misleading. Peat ages, pH and stochiometric ratios from table 1 were removed (as the information can be found in the SBB paper), instead, information on plant species abundances at each site were added in order to better illustrate the vegetation gradient. Moreover, the reader was oftentimes referred to the SBB-paper.

The introduction was rewritten (page 2, line 17 to page 4, line 16) according to your suggestion. The paragraph (starting on page 2, line 24 of the previous manuscript version) dealing with factors affecting peatland C cycling was extended towards a description of the effects of anthropogenic activities. In this regard, results from previous long-term peatland fertilization experiments (e.g. Mer Bleue, Whim Bog, Degerö Stormyr) as well as impacts of inundation on neighboring peatland ecosystems (e.g. Kim

et al., 2015; Ballantyne et al., 2014) were better summarized and we also provided a more adequate summary of what is known about gaseous carbon fluxes in relation to an altered plant community (e.g. Robroek et al., 2015) to set the picture for our study. Taking also into account our own recently published paper (Berger et al., 2017) it is now much more obvious why there still is a need to study changes in peatland C-cycling after increased nutrient inputs and changed water levels.

As far as a more specific research question is concerned, we developed new hypotheses which better meet the focus of our study (page 4, lines 17-21):

1) hydrologically altered and nutrient enriched peripheral sites feature accelerated C cycling, reflected in more decomposed peat,

2) increased abundance of vascular plants can increase  $CO_2$  uptake but also change patterns of  $CH_4$  production and emission, in particular if graminoids dominate, and

3) long-term nutrient enrichment in combination with hydrologically altered conditions may therefore cause differential responses of carbon cycling and does not necessarily cause a loss of the C-sink function of peatland ecosystems.

The concluding remarks section was adjusted accordingly. Please see page 19, line 18 to page 20, line 6.

\* As you state in the discussion section (page 14, lines 13-15), it is hard if not impossible to separate the wetness effects from the nutrient infiltration effects. Thus, to draw any conclusions about anthropogenic effects on C cycling, it should be carefully considered how the data presentation is organized to serve that purpose.

– We agree. Please see our explanations above on how we improved our manuscript with respect to this aspect. Moreover, the discussion was in large parts completely rewritten.

\* 2. The MS includes interesting isotopic data of the CH4 emission and porewater DIC and CH4. A better explanation of how the stable isotopic data can be interpreted would be very much needed already in the introduction section.

– A better explanation was added, which covers the interpretation regarding underlying pathways and methanotrophic activity. Please see page 2, lines 17-32.

\* In the discussion section (page 15, line 24) you mention that the isotopic signature in methane is affected by methane production, oxidation and transport but you do not explain anywhere why and how the isotopic composition is affected by these processes.

– Such explanations were added to the introduction (page 2, lines 23-25; lines 27-32) and to the discussion (page 17, lines 7-22; page 18, lines 33-34).

Further, the discussion of isotopic data is related to methane oxidation. Could the dominant methane production pathway (acetoclastic, hydrogenotrophic) or transport pathway have caused differences in isotopic signatures and how? At the moment, the discussion on isotopic signatures is related to methane oxidation only.

– We understand your concern. Of course, the dominant CH4 production and transport pathway cause differences in isotopic signatures. This aspect is now included in the revised version of our manuscript. According to Whiticar et al. (1986) isotope fractionation factors  $\alpha_c$  were calculated for depths below 35 cm as for such depths we could assume water-saturated, anoxic conditions. We think distinguishing

methane production pathways for the upper depths with our data would be critical. That is why we would prefer to refrain from it.
Explanations were added:
-methods: page 10, lines 20-23
-results: page 13, line 29 to page 14, line 3
-discussion: page 17, lines 7-22; page 18, lines 13-29; page 19, lines 28-32.

\* 3. In many occasions, you refer often to your own, yet unpublished work (Berger et al., submitted). Since that work seems to contain information quite crucial for the present paper, it is somewhat problematic that the paper is not available for the reader. If the submitted paper has not been published meanwhile, you should consider elaborate those results in more detail when necessary, e.g., in the methods section, page 4, line 2-3 about the hydrological changes caused by the reservoirs and page 4, line 23.

The paper is published now in Soil Biology & Biochemistry 114 (2017) 131-144
 http://dx.doi.org/10.1016/j.soilbio.2017.07.011
 We apologize that this information could not be presented earlier.

\*\*\*Minor comments

\* page 1, lines 18-19 & page 11, lines 19-21: The study period includes a full year of measurements. It would be good to give values also on cumulative annual fluxes. This was enable using this carefully collected data in flux syntheses, and facilitate comparison with literature values.

Done. Please see:
-page 1, line 23, 24
-page 10, lines 9-12; lines 22-24
-page 15, line 26 to page 16, line 4

\* page 2, lines 3-5: One-sentence paragraphs should be avoided. I suggest combining this sentence with the next paragraph. See also page 4, line 29; and page 17, lines 16-19.

- Done. Please see: page 2, lines 9-11; page 5, lines 6-7; page 19, lines 29-31.

\* page 2, lines 26-28: Reference missing.

- The respective sentence was removed.

\* page 3, lines 12-15 & page 4, line 2: Also here, I would like to see a mentioning about how the hydrology is altered – drying or wetting, or more variable in the course of the year?

– Information was added. Please see page 4, line 31 to page 5, line 3.

\* page 4, line 13: microtopography is a single word

- This sentence was removed.

\* page 4, lines 18-19: Listing the sites starting from number 4 is counterintuitive. Would it be possible to change the order in which you mention the sites, or simply change the numbering? You indicate that site 2 was further away from the reservoir than site 3, but it would be better to describe the whole transect, e.g., that the distance from reservoir decreases with growing number.

– We are sorry, but we don't think that would be possible. As the site names were established in the SBB paper, changing the names now would be a bit confusing...

\* page 4, lines 25-28: It is not clear if and how this is related to the vicinity of reservoir?

- You are right. This sentence was removed.

\* page 5, line 8: At the first appearance, write the complete instrument type instead of the abbreviation FTIR.

– Done. Please see page 6, line 13.

\* page 5, line 18: Regarding UV-VIS, see the previous comment.

– Done. Please see page 6, lines 24-25.

\* page 6, line 12: Was the image analysis based on satellite/aerial or other imagery? Please specify!

- It was done via aerial imagery with images obtained from UAV flights. This information was added. Please see: page 7, lines 15-16.

\* page 8, lines 5-6: Have you tested if there were any discrimination against the lighter isotope during diffusion into the silicon collectors?

- We are very sorry for confusing "silicone" (in German: "Silikon") with "silicon". This funny mistake was removed from the manuscript. Moreover, explanations on equilibrium times and fractionation at the silicone membrane were added. Please see page 9, lines 11-13.

\* page 8, lines 13: Please check the sentence (some words missing/in a wrong order).

– Done. Please see page 9, lines 7-10.

"Silicone tubes for isotope sampling had an inner diameter of 1 or 0.5 cm, corresponding to a volume of 20 or 5 ml. The samplers with a volume of 20 ml were installed in 5 cm depth and the smaller samplers below, as close to surface larger volumes of samples were necessary in order to obtain sufficiently high concentrations (2.5 < x < 2000 ppm) for isotope analysis."

\* page 9, lines 29-31: Please check the sentence (some words missing/in a wrong order).

- Done. Please see page 10, line 34 to page 11, line 2.

"Means were compared with t-Tests (if data was normally distributed) respectively Kruskal-Wallis and post hoc Wilcoxon-Mann-Whitney-Test (if data was not normally distributed). The confidence level for the statistical tests was  $\alpha = 0.05$ ."

\* page 11, lines 22-24: These are important results for this paper. But, can you really say that it is anthropogenic effect, or just a consequence of different location (edge effect, more mineral site?). It's interesting that the site receiving more nutrients is showing lower CO2 uptake.

- We agree that the observed effects can have different causes. However, looking at the entire transect the peat quality found at the sites suggests a quite similar history before dam construction at the site. So, based on our analysis the factor we can identify is the enrichment in nutrients and the concomitantly altered vegetation. Of course, site 4 thus had longest exposure to more minerotrophic conditions from intrusion of lake water. However, we would also consider this effect as anthropogenic then.

We provided statements in this regard in our methods (page 5, line 33 to page 6, line 2) and discussion (page 14, lines 18-22) sections.

\* page 13, lines 19-21: Or, is the higher lability of organic matter caused by higher productivity and high input of labile compounds from vegetation? This site showed the highest C accumulation (page 11, lines 19-21). If it is a reservoir effect, should not the organic matter at site 4 be even more labile? Now, the site 4 was showing the highest proportion of aromatic compounds.

– That is a very interesting question! You are right, it could very well be the case that the higher lability of organic matter is caused by the vegetation (higher productivity  $\rightarrow$  higher input of labile compounds). That idea is now included in the discussion. (Please see page 15, lines 13-19.) We first came to our conclusions as DOM is usually a small but easily accessible pool and should thus reflect a residual enrichment of refractory compounds. Presence of labile DOM we thus attributed to external input. Indeed, this would then also apply to site 4. This discussion was not straightforward and was now modified considering your suggestion.

\* page 13, lines 19-21: Please add references: In recent studies by Bragazza et al. ...

- Sentence was removed.

\* page 14, line 21: Decrease in the CO2-sink strength in response to what?

- Sentence was removed.

\* page 16, lines 3-4: You write about "deepening of soil oxygenation probably promoting a highly active methanotrophic bacteria community, which drew CH4 from the atmosphere down to that depth". Why do you think it was atmospheric and not peat-derived CH4 that was oxidized at 15 cm? Atmospheric methane cannot diffuse to the soil against the concentration gradient (when the porewater concentrations are above ambient).

– You are right. That is speculative. We thought it probably was atmospheric CH4, which entered the soil because the CO2 and CH4 concentrations in the upper peat layers were particularly low on that day. The water table even was below -15 cm depth. Also, the surface peat looked quite dry during that summer period and the *Sphagnum* mosses were white and inactive (later in the year they recovered). So, our idea was that the CH4 in 5 cm depth was of atmospheric origin. But we had indeed no proof for this interpretation. Therefore, we adjusted our discussion here. Please see page 17, line 31 to page 18, line 1.

\* page 16, line 5: Why enriched signals would mean low CH4 production? Do you mean more CH4 production via the acetoclastic pathway that results in heavier methane than the hydrogenotrophic pathway?

– Actually, we were trying to avoid speculations about acetoclastic and hydrogenotrophic pathways for depths above -35 cm, because values of  $\alpha_C$  typically observed for the acetoclastic pathway could also arise from methanotrophic activity. Both processes would yield an enrichment in 13C of the residual CH4. We believe that it is rather CH4 oxidation, which took place because of the low water tables and the unsaturated conditions during the summer. So, for depth above -35 cm, we boiled it down to: methane production yields less enriched 13C - CH4, while CH4 oxidation would leave behind comparatively more enriched CH4. We then interpreted more oxidation as less net production. Only for depths below -35 cm we distinguished methanogenic pathways.

Therefore, section 4.3 was completely rewritten. Please see page 17, line 6 to page 19, line 16.

\* page 16, lines 20-21: Besides transporting CH4 through the aerobic peat layers without exposing it to oxygen plant-mediated transport also strongly discriminates against the heavier methane (Chanton, 2005). Because of this, plant transport can create even lighter methane that is present anywhere in the peatland. It would be important to mention this in the discussion about isotopic signatures.

– Done. Please see page 18, lines 33-34.

\* page 16, lines 23-28: Yes, probably most of the methane is oxidized during the diffusion, and thus, the amount of methane reaching the atmosphere by diffusion is low. So even with low coverage of aerenchymous plants, most of the methane that is actually entering the atmosphere is emitted through them.

– That was our idea, too.

\* page 16, lines 28-30: It seems to me that you have done all the necessary pre-cautions to avoid methodological biases in the data. Please specify, what actually makes you suspect some methodological problems particularly in low water table conditions.

– For gas sampling for later isotope abundance analyses we used the same chambers as used for the flux measurements. It is known that chamber measurements tend to overestimate CH4 fluxes a bit for several reasons (spatial heterogeneity, artificial pressure fluctuations induced by the chambers...). So, artefacts cannot be fully excluded. To counteract possible concerns in terms of data quality we verified our chamber flux data with eddy covariance flux data (see the figure below). The sets of data are nicely comparable, however, CH4 fluxes measured with chambers were slightly increased in July and August when the water tables were lowest, which we think could have something to do with deeper CH4 pools becoming connected to the atmosphere under unsaturated conditions with dropping water tables as explained in Estop-Aragones et al. (2016). With chamber induced pressure fluctuations such CH4 pools might have been forced out of the peat.

We would like to clarify that in the course of quality assurance/ quality checks while processing data, most likely all low-quality data was eliminated. Thus, we are sure that our data provided (fluxes and isotope data) is of very high quality. By mentioning about the issue in the first place we intended to point out the common shortcomings of the method; instead it probably downgraded our results, so we removed the related sentences from the manuscript to avoid any misunderstandings.

A comparison of our eddy covariance vs. chamber measurements for the NEE (left) and CH4 flux (right). The gray lines indicate the 1:1 lines. A good agreement is apparent; however, the chamber measurements detected higher CH4 emissions in July and August (circled in gray).

\* page 17, line 9: Instead of just saying results, it would be better to specify which particular parameter you mean here.

- The concluding remarks section was rewritten and the confusing statement was removed.

\* page 17, lines 16-19: Long and complicated sentence, please consider splitting it into two sentences.

- Done. Please see page 19, lines 28-32.

\* Fig. 1: For better clarity, please mark the Luther lake reservoir in the figure.

– Done.

\* Fig 6 (and page 12, lines 28-29). In this figure, you have decided to use the porewater d13C-CH4 at 5 cm. However, the methane pool at this depth does not necessarily represent very well the origin of the methane emissions, since the ebullitive and plantmediated fluxes are originating from deeper layers. Hornibrook (2009) was using the average from 0 to 50 cm. It would be interesting to see how the figure looks if the porewater methane at depth is not excluded. Although the differences between different depths were not significant, Fig. 5 shows that especially at sites 1 and 2, the porewater CH4 has different isotopic composition at 5 cm than at deeper depths.

– We redrew the panel (e) of figure 6 using also  $\delta^{13}$ C-CH4 -values from the deeper CH4. (please see Fig. 6 in the revised manuscript.

\* Fig 6. caption, line 5: A typo in the word 'circles'.

- Was corrected.

\* Table 1: The differences in stoichiometry are not evident, and I do not see a clear transect here. Could you test the differences statistically and mark it in the table? Or, would the amounts instead of ratios reveal the pattern more clearly?

- Thank you for pointing it out. We removed the stochiometric ratios from the table, added more information on the vegetation and described the transect in the text while referring the reader to the SBB paper for more information on nutrient supply to the study site.

**References**:**

Ballantyne, D. M., Hribljan, J. A., Pypker, T. G., Chimmer, R. A.: Long-term water table manipulations alter peatland gaseous carbon fluxes in Northern Michigan, Wetlands Ecology and Management, 22, 35–47, 2014.

Berger, S., Gebauer, G., Blodau, C., Knorr, K.-H.: Peatlands in a eutrophic world – Assessing the state of a poor fen-bog transition in southern Ontario, Canada, after long term nutrient input and altered hydrological conditions, Soil Biology and Biochemistry, 114, 131–144, 2017.

Estop-Aragones, C., Zajac, K., Blodau, C.: Effects of extreme experimental drought and rewetting on CO2 and CH4 exchange in mesocosms of 14 European peatlands with different nitrogen and sulfur deposition, Global Change Biology, 22 (6), 2285–2300, 2016.

Kim, Y., Ullah, S., Roulet, N. T., Moore, T. R.: Effect of inundation, oxygen and temperature on carbon mineralization in boreal ecosystems, Science of the Total Environment, 511, 381–392, 2015.

Robroek, B. J. M., Jassey, V. E. J., Kox, M. A. R., Berendsen, R. L., Mills, R. T.E., Cecillon, L., Puissant, J., Meima-Franke, M., Bakker, P. A. H. M., Bodelier, P. L. E.: Peatland vascular plant functional types affect methane dynamics by altering microbial community structure, Journal of Ecology, 103, 925–934, 2015.

Whiticar, M. J, Faber, E., Schoell, M.: Biogenic methane formation in marine and freshwater environments.  $CO_2$  reduction vs. acetate fermentation—Isotope evidence, Geochimica et Cosmochimica Acta, 50 (5), 693–709, doi:10.1016/0016-7037(86)90346-7, 1986.

Dear Reviewer 2,

Thank you very much for your thoughtful review of our manuscript. We very much appreciate your time your time and your valuable ideas shared to improve our manuscript. Please find our answers to your comments below. As some of your concerns were also raised by reviewer 1, we kindly ask you to also read our response to reviewer 1.

\* The manuscript by Berger and co-authors is very interesting because the authors provide a comprehensive dataset on how soil carbon cycling changes along a transect of four study sites (from undisturbed to disturbed conditions) in a peatland complex in Ontario from April 2014 until September 2015. They used a variety of methods that complement each other in space and time (e.g. chamber flux measurements of CO2 and CH4, DIC and CH4 concentration measurements at different soil depths, stable isotope measurements of CO2 and CH4, FTIR analysis of organic matter and porewater and measurements of ancillary variables such as air and water temperature, photosynthetically active radiation and water table depth below surface). The authors raise the major question, how peatland carbon fluxes respond to anthropogenically changed hydrological conditions and long-term nutrient-infiltration effects. Their major answer is that plant functional type may be a key variable to predict how soil carbon cycling in peatlands will respond to future nutrient inputs and changes in hydrology. Shrub dominated disturbed peatlands may turn into carbon sources, while

graminoid-moss dominated peatlands "may maintain the peatland's carbon storage function". However, I have few major concerns but after a thorough revision and/or modification of the manuscript it would be great to see this manuscript published in the Biogeoscience journal.

**\* Major comments:**

The authors point out that it is not new that plant functional types may have a strong influence on ecosystem soil carbon dynamics but I completely agree with the authors that "there is a gap of knowledge in terms of interactions between peat and plants under INSITU CONDITIONS". This makes this manuscript very valuable. However, I am not an author of the paper "Peatland vascular plant functional types affect methane dynamics by altering microbial community structure. (Robroek et al. 2015, doi: 10.1111/1365-2745.12413)" but the authors of this manuscript should cite that paper and compare both results. Robroek et al. (2015) nicely demonstrate that resilience of peatland CH4 dynamics, and therefore also CO2 dynamics, to climate change may depend on interaction between microbes and plant functional types.

- Done. Please see page 3, lines 23-25; page 14, lines 24-25; page 15, lines 13-14.

\* I think the manuscript would greatly benefit from a more thorough discussion about the potential role of methanogens driving soil methane dynamics at the four different sites. In the current study, the authors measured stable carbon isotope ratios of CH4 and CO2 comprehensively. Hence, apparent fractionation factors could be easily measured (Angel et al. 2011; doi:10.1371/journal.pone.0020453 or McCalley et al. 2014; doi:10.1038/nature13798), the different pathways of methanogenesis identified and discussed. Now, the authors attribute the change of isotopic signals to changes in methane oxidation. This is very speculative and not sufficient. The change in 13CH4 may result from a shift from hydrogenotrophic to aceitclastic methanogenesis, especially during drier months (see Hodgkins et al. 2014; www.pnas.org/cgi/doi/10.1073/pnas.1314641111 2014 or McCalley et al. 2014; doi:10.1038/nature13798). However, this should be discussed in the manuscript.

**- We understand yours and reviewer 1's concern.**

Of course, the dominant CH4 production and transport pathway cause differences in isotopic signatures. It is not that we forgot to obtain fractionation factors and to include those pathways in the previous version of the manuscript but we had decided to exclude the issue because we thought and still think distinguishing methane production pathways with our data would be critical. Methane production strictly depends on water-saturated, anoxic conditions, but our peatland site experienced water level dropdowns down to -32 cm below peat surface. Under such conditions we assumed that methanotrophy would probably be a more likely process to leave  $^{13}$ C-enriched CH4 behind as compared with acetoclastic methanogenesis. However, since both reviewers raised that point, we decided to include fractionation factors for depths below -35 cm, assuming that here conditions would be anoxic and water-saturated.

In the revised manuscript please see: -page 2, lines 2-3 -page 3, lines 17-32 -page 10, lines 20-23 -page 13, line 27 to page 14, line 3 -page 17, lines 7-27 -page 18, lines 13-29 -table 2

\* The authors state that it is clearly visible that ratios of C/N, C/Mg and C/K in peat soil are decreasing from site 1 to site 4. I do not see this pattern when I look at Table 1. C/K is higher at site 2 and 3 than at site 1. C/Mg is lowest at site 1. I guess C/N ratios do not differ between site 1, 2 and 3.

*Furthermore, I guess there are no significant differences in C/P ratios between the different sites. N/P ratios are higher at site 2 than site 3 and C/Ca ratios are lowest at site 2. Please, check your data.*

You are right. This table was supposed to be a short summary of the previous study, which is now published in Soil Biology & Biochemistry (114 (2017) 131-144 <a href="http://dx.doi.org/10.1016/j.soilbio.2017.07.011">http://dx.doi.org/10.1016/j.soilbio.2017.07.011</a>). We agree that this table, providing only this little information, is not convincing, as the results from the SBB paper are quite complex.

We removed the stochiometric ratios from table 1 and provided a better description of the most important results from the SBB paper, which is providing more convincing evidence that those sites in closer vicinity to the reservoir are more affected.

Instead, information on plant species abundances at each site were added to better illustrate the vegetation gradient.

\* However, it would be great to have a look at the submitted publication or if the authors would incorporate more convincing information. Otherwise the authors cannot state that "it becomes evident that the peatland was exposed to nutrient infiltration form the water reservoir and thus elevated nutrient concentrations occurred in vicinity to the water reservoir." (P13, L2-L6) and should reformulate the whole discussion section!

- We would be pleased if you were to have a look at the paper (SBB 114 (2017) 131-144). Here, the observed differences in the transect are elaborated. Due to the complexity of the dataset only a short summary can be provided in the present manuscript. We revised our text to improve clarity and referred to the now published study. We apologize that the manuscript had not been available at the time of submitting this paper.

**\* Specific comments:**

Titel: Currently, I do not see that nutrients drive carbon dynamics at your sites.

– We changed the title:

"Differential response of carbon cycling in a continental Canadian peatland to long-term nutrient input and altered hydrological conditions"

\* Abstract: If you mention the other methods in the abstract, you should mention FTIR analysis as well.

– Done. Please see page 1, line 19.

\* P1, L17-L19: All the sites are characterized by wet conditions. These are peat soils.

– What we were trying to say was that the site 3 hollows experienced higher water levels. This sentence was anyway removed.

\* P1, L19-L20: Low 13CH4 may be caused by more hydrogenotrophically produced CH4.

\* *P1, L24: or more aceticlastically produced CH4. More labile organic matter may favor aceticlastic methanogenesis.*

- Yes, we agree, but given the unsaturated conditions and strong water table drop downs during the summer and as explained further above, we think that it would be critical to distinguish methane production pathways for the shallower depths; an influence of methanotrophic conditions is much more likely. Indeed, we think that currently not many studies have presented results from around the unsaturated zone and thus most existing studies focused on discussion pathways at greater, saturated depths.

**\* P3, L8: I do not see a gradient in nutrient availability.**

– That is probably because the provided data was in its previous presentation not convincing. The table presented only a small fraction of the data provided in the SBB paper, however, the line of argument of the SBB paper is based on a greater data set (peat ages and accumulation rates, depth profiles of element concentrations, stochiometric ratios of surface peat,  $\delta^{15}$ N-values and C/N ratios of the vegetation as well as composition of the vegetation) in order to properly describe the impact of the water reservoir on the study area and its sites. When considering only single factors (e.g. only stochiometric ratios of surface peat), their explanatory power decreases; it is the entirety of factors which shapes our knowledge of the study area. We are sorry for taking the stochiometric ratios out of context. We removed the stochiometric ratios from the table and provided a more convincing description of the study sites in terms of nutrient availability in the methods section while referring the reader to the SBB manuscript. Please see page 5, lines 24-32.

After removing the stochiometric ratios from table 1 we added more information on the vegetational gradient so that differences between sites become more obvious.

**\* P3, L10: Please, calculate apparent fractionation factors for methanogenic pathways.**

– Please see our answer above. We have calculated such fractionation factors for depths below -35 cm and added corresponding explanations. Please see page 10, lines 20-23; page 13, line 27 to page 14, line 3; and table 2.

\* P3, L16: I do not see that nutrient inputs are greatest in peatland periphery (see Table 1).

– Please see our answer above.

\* P3, L 18: Why should CH4 emissions be greatest at the graminoid dominated sites? There is no link to this hypothesis in the introduction.

– All 3 hypotheses were revised as also reviewer 1 requested for more specific hypotheses. Accordingly, most of the introduction needed to be rewritten.

\* P3, L19-21: You should also discuss CH4 production pathways.

– Done. Please see page 2, lies 17-27.

\* P4, L22-28: This paragraph is very essential for the main message of the manuscript. Unfortunately, the data do not support these statements. It would be great to see more data that support these ideas.

– We agree, the table and the statements were not convincing here. Please see our answers above on how we improved table 1 and the statements throughout paragraph 2.1.

\* P5, L8: Please, write out FTIR analysis.

– Done. Please see page 6, line 13.

\* P5, L8-L10: I am not familiar with FTIR analysis. "For pore-water samples 2 mg of oven-dried organic matter..." is that correct?

– Yes, it is correct, but maybe the sentence was a bit misleading. The pore water samples were dried in an oven until all the water was gone. Then the remaining solid material was scratched off the bottom of the sample containers and underwent FTIR analysis. We provided a better description. Please see page 6, lines 14-15.

\* RESULTS section: The presentation of the results are too cluttered. Results that are not significant are described quite often (see P10, L7, L10) and sometimes it is not clear if results are significant or not (see P10, L16-17, L19-26). It would be better to mark significant differences in the figures and to highlight significant results or only few non-significant results in text, if they really enrich and/or support the guiding questions in the manuscript.

– As the study design was indeed very complex, we ended up with very complex results. However, we very much agree with your comment and tried to improve the data presentation. Therefore, in our revised text we focused more on significant results and provided less space for results that were not significant. To mark significant differences in the figures (in particular in figures showing time series data) was difficult because sometimes site 3 differed significantly from site 4 and sometimes site 3 and 4 differed from site 1 and 2, sometimes there were no differences, sometimes there were differences for certain dates/depths etc. but not for all sites, etc. So, we believe the figures would have gotten too complex if we added additional information. To improve clarity, we re-arranged the figures 3 and 4, following also reviewer 1's suggestion. Moreover, less space was provided for non-significant results when describing the results in the text. We also tried to organize the description of the results in a way that it now always follows the same pattern. Therefore, almost the entire results section was rewritten.

\* P13, L15-L16: This is repetition of results.

- The sentence was removed.

\* P13, L19-L21: This is very speculative. Did you check FTIR ratios of inflowing water? Maybe you can provide some references.

– We agree. Unfortunately, we did not sample the inflowing water. Reviewer 1 was also concerned about this statement and provided an alternative explanation (higher productivity of the graminoid vegetation  $\rightarrow$  higher input of labile compounds), which we included into the discussion. Please see page 15, lines 13-19.

\* P13, L22-L23: I am not convinced by this statement. The difference between site 4 and the three other sites may be simply by chance.

– We understand your concern. You are right, the observed effects can have different causes. However, looking at the entire transect the peat quality found at the sites suggests a quite similar history before

dam construction at the site. So, based on our analysis the factor we can identify is the enrichment in nutrients and the concomitantly altered vegetation. Of course, site 4 thus had longest exposure to more minerotrophic conditions from intrusion of lake water. However, we would also consider this effect as anthropogenic then.

Our results are derived from an in-situ study; of course, experimental set-ups under controlled (laboratory) conditions provide more explicit results and such results can be more reliably related to certain factors. As compared to such studies, in-situ studies have short-comings indeed; however, in-situ studies are needed to verify concepts based on such controlled conditions and we think that we have taken all necessary pre-cautions to avoid misinterpretations and to not over-interpret our data. Hopefully you agree with us in terms of the significance of results derived from our study.

We provided statements in this regard in our methods (page 5, line 33 to page 6, line 2) and discussion (page 14, lines 18-22) sections.

**\* *P13, L24-L27: So, it is not the vicinity to the reservoir but the vegetation that drives carbon cycling processes?**

– We are not sure if we can provide a final answer to that question with our data. We think we are dealing with a complex interplay between vegetation, microorganisms and location factors. Site 4 and site 3 appeared to have received a similarly high amount of nutrients (well, site 4 probably received a bit more); around site 4 a dense *Myrica* belt established while at site 3 graminoids established. (The SBB paper provides information on vegetation etc.) With *Myrica* being present at the site, site 4 developed in a different way than site 3, where graminoids are established. So, we think it is both, the vicinity to the reservoir and the vegetation community that drives carbon cycling. By our study we thus also want to support that the response of a peatland to nutrient input and altered hydrological conditions may not be as simple as identified in studies with controlled variation of individual factors. Please see page 15, lines 20-24.

\* P14, L11-L19: This is repetition of results.

- The paragraph was almost entirely rewritten. Please see page 15 line 26 to page 17 line 5.

\* P14, L25: It would be great to see the data.

– Please see the SBB paper.

\* P15, L1-L2: Site 4 shows second highest CH4 release. Then you cannot state that graminoid sites show highest CH4 emissions. I would emphasize to reformulate the introduction and the hypothesis in such a manner that it becomes clearer to the reader why you have stated your hypothesis. Now, the discussions seems to be much too blurred.

- Introduction, hypotheses and discussion were rewritten and the misleading statement was removed.

\* P15, L6: What means "healthy" Sphagnum moss community?

- The mosses at site 4 showed severe signs of desiccation and thus inactivity in 2014 and 2015 during the summer months (June ~ September) and recovered afterwards. (Reduced photosynthetic activity of *Sphagnum* mosses while facing severe drought was previously observed in several studies (e.g. Alm et al., 1999, Aurela et al., 2007)). Given the pitiful appearance of *Sphagnum* mosses during the summers at site 4, and given also that *Sphagnum* covers only 60 % of the site 4 area, we concluded

that *Sphagnum* was in retreat at site 4. In contrast, at the sites 1, 2 and 3 the *Sphagnum* mosses looked green or red (whatever their natural color was) and always moist, which we then termed a "healthy" appearance of *Sphagnum* mosses. We apologize for utilizing the colloquial expression "healthy". We worked in a more appropriate description of the vegetation of our study sites into our revised manuscript, to avoid any misunderstandings. Please see page 5, lines 8-23.

\* P15, L22 – P16, L33: see major comments.

\* P17, L2 – L19: see major comments.

- Please see our answers above.

\* Figures: It would be great to have figures with higher resolution. In Figure 6, I can hardly identify the difference between the circles.

– Figure 6 is now provided in a higher resolution. Also, the figures 3 and 4 were redrawn.

\* Table1: Please, mark significant differences.

- Stochiometric ratios from table 1 were removed as explained above.

10 Correspondence to: Sina Berger (gefleckterschierling@gmx.de), Klaus-Holger Knorr (kh.knorr@uni-muenster.de)

Abstract. Peatlands play an important role in global carbon cycling, and their responses to long-term anthropogenically changed hydrologic conditions and nutrient infiltration are not well knownx While experimental manipulation studies, e.g. fertilization or water table manipulations, exist on the plot scale, only few studies have addressed such factors under in-situ conditions along gradients within larger sites. Therefore, an ecological gradient from center to periphery of a continental

- 15 Canadian peatland bordering a eutrophic water reservoir, as reflected by increasing nutrient input, enhanced water level fluctuations, and increasing coverage of vascular plants, was used for a case study of carbon cycling along a sequence of four differently altered sites. Here we monitored carbon dioxide (CO2) and methane (CH4) fluxes at the soil/atmosphere interface and dissolved inorganic carbon (DIC) and CH4 concentrations along peat profiles from April 2014 through September 2015. Moreover, we studied bulk-peat and pore-water quality and we applied δ13C-CH4 and δ13C-CO2 stable isotope abundance
- 20 analyses to examine dominant CH4 production and emission pathways during the growing season of 2015. We observed differential responses of carbon cycling at the four sites, presumably driven by abundances of plant functional types (PFTs) and vicinity to the reservoir. A shrub dominated site in close vicinity to the reservoir, was a comparably weak sink for CO2 (in 1.5 years: -1093 ±794, in 1 year: +135 ±281 g CO2 m-2 (=net release)) as compared to two graminoid-moss dominated sites and moss dominated site, (in 1.5 years: -1552 to -2260 g CO2 m-2, in 1 year: -896 to -1282 g CO2 m-2). Also, the shrub-
- 25 dominated site featured notably low DIC concentrations along peat pore-gas profiles as well as comparably 13C enriched CH4 (ô13C-CH4: -57.81 ±7.03 ‰) and depleted CO2 (ô13C-CO2: -15.85 ±3.61 ‰) in a more decomposed peat, suggesting a higher share of CH4 oxidation and differences in predominant methanogenic pathways. The graminoid-moss dominated site in closer vicinity to the reservoir featured a in comparison to all other sites by ~30 % increased CH4 emission (in 1.5 years: +61.4 ±32,

1

Deleted: however, the response of peatland carbon fluxes to anthropogenically changed hydrologic conditions and long-term infiltration of nutrients is still understudied. Along a transect of 4 study sites, spanning from largely pristine to strongly altered conditions within the Wylde Lake peatland complex in Ontario (Canada), Deleted: we

in 1 year:  $+39.86 \pm 16.81$  g CH4 m-2), and low  $\delta^{13}$ C-CH4 signatures (-62.30 \pm 5.54 ‰), indicating only low mitigation of CH4 emission by methanotrophic activity here. Methanogenesis and methanotrophy appeared to be related to the vicinity to the water reservoir: the importance of acetoclastic CH4 production apparently increased toward the reservoir, whereas the importance of CH4 oxidation increased toward the peatland center. Plant mediated transport was the prevailing CH4 emission

5 pathway at all sites even where graminoids were rare. Our study thus illustrates an accelerated carbon cycling in a strongly altered peatland with consequences for CO2 and CH4 budgets. However, our results suggest that long-term excess nutrient input does not necessarily lead to a loss of the peatland C-sink function.

**1** Introduction**

Since the end of the last glaciation, northern peatlands have played an important role in global carbon (C) cycling by storing atmospheric carbon dioxide (CO2) as peat, but also emitting significant amounts of C as methane (CH4) (Succow and Joosten, 2012). Carbon sequestration and CO2 and CH4 release are driven by numerous processes and the accumulation of peat results

- from only a small imbalance of photosynthetic carbon uptake over respiratory losses. CO2 can be released through autotrophic and heterotrophic respiration under aerobic and anaerobic conditions (Limpens et al., 2008). Controls on heterotrophic respiration have been intensively studied and depend e.g. on temperature, substrate quality, energetic constraints and other
  factors (Blodau, 2002). Methanogenesis is strictly limited to anaerobic conditions (Conrad, 2005). Due to thermodynamic controls, CH4 production is only competitive upon depletion of alternative, energetically more favorable electron acceptors for
- anaerobic respiration, such as nitrate, iron, sulfate or oxidized humics (Blodau, 2002; Klüpfel et al., 2014). CH4 is predominantly produced via two pathways: hydrogenotrophic and acetoclastic methanogenesis. During hydrogenotrophic methanogenesis  $CO_2$  is reduced to  $CH_4$ , while during acetoclastic methanogenesis acetate is split in  $CH_4$  and  $CO_2$ . These
- 20 pathways differ with respect to their discrimination against the heavier 13C-isotopes due to the kinetic isotope effect (Hoefs, 1987). Differences in the isotopic composition are thereby commonly presented as δ13C values, expressed as: δ13C = (Rsample/Rstandard 1) 1000 [‰], where R is the ratio of heavy isotope to light isotope of the samples and the respective standard. Acetoclastic methanogenesis results in δ13C-CH4 values of -65 to -50 ‰, while hydrogenotrophic methanogenesis, discriminates stronger against the heavier carbon isotope and results in δ13C-CH4 values of -110 to -60 ‰ and considerably
- 13C enriched CO2 compared to the acetocalstic pathway (Whiticar et al., 1986). Specific patterns have been observed in terms of spatial and temporal occurrence of the major CH4 production pathways, with acetoclastic methanogensis typically increasing in contribution towards the surface or within the rhizosphere (Holmes et al., 2015). On the other hand, an assignment of methanogenic pathways based on 13C signatures of CH4 can be biased by microbial oxidation of CH4. This can in particular be the case near and in the unsaturated profile where oxygen can enter by diffusion, or in the rhizosphere where plants deliver
- 30 oxygen through aerenchyma roots (Chasar et al., 2000). Upon conversion of CH4 into CO2, the residual CH4 gets enriched in 13C compared to the source CH4 (Teh et al., 2006), a process which yields similar d13C-CH4 signatures as observed upon CH4 production by the acetoclastic pathway. CH4 is released to the atmosphere by three different processes: i) through

2

**Deleted:** found that a graminoid-moss dominated site, which was exposed to wet conditions and ung-term infiltration of nutrients, was a great site of CO2 (2260 ± 480 g CO2 m2) but a great source of CH4 (61 4 ± 32 g CH4 m2). Comparably low  $\delta^{13}$ C-CH4 signatures (-62.30 ± 5.54 ‰) indicated only low mitigation of CH4 emission by methanotrophic activity here. On the contrary, a shrub dominated site, which has been subjected to similarly high moisture conditions and loads of nutrients, was a much weaker sink of CO2 (1093 ± 794 g CO2 m2) as compared with all other sites. The shrub dominated site featured notably low DIC concentrations in the peat as well as comparably 11C enriched CH4 ( $\delta^{10}$ C-CH4 - 57.81 \pm 7.03 ‰) and depleted CO2 ( $\delta^{13}$ C-CO2: -15.85 \pm 3.61 ‰) in a more decomposed and surficial aerate peat, suggesting a higher share of CH4 
[revised manuscript text omitted]
 13C isotopic composition of CO2 and CH4 with a precision of <0.16 ‰ for δ13C-CO2 and <1.15 ‰ for δ13C-CH4. The analyzer was calibrated before each measurement with two working standards of CO2 (1000 ppm, -31.07 ‰) and CH4 (1000 ppm, -42.48 ‰). Standard deviation for δ13C-CO2 was below 2 ‰ and below 4 ‰ for δ13C-CH4. Isotopic signatures were expressed in the δ-notation in ‰ versus VPDB-Standard according to Eq. 3:
- 30

5

 $\delta^{13}C = (R_{sample}/R_{standard} - 1) \cdot 1000 \ [\%]$

9

| Deleted: These latter silicon                                                                                                                                                                                                                                                                                  |
|----------------------------------------------------------------------------------------------------------------------------------------------------------------------------------------------------------------------------------------------------------------------------------------------------------------|
| Deleted: 0.5 or                                                                                                                                                                                                                                                                                                |
| Deleted: and                                                                                                                                                                                                                                                                                                   |
| Deleted: ml in 5                                                                                                                                                                                                                                                                                               |
| Deleted: depth                                                                                                                                                                                                                                                                                                 |
| Deleted: and 5                                                                                                                                                                                                                                                                                                 |
| Deleted: order to                                                                                                                                                                                                                                                                                              |
| Deleted: Samples were taken once a month from May 2015 to September 2015 with 10 and 60 ml syringes and filled in 10 respectively 40 ml crimp vials that were before flushed with $N_2$ and sealed with rubber stoppers. Silicon samplers were refilled with $N_2$ to avoid oxygen entering the system. |
| Deleted: All samplers were installed one month prior to the first                                                                                                                                                                                                                                       |

where  $R_{Sample}$  is the 13C/12C ratio of the sample and  $R_{Standard}$  is the 13C/12C ratio of the standard.

As the accuracy of  $\delta^{13}$ C-CO2 values was affected by high CH4 concentrations present in the samples, we established a correction formula to revise  $\delta^{13}$ C-CO2 values. This formula was applied for molar concentration ratios of CO2:CH4 between 0.3 and 1.5. Samples with CO2:CH4 ratios < 0.3 could not be corrected and were discarded; samples with higher ratios did not

[revised manuscript text omitted]

... [4]

**3.3 Fluxes of CO2 and CH4 at the soil/atmosphere interface, concentrations of CH4 and DIC along soil profiles during the study period**

Fluxes of CH4 and CO2 (Fig. 3, panels (c) - (f)) showed strong annual variability. CH4 fluxes (Fig. 3 (c)) were positive (fluxes from soil to atmosphere) throughout the entire study period; greatest fluxes occurred during the growing season, minute fluxes fluxes occurred during the growing season, minute fluxes fluxes occurred during the growing season for the fluxes fluxes occurred during the growing season fluxes fluxes fluxes fluxes fluxes occurred during the growing season fluxes fl

- 5 were detected during the dormant season. In general, site 3  $CH_4$  fluxes plotted above the other sites' fluxes, whereas site 4 fluxes plotted below the other sites' fluxes (except for August 16th, 2015 when a mean flux of 0.76 ± 0.58 g  $CH_4$  m-2 d-1 was detected, exceeding the fluxes measured at all other sites. During the entire study period (April 2014 through September 2015), hollows of site 3 released significantly (p < 0.001) more methane (61.4 ± 32 g  $CH_4$  m-2) than the sites 1 (41.8 ± 25.4 g  $CH_4$  m-2), 2 (44.6 ± 13.7 g  $CH_4$  m-2), and 4 (46.1 ± 35.2 g  $CH_4$  m-2); see also Fig. S5. Annual  $CH_4$  emissions from May 2014 to May
- 10 2015 were 22.18  $\pm$  8.96 at site 1, 30.66  $\pm$  7.63 at site 2, 39.86  $\pm$  16.81 at site 3, and 12.53  $\pm$  11.38 g CH4 m-2 at site 4; thus emissions at site 3 were significantly (p<0.05) higher than at site 4, but CH4 emission at sites 3 and 4 did not differ significantly from sites 1 and 2 emissions.

Fluxes of CO2 (Fig. 3 (d), (e), (f)) showed a strong seasonal variability, with highest photosynthetic uptake and highest ecosystem respiration in summer and lowest fluxes in the dormant season. Site 3 NEP plotted below all other sites' fluxes,

- 15 indicating most  $CO_2$  net uptake, whereas site 4 NEP apparently plotted above all other sites' fluxes, indicating less net uptake of  $CO_2$  if not a net emission of  $CO_2$  (Fig. 3 (d)). Regarding  $R_{geo}$  (Fig. 3 (e)), paztterns were similar at all sites. Regarding GPP (Fig. 3 (f)), site 3 plotted below all other sites, indicating highest photosynthetic uptake here, whereas site 4 mostly plotted above the other sites, indicating smallest productivity. During the study period, the cumulative NEP of hollows of site 1 was 1552 ± 652 g  $CO_2$  m-2, site 2 accumulated 1637 ± 184 g  $CO_2$  m-2, site 3 accumulated 2260 ± 480 g  $CO_2$  m-2 and site 4
- 20 accumulated 1093 ± 794 g CO2 m-2 (see Fig. S4). Thus, between May 19th, 2014 and September 23rd, 2015 site 4 accumulated significantly less CO2 as compared with the other three sites (p < 0.001), while there were no statistically significant differences in terms of CO2 uptake for the sites 1, 2 and 3. Annual cumulative NEP from May 2014 to May 2015 was -896 ± 151 g CO2 m-2 at site 1, -1023 ± 615 g CO2 m-2 at site 2, -1282 ± 361 g CO2 m-2 at site 3, while site 4 released 135 ± 281 g CO2 m-2. Annual cumulative NEP of the sites 1, 2 and 3 was significantly lower than the site 4 NEP (p < 0.05).
- 25 Interestingly, site 4 CH4, NEP and GPP fluxes differed notably between the growing seasons of 2014 and 2015. This was particularly caused by two plots, which in 2015 dramatically increased productivity and CH4 emissions as compared to the previous year (data not shown).

Concentration of  $CH_4$  along depth profiles (Fig. 4, top panels) of all sites varied strongly throughout the year; they generally increased during the growing season, reached maximum values in the winter season 2014/2015 and comparably decreased

30 during snowmelt in spring. A similar pattern was observed for DIC concentrations along depth profiles (Fig. 4, lower panels). Maximum DIC concentrations were observed below 20 cm depth in autumn 2014 and winter 2014/2015 and minimum concentrations were observed during snowmelt in March and April 2015. DIC concentrations at site 4 at all depths were overall lower and significantly decreased (p < 0.05) in comparison to all other sites from February 23rd through April 4th, 2015;

**12**

| Moved (insertion)            | [2] |
|------------------------------|-----|
| Deleted: CO 2 and |     |
| Deleted:                     |     |
| Deleted: and 4               |     |
| Deleted: a                   |     |

**Deleted:** Overall, NEP, GPP, and Reco at all sites were highest during the growing seasons. Also, CH4 fluxes were highest during the growing season, despite comparably low concentrations of dissolved CH4 in the investigated upper 50 cm of the peat profile, suggesting that exchange of gases at the peat/atmosphere interface and concentrations in the uppermost peat were decoupled.

**Deleted:** Accordingly, NEP, GPP,  $R_{eco}$  and  $CH_4$  fluxes at almost all sites were positively correlated with  $T_{water}$  (p < 0.05), however, negatively correlated with wid (lowest water tables co-occurred with highest fluxes). As suggested from visual inspection,  $R_{eco}$ , GPP and  $CH_4$  fluxes of the sites 1, 2 and 3 were in most cases negatively correlated with DIC and  $CH_4$  concentrations along soil profiles, i.e. increasing  $CO_3$  and  $CH_4$  fluxes were correlated with decreasing DIC and  $CH_4$  concentrations. Only at site 4 greater DIC and  $CO_3$  concentrations along peat profiles coincided with higher effluxes of  $CO_3$  and  $CH_4$ . Statistical results are summarized in the Table  $\{\dots, [5], Deleted: 3$  and

moreover, site 4 DIC concentrations were significantly (p < 0.05) lower than site 3 DIC concentrations on August 6th, 2014 and between April 19th through July 18th, 2015. Concentrations in the uppermost depths of both CO2 and CH4 were strongly affected by fluctuations of wtd, with strong decreases upon water table decline and vice versa (see table S4 for statistical results).

**5**

20

**3.4 Temporal and spatial variability of $\delta^{13}$ C-CO2 and $\delta^{13}$ C-CH4 -values in peat pore-gas profiles during the growing season in 2015**

Values of δ13C of the sampled CH4 in the peat ranged from -78.74 to -26.77 ‰, δ13C signatures of CO2 ranged from -25.81 to
 +4.03 ‰ (see Fig. 5). Highest δ13C-CH4 and CO2 values were measured at site 1 in 5 respectively 35 cm depth in September.
 Lowest δ13C-CH4 and CO2 values were detected at site 1 in 15 cm depth in June and at site 2 in 15 cm depth in August respectively.

Overall,  $\delta^{13}$ C-CH4 values showed an increasing trend with time from June to August in all depths. Average signatures in 5 to 35 cm depth differed significantly between sampling dates at all sites except between August and September (p < 0.05).

15 Concomitant to a decline in water table levels in August and September,  $\delta^{13}$ C-CH4 signatures shifted to less negative values in the upper 5 cm at sites 1 to 3; this shift was most distinctive at site 1 and least distinctive at site 3. At site 4, such shift occurred at 15 cm depth.

For  $\delta^{13}$ C-CO2 signatures, significant differences between some sampling dates were found at sites 1, 2 and 4 for average values in 5 to 35 cm depth. At sites 1 and 2, signatures in August and September were higher than in June and July, paralleling the trend in  $\delta^{13}$ C-CH4. At site 3 and 4, such significant shifts could not be observed.

- No significant differences between depths could be found for either  $\delta^{13}$ C-CH4 or  $\delta^{13}$ C-CO2 signatures at any site. Anyway, at sites 1 and 2,  $\delta^{13}$ C-CH4 signatures apparently increased with depth in June and July, no trend was observable at sites 3 and 4. In August and September,  $\delta^{13}$ C-CH4 signatures seemed to decrease with depth except for site 4. Values of  $\delta^{13}$ C of CO2 increased with depth except at site 1 in July and at site 2 in July and August.
- 25 Mean signatures of  $\delta^{13}$ C-CH4 at site 4 (-57.81 ±7.03 ‰) differed significantly from those at the other sites (site 1: -61.48 ±10.71 ‰, site 2: -60.28 ±5.57 ‰, site 3: -62.30 ±5.54 ‰) for the whole sampling period (p < 0.01, p < 0.05, p < 0.001).  $\delta^{13}$ C-CO2 signatures at site 3 were significantly higher than at the other sites in July (p < 0.05, p < 0.01, p < 0.01). Overall, highest mean values were found at site 1 (-12.05 ± 8.23 ‰) whereas site 4 revealed lowest  $\delta^{13}$ C-CO2 signatures (-15.85 ± 3.61 ‰).
- 30 Isotopic composition of CH4 and CO2 as determined from porewater peepers confirmed results obtained from the silicon gas samplers. Data is presented in the Fig. S5 in the supplemental information.

**13**

**Deleted:**

**Moved up [2]:** Fluxes of CO2 and CH4 (Fig. 3 and 4) showed a strong annual variability. Overall, NEP, GPP, and Reco at all sites were highest during the growing seasons. Also, CH4 fluxes were highest during the growing season, despite comparably low concentrations of dissolved CH4 in the investigated upper 50 cm of the peat profile, suggesting that exchange of gases at the peat/atmosphere interface and concentrations in the uppermost peat were decoupled. -

[... [6]

Fractionation factors  $\alpha_{\rm C}$  to characterize methanogenic pathways (according to Whiticar et al. (1986)) were calculated for water saturated, presumably anoxic conditions at -35 cm depth only (Table 2), as frequent or prevailing unsaturated conditions above this depth would rather favor methanotrophy. Given that  $\alpha_{\rm C}$  values between 1.04 and 1.055 indicate the prevalence of the acetoclastic methane production pathway, whereas  $\alpha_{\rm C}$  values higher than 1.065 support a shift towards the hydrogenotrophic

5 pathway, the acetoclastic pathway was apparently favored in July and August at the sites 1 and 2, in August at site 3 and in July, August and September at site 4, indication shift towards a higher contribution of the hydrogenotrophic pathway was only observed in June and September at site 1, and in June at site 2

**$3.5 \, \delta^{13}$ C signatures of emitted CH4 during summer 2015**

Values of δ13C of emitted CH4 ranged from -81.87 ± 3.81 and -55.61 ± 1.20 ‰ (see Fig. 6, panel (a) to (d)). Thereby, δ13CCH4 
[revised manuscript text omitted]

---

## Author Response (AR2)

Dear Reviewer. Thank you again for your very detailed review. We very much appreciate it and tried to follow your suggestions. Please read our point-by-point-answers to your comments below.

GENERAL

You have taken a good job in taking the comments of both reviewers into careful consideration and incorporating related changes to the manuscript. I feel that the conclusion drawn are now better supported by the data than in the previous version, and the title matches much better with the content.

Despite these clear improvements, I would like to still recommend major changes to the manuscript prior to its publications

As the editor decided, that a minor revision should be done, we refrained from a major revision. Nevertheless, we tried to come up with improvements for all points raised by you.

MAJOR COMMENT

Currently, the massive length of the paper (more than 11000 marks and 19 pages!), and the wordy style make it difficult to approach and digest the main message. The work would greatly benefit from shortening and focusing it better to the most relevant content.

By following your suggestions listed below, by eliminating redundant explanations and by expressing some contents in a more straightforward way, we shortened the length of the manuscript by almost one page (~ 800 words). We believe this also improved readability.

The paragraphs and sentences are often very long and loaded with details. I suggest following some simple guidelines to improve the readability: For example, the length of the paragraphs should not exceed 200-300 words and the length of sentences should not exceed three lines. Also, reading the text aloud would reveal unnecessarily complicated expressions.

Thank you for providing those guidelines. We adjusted our manuscript accordingly.

Here are some examples of complicated sentences that would need revision:
• page 1, lines 27-28
• page 3, lines 15-23
• page 4, lines 2-6

We broke the questionable sentences into two or more sentences. Please see:
page 1, line 26 to page 2, line 3;
page 3, line 14 to 21;
page 3 line 32 to page 4, line 2.

• page 12, line 5: A complicated sentence, how about 'In general, site 3 had higher and site 4 lower CH4 fluxes than the other sites, with some exceptions'? Please revise similarly the sentences on lines 14-18. The expression 'plotted below other sites' fluxes' is complicated and a simpler and more direct way to say the same thing would improve the readability a lot. Check also elsewhere in the MS.

We agree. The sentences were very complicated indeed. We simplified them. Please see:
page 11, line 24 to line 26 as well as
page 11, line 30 to page 12, line 3.

• Page 13, line 30
• Page 14, 28-31
• Page 15, lines 16-19
• Page 15, lines 26-29
• Page 18, lines 13-17
• Page 18, lines 26-29
• Page 19, lines 5-6

We broke the long and complicated sentences into two or more sentences and avoided complicated expressions. Please see:

page 13, line 14 to line 16;
page 14, line 11 to line 13;
page 15, line 1 to line 4;
page 15, line 11 to line 13;
page 17, line 18 to line 22;
page 17, line 30 to line 34;
page 18, line 10 to line 11.

In the discussion section, instead of elaborating all the results and comparing them with literature, you should focus in the results related to your hypotheses and main objectives. The abstract and concluding remarks nicely summarize the main conclusions that can be drawn from this rather complex dataset, and they can be used as guidelines for streamlining and shortening the content.

We understand your concern and thank you for the comment. The structure you are suggesting for our discussion is indeed a different writing style. We agree that this would be an alternative way of discussing our results. Nevertheless, as we refrained from a major revision, we kept the general structure of our discussion, but we shortened the discussion and improved readability by trying as much as possible to avoid complicated expressions.

MINOR COMMENTS

page 2, line 5: accelerated carbon cycling does not require the article 'an'

Corrected.

page 2, line 25: check spelling of 'acetoclastic'

Corrected.

page 2, line 26: check spelling of methanogenesis

Corrected.

page 3, line 1: remove the article before 'fast evasion'

Done.

page 3, line 13: check spelling of Mer Bleue

Corrected.

page 4, line 1: Replace coma with a full stop before 'However'

Done.

page 4, lines 17-21: Classically, hypotheses should include some explanations and argumentation on why these kind of results are expected. 'Hydrologically altered and nutrient enriched peripheral sites feature accelerated C cycling, because…' Would you be able to add something like this to each of the hypotheses?

We changed our hypotheses 1 and 2 accordingly. Please see page 4, line 14 to line 19. We left hypothesis 3 the way it was before as it refers to the hypotheses 1 and 2.

page 4, line 29: Add an article (the) before 'peat depth'.

Done.

page 5, paragraph starting on line 8: The longer site names mentioned here (e.g., shrub dominated site 4) are not used afterwards but you are using the simple names site 1, site 2 etc. The use of site names should be systematic. I am not sure if longer site names are needed. The site names were shortened.

page 5, lines 27-29: Do you mean "C/P and N/P ratios of surface peat suggested P limitation typical of fens" and "compared to typical values for bogs"?

Corrected.

page 5, line 30: An article not needed for the word 'quality'.

Article was deleted.

page 12, line 4: Should it be 'minor fluxes' instead of 'minute fluxes'?

Corrected.

Section 3.3: The very many numbers listed in this section make is very difficult to read, and the message is very much lost into details. Please revise by removing unnecessary details and focusing to the overall patterns.

Section was revised. Please see page 11, line 23 to page 12, line 20.

The sentence starting here is very long and complicated, please consider rewording and/splitting into two sentences.

Done.

Page 14, line 18: 'at all sites' instead 'at any site'

Done.

Page 14, line 19: 'in-situ' is commonly written in italics. Please check also elsewhere.

Done throughout the manuscript

Page 17, paragraph starting on line 7: This is a fully valid discussion but far too lengthy, and can be surely cut down significantly without losing useful information. Instead, the main idea would become clearer with less wordy expression.

Section was revised. Please see page 11, line 23 to page 12, line 20.

Page 20, line 5: Instead of hyphens, dashes (longer) should be used. Please check also elsewhere.

Done throughout the manuscript

**Differential response of carbon cycling to long–term nutrient input and altered hydrological conditions in a continental Canadian peatland**

[revised manuscript text omitted]

**2.7 Statistical analysis**

Statistics software R i386 version 3.1.0 was used to verify differences in organic matter quality between depths and sites. Data was tested for normal distribution (Shapiro-Wilk-Test, $\alpha = 0.05$) and homogeneity of variance (Levene-Test, $\alpha = 0.05$). In case

20 both requirements were met, we carried out a one-way ANOVA (Analysis of Variance) ($\alpha = 0.05$) with a post-hoc Tukey's Honest Significant Difference (HSD) test ($\alpha = 0.05$) to identify which depths or which sites differed significantly from each other. If either normal distribution or homogeneity of variance were not met, a Kruskal-Wallis test ($\alpha = 0.05$) with a multiple comparison test after Kruskal-Wallis ($\alpha = 0.05$) as post-hoc test was applied.

Using *RStudio* Version 0.99.902 as well as R i386 3.2.3 we examined differences in $\delta^{13}C$ values of $CO_2$ and $CH_4$, $CO_2$ and

25 $CH_4$ concentrations and cumulative emissions between the sites. Means were compared with t-Tests (if data was normally distributed) respectively Kruskal-Wallis and post hoc Wilcoxon-Mann-Whitney-Test (if data was not normally distributed), with confidence levels of $\alpha = 0.05$ for all statistical tests. Normality was tested with Shapiro-Wilk-Test ($\alpha = 0.05$) and homogeneity of variance was confirmed with a Levene-Test ($\alpha = 0.05$). Correlations between environmental variables and fluxes, concentrations and isotopic signatures were determined with Pearson's product-moment correlation for normally

30 distributed data or with Spearman's rank correlation if data was not normally distributed. With ANOVA ($\alpha = 0.05$), the effect of categorical variables on $CH_4$ fluxes and $\delta^{13}C$ values was computed.

**3 Results**

**3.1 Organic matter quality of peat and pore-water**

The highest degree of bulk peat decomposition, as indicated by the highest 1618.5/1033.5 absorption ratios, was found at site 4 between 5 and 20 cm depth (p < 0.05 in 10 and 20 cm depth, Fig 2 (a)). The 1618.5/1033.5 ratios of the sites 1–3 were not significantly different. Pore–water samples´ 1618.5/1033.5 ratios of site 3 were smallest between 5 and 20 cm depth as compared to all other sites (p < 0.05), indicating the lowest degree of decomposition of DOM here (Fig 2 (b)). Aromaticity as determined with $SUVA_{254}$ (Fig 2 (c)) did not significantly differ between sites in pore–water samples (exception: site 1 and site 3 in 20 cm depth (p = 0.033), where site 1 $SUVA_{254}$ was significantly higher than site 3 $SUVA_{254}$). The degree of humification, as expressed by HIX (Fig 2 (d)), was significantly lowest in site 3 pore–water (5 cm site 3 and 4: p = 0.026; 10 cm site 1 and 3: p = 0.014; 20 cm site 3 and 4: p = 0.020). The slope ratio E2:E3 (Fig 2 (e)), indicative of molecular size and aromaticity, did not significantly differ between sites.

**3.2 Development of wtd and $T_{water}$ during the study period**

During our study period, hollow wtd showed strong seasonal fluctuations. Maximum wtd (i.e. highest water table levels) throughout the study period were reached during snowmelt in spring 2014 (site 1: 6.94 cm, site 2: 4.99 cm, site 3: 16.26 cm, site 4: 23.18 cm above hollow surface). Minimum wtd (i.e. lowest water table levels) were reached during the summer of 2015 (site 1: 32.5 cm, site 2: 31.75 cm, site 3: 13.34, site 4: 19.11 cm below hollow surface). All sites showed similar courses of wtd, however, site 3 and site 4 water levels were generally higher than site 1 and 2 water levels (p < 0.05). The amplitude between maximum and minimum wtd at all sites was overall similar (site 1: ~39.5 cm, site 2: ~36.7 cm, site 3: ~30 cm (logger failure when water levels were lowest), site 4: ~42.3 cm). $T_{water}$ varied between ~2 °C in winter and ~16 °C in summer. Detailed courses of wtd and $T_{water}$ are presented in Fig. 3 (a) and (b).

**3.3 Fluxes of $CO_2$ and $CH_4$ at the soil/atmosphere interface and concentrations of $CH_4$ and DIC along peat profiles during the study period**

Fluxes of $CH_4$ and $CO_2$ (Fig. 3, panels (c) – (f)) showed strong annual variability. Greatest $CH_4$ emission (Fig. 3 (c)) occurred during the growing season, minor fluxes were detected during the dormant season. In general, site 3 emitted more and site 4 less $CH_4$ than the other sites with an exception on August 16th, 2015 when a mean flux of $0.76 \pm 0.58$ g $CH_4$ m$^{-2}$ d$^{-1}$ was detected at site 4, exceeding the fluxes measured at all other sites. During the entire study period, site 3 released significantly (p < 0.001) more $CH_4$ ($61.4 \pm 32$ g $CH_4$ m$^{-2}$) than the sites 1 ($41.8 \pm 25.4$ g $CH_4$ m$^{-2}$), 2 ($44.6 \pm 13.7$ g $CH_4$ m$^{-2}$), and 4 ($46.1 \pm 35.2$ g $CH_4$ m$^{-2}$); see also Fig. S5. Annual cumulative $CH_4$ emissions from May 2014 to May 2015 were $22.18 \pm 8.96$ at site 1, $30.66 \pm 7.63$ at site 2, $39.86 \pm 16.81$ at site 3, and $12.53 \pm 11.38$ g $CH_4$ m$^{-2}$ at site 4. Thus, site 3 emitted significantly (p < 0.05) more $CH_4$ than site 4, but $CH_4$ emission at sites 3 and 4 were not different from emissions at the sites 1 and 2. Site 3 had the highest negative NEP, indicating greatest $CO_2$ net uptake, whereas site 4 had the lowest negative, sometimes even positive

NEP, indicating little net uptake if not a net emission of $CO_2$ (Fig. 3 (d)). Regarding $R_{eco}$ (Fig. 3 (e)), patterns were similar at all sites. In accordance with the NEP results, GPP (Fig. 3 (f)) was lowest at site 3, indicating highest photosynthetic uptake here, whereas site 4 had the highest GPP, indicating smallest uptake. Between May, 2014 and September, 2015 site 4 accumulated significantly less $CO_2$ ($-1093 \pm 794$ g $CO_2$ m$^{-2}$, $p < 0.05$) than the other three sites ($-1552$ to $-2260$ g $CO_2$ m$^{-2}$),

5 while there were no significant differences in terms of $CO_2$ uptake for the sites 1, 2 and 3. Between May 2014 and May 2015 NEP of the sites 1, 2 and 3 was strongly negative ($-896$ to $-1282$ g $CO_2$ m$^{-2}$) compared to site 4 NEP ($+135 \pm 281$ g $CO_2$ m$^{-2}$, $p < 0.05$).

[revised manuscript text omitted]

Distinguishing $CH_4$ production pathways in peatlands using $\delta^{13}C$–signatures along depth profiles is a common approach (e.g. Holmes et al., 2015; McCalley et al., 2014; Hodgkins et al., 2014; Kotsyurbenko et al., 2004; Chasar et al., 2000). However, methanogenesis is a strictly anaerobic process and thus saturated, anoxic conditions are a prerequisite for an unbiased differentiation of pathways using $^{13}C$ only (Conrad, 1996). Methanotrophy would otherwise bias the interpretation of $^{13}C$ isotopic signatures of $CH_4$, as residual $CH_4$ gets enriched in $^{13}C$, mimicking values as observed under methanogenic conditions predominated by the acetoclastic pathway (Whiticar, 1999; Alstad and Whiticar, 2011). Indeed, summer wtd at all study sites dropped down to $-32.5$ cm (site 1), $-31.8$ cm (site 2), $-13.3$ (site 3) and $-19.1$ cm (site 4) below surface and we could thus only assume saturated, anoxic conditions below that depth. We will limit the discussion of $CH_4$ production pathways accordingly. For shallower depths, effects of under such conditions much more favorable methanotrophy can be expected to predominate: If the proportion of methanogenesis vs. methanotrophy is comparatively shifted toward methanogenesis, a relative $^{13}C$–$CH_4$ depletion would be detected, and if the proportion of methanogenesis vs. methanotrophy is comparatively shifted toward methanotrophy, a relative $^{13}C$ enrichment in $CH_4$ would be detected.

CH$_4$ oxidation was accordingly observed in the top –5 to –15 cm along our study transect during the summer months, with least negative $\delta^{13}$C–CH$_4$ values at 5 cm depth of site 1. Moreover, $\delta^{13}$C–CH$_4$ signatures at 5 cm depth of different sampling dates appeared to be most variable at the sites 1 and 2, which were also found to be drier than the sites 3 and 4, where less pronounced shifts of $\delta^{13}$C–CH$_4$ signatures occurred throughout the sampling period. However, also at the latter sites, variations

5  in $\delta^{13}$C–CH$_4$ were apparently driven by fluctuations of the water table levels, suggesting that CH$_4$ oxidation must have been an important factor throughout the dry season in summer. Another interesting finding was the strong $\delta^{13}$C–CH$_4$ signal pointing to notable CH$_4$ oxidation at 15 cm depth of site 4 in August 2015 (–39.10 ‰) as compared to more $^{13}$C depleted CH$_4$ (–57.73 ‰) in 5 cm depth. This was probably due to the particularly low CH$_4$ concentrations, suggesting an input of atmospheric CH$_4$ (~ –55 ‰) into the surface peat. Site 4 also featured the most enriched $\delta^{13}$C–CH$_4$ signatures in general, suggesting either least

10  CH$_4$ production or most CH$_4$ oxidation here. Site 3, showed the smallest variations in $\delta^{13}$C–CH$_4$ signatures throughout the sampling period, suggesting least modification of $\delta^{13}$C–CH$_4$ from oxidation here, which corresponds well with greatest CH$_4$ emissions and in general highest water levels measured at that site.

Overall lowest $\delta^{13}$C–CO$_2$ values were found at site 4, where simultaneously least negative values of $\delta^{13}$C–CH$_4$ were observed, suggesting a higher share of CO$_2$ from increased CH$_4$ oxidation. CO$_2$ generally got more enriched in $^{13}$C with depth at all sites

15  and sampling dates, as expected from ongoing fractionation by methanogenesis. Great shifts in $\delta^{13}$C–CO$_2$ values of the drier sites 1 and 2 during the entire sampling period could again be explained by increased exchange of peat derived CO$_2$ with atmospheric CO$_2$ under unsaturated conditions with dropping water tables in August.

Regarding observed ranges of $\alpha_C$ values at –35 cm depth at the sites, also a gradient in terms of the CH$_4$ production pathway along the transect of study sites became apparent. The sites 1 and 2, which experienced the lowest water tables during the

20  summer, and which were located in farthest distance to the water reservoir, featured a distinct shift from mostly hydrogenotrophic CH$_4$ production in June to acetoclastic CH$_4$ production in July and August and another shift back to hydrogenotrophic CH$_4$ production in September, with these shifts being more pronounced at site 1. This could be related to increased vascular plant activity in the growing season and concomitant substrate supply to methanogens, e.g. through exudation; an increased share of acetoclastic methanogenesis within the rhizosphere has previously been reported (Chasar et

25  al., 2000; Hornibrooket al., 2007). At the sites 3 and 4, such obvious shifts of CH$_4$ production pathways could not be observed, though; $\alpha_C$ values indicated either acetoclastic CH$_4$ production or a co–occurrence of acetoclastic and hydrogenotrophic CH$_4$ production. As acetoclastic methanogenesis is in particular supported in minerotrophic peatlands in presence of vascular plants (Alstad and Whiticar, 2011; Chasar et al., 2000), predominance of that pathway –in particular in closer vicinity to the reservoir– is not a surprising finding for Wylde Lake peatland. Indeed, under predominance of sedges, which supply labile organic matter

30  through roots, aceotclastic CH$_4$ production prevailed (Bellisario et al., 1999; Popp et al., 1999; Strom et al., 2003). However, the fact that CH$_4$ production pathways at the sites 3 and 4 (different vegetation), were similar, whereas CH$_4$ production pathways at the sites 2 and 3 (similar vegetation) were different, suggested that variation of $\alpha_C$ would rather reflect the impact of the reservoir, by either a) sustaining higher water tables, or b) increased nutrient input, than the presence of sedges at the sites.

The emitted $CH_4$ (see Fig. 6 (a) – (d)) was in general depleted in $^{13}C$ compared to the $CH_4$ in (see Fig. 5) all sampled peat layers. This suggests that the emitted $CH_4$ must have been produced in the deeper peat layers (Marushchak et al., 2016), where $\delta^{13}C$–$CH_4$ signatures were probably more depleted and during transport through plant aerenchyma, the lighter $CH_4$ could bypass oxidation. Moreover, plant–mediated transport also slightly discriminates against the $^{13}C$–$CH_4$ (Chanton, 2005), favoring more negative values of $\delta^{13}C$ in emitted $CH_4$. 
[revised manuscript text omitted]